

# Disordered non-Fermi liquid fixed point for two-dimensional metals at Ising-nematic quantum critical points

Kyoung-Min Kim[1,2][*] and Ki-Seok Kim[2,3][†]

**1** Center for Theoretical Physics of Complex Systems,
Institute for Basic Science, Daejeon 34126, Korea
**2** Department of Physics, Pohang University of Science and Technology,
Pohang, Gyeongbuk 37673, Korea
**3** Asia Pacific Center for Theoretical Physics, Pohang, Gyeongbuk 37673, Korea

[*] kmkim@ibs.re.kr , [†] tkfkd@postech.ac.kr

## Abstract

Understanding the influence of quenched random potential is crucial for comprehending the exotic electronic transport of non-Fermi liquid metals near metallic quantum critical points. In this study, we identify a stable fixed point governing the quantum critical behavior of two-dimensional non-Fermi liquid metals in the presence of a random potential disorder. By performing renormalization group analysis on a dimensional-regularized field theory for Ising-nematic quantum critical points, we systematically investigate the interplay between random potential disorder for electrons and Yukawa-type interactions between electrons and bosonic order-parameter fluctuations in a perturbative epsilon expansion. At the one-loop order, the effective field theory lacks stable fixed points, instead exhibiting a runaway flow toward infinite disorder strength. However, at the two-loop order, the effective field theory converges to a stable fixed point characterized by finite disorder strength, termed the "disordered non-Fermi liquid (DNFL) fixed point." Our investigation reveals that two-loop vertex corrections induced by Yukawa couplings are pivotal in the emergence of the DNFL fixed point, primarily through screening disorder scattering. Additionally, the DNFL fixed point is distinguished by a substantial anomalous scaling dimension of fermion fields, resulting in pseudogap-like behavior in the electron's density of states. These findings shed light on the quantum critical behavior of disordered non-Fermi liquid metals, emphasizing the indispensable role of higher-order loop corrections in such comprehension.

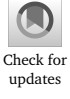

## 1  Introduction

Despite significant advancements in the understanding of metallic quantum critical points (QCPs) [1, 2], the challenge of addressing metallic QCPs in the presence of quenched disorder persists. Early renormalization group (RG) studies [3] on this issue applied the Hertz approach, wherein fermionic degrees of freedom are integrated out to derive an effective bosonic theory [4, 5]. However, this approach proves inadequate in two-dimensional (2D) systems due to uncontrolled quantum fluctuations associated with Fermi-surface electrons [6–8]. A contemporary perspective emphasizes equal treatment of fermionic and bosonic excitations [1]. Recent studies [9, 10] utilizing this modern approach revealed that random potential disorder destabilizes the clean non-Fermi liquid (CNFL) fixed point for spin-density-wave quantum criticality [11, 12]. However, finding a stable fixed point replacing this unstable fixed point, which we term a "disordered non-Fermi liquid (DNFL) fixed point," remains unresolved in these studies.

Identifying a DNFL fixed point is crucial for comprehending anomalous transport properties near metallic QCPs. For instance, strange metallic behaviors, including linear temperature dependence of electrical resistivity, are commonly observed in strongly correlated materials like heavy fermion materials, iron-pnictides, and cuprates [13–15]. Accurate modeling of these transport properties necessitates consideration of momentum relaxation processes, such as disorder scattering or Umklapp scattering. Previous studies calculated the temperature dependence of electrical resistivity by incorporating disorder scattering, using either a Boltzmann equation [16–18] or a memory matrix method [19–21]. A more recent study found a DNFL fixed point in the vicinity of a CNFL fixed point and derived scaling equations for resistivity [22], which extends the Finkelstein-type RG analysis [23–26] toward quantum criticality. Notably, this study considered a matrix-type order parameter field for the large $N$ controllability instead of vector-type quantum critical fluctuations. In this respect, the discovery of a DNFL fixed point will facilitate a reevaluation of these previous approaches and provide a more robust theoretical foundation for future advancements.

The existence of a Fermi surface in metallic systems presents a formidable challenge in the quest for the DNFL fixed point. The Fermi surface essentially reduces the effective dimensionality of the system to unity [27] or so [6], thereby classifying both interaction and disorder as "strong" or relevant in the RG sense [9, 10, 28]. The strong coupling nature of these interactions hinders the direct application of standard theoretical frameworks, such as the Hertz theory [4] or the Finkelstein theory [23], which inherently assumes a perturbative nature of the couplings. This is in stark contrast to the analysis of commonly studied semimetallic systems [29–32, 32, 33, 33–39], where both couplings are deemed irrelevant or marginally relevant, at most. Consequently, the establishment of theoretical frameworks capable of effectively addressing both interaction and disorder is imperative to propel advancements in the pursuit of the DNFL fixed point.

One promising approach to address this challenge is to begin with CNFL fixed points, where interaction effects can be systematically incorporated [7,8,12,28,40], and then introduce weak disorder. However, this strategy encounters several obstacles. Firstly, the previous observation that the disorder causes the theory to flow to strong coupling at the one-loop level [9, 10] may cast doubt on the viability of solving the problem within the weak disorder framework.

Secondly, elastic disorder scattering leads to an ultraviolet-infrared (UV-IR) mixing issue [9, 10], potentially challenging the patch description of the Fermi surface [41,42]. Finally, there is a concern that the weak disorder approach may overlook the disorder-driven localization effect responsible for Anderson localization [43].

Addressing these challenges, we establish a controlled RG framework tailored for 2D metallic QCPs in the presence of random potential disorder. We employ a dimensional-regularized field theory developed by Dalidovich and Lee [28], which allows for a systematic perturbative epsilon expansion for Yukawa couplings between electrons and bosonic order-parameter fluctuations. By reformulating this theory, we develop an RG scheme that facilitates perturbative treatments for both Yukawa couplings and random potential disorder for electrons. Key technical advancements include:

1. Single Epsilon Expansion Scheme: Tailored for regularizing loop corrections from both interaction and disorder using a unified epsilon parameter. Refer to Sec. 2.2 for detailed explanations.

2. Cutoff Regularization Scheme: Implemented to regularize divergent integrals arising from disorder, effectively avoiding UV-IR mixing. Additional details can be found in Sec. 2.3.

3. Identification of Critical Two-loop Corrections: These corrections play a key role in the emergence of the DNFL fixed point. Details are available in Sec. 3.2.

4. Large $N$ Expansion: Employed to control the strong IR enhancement factors originating from disorder, as detailed in Sec. 3.2.

In our study, we employ an effective two-patch model tailored to Ising-nematic QCPs, which are observed in various strongly correlated materials such as cuprates [44–51], pnictides [52–61], and ruthenates [62]. Our investigation focuses on the impact of the random potential disorder on two scattering channels: one involving small momentum transfer ($|\boldsymbol{q}| \ll k_F$) and the other with $2k_F$-momentum transfer ($|\boldsymbol{q}| \approx 2k_F$). Here, $k_F$ denotes the characteristic Fermi momentum of the two patches in our two-patch model. We assume short-range correlated disorder potentials characterized by a white-noise Gaussian distribution. Our focus is specifically on the weak disorder limit, utilizing a ballistic fermion propagator without an elastic scattering rate and an overdamped boson propagator with ballistic Landau damping at the CNFL fixed point.

Conducting a two-loop-level RG analysis on this model, we illustrate the appearance of a DNFL fixed point that governs the universal low-energy physics of 2D non-Fermi metals in the presence of random potential disorder. Additionally, we calculate various scaling exponents associated with this fixed point using a systematic epsilon expansion up to two-loop order.

The remainder of this paper is organized as follows. In Sec. 2, we introduce a controllable RG framework for 2D metallic QCPs, offering insights into crucial technical aspects within our approach, including the implementation of a single $\epsilon$ expansion and a cutoff regularization scheme. Moving to Sec. 3, we present the two-loop RG results, while detailed technical information is deferred to the Appendix. Transitioning to Sec. 4, we explore the robustness of our results against disorder scattering mechanisms not explicitly considered in our model and investigate potential applications of our theory to other systems. Finally, in Sec. 5, we summarize our findings.

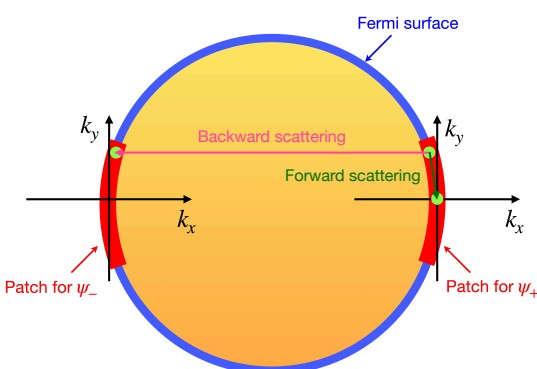

Figure 1: Schematic illustration of a two-patch model used for our renormalization group (RG) analysis. The blue circle represents the entire Fermi surface, while the red curved segments depict two antipodal patches incorporated into the effective field theory. The axes indicate the momentum coordinates of fermions near each patch. Additionally, the green and magenta arrows represent the transfer of fermions in two disorder scattering terms: forward and backward disorder scattering, respectively.

## 2 Model

### 2.1 Effective field theory

We consider 2D metallic systems in the vicinity of Ising-nematic quantum phase transitions [44–62]. The scaling behavior of these systems can be described using an effective two-patch model [7, 28]:

$$
\begin{aligned}
S_0 = &\sum_{j=1}^{N} \int \frac{dk_0 d^2\mathbf{k}}{(2\pi)^3} \bar{\Psi}_j(k)\big(ik_0\gamma_0 + i\delta_{\mathbf{k}}\gamma_1\big)\Psi_j(k) \\
&+ \frac{1}{2}\int \frac{dq_0 d^2\mathbf{q}}{(2\pi)^3} \Phi(-q)\big(q_0^2 + q_x^2 + q_y^2\big)\Phi(q) \\
&+ \sum_{j=1}^{N} \int \frac{dk_0 d^2\mathbf{k} dq_0 d^2\mathbf{q}}{(2\pi)^6} \frac{ig}{\sqrt{N}} \Phi(q)\bar{\Psi}_j(k+q)\gamma_1\Psi_j(k).
\end{aligned}
\tag{1}
$$

Here, $\Psi_j(k)$ represents a Nambu spinor given as:

$$
\Psi_j(k) = \begin{pmatrix} \psi_{+,j}(k) \\ \psi_{-,j}^{\dagger}(-k) \end{pmatrix},
\tag{2}
$$

and $\bar{\Psi}_j(k) = \Psi_j^{\dagger}(k)\gamma_0$ represents the adjoint of $\Psi_j(k)$. The gamma matrices associated with the spinor are defined as $\gamma_0 = \sigma_y$, $\gamma_1 = \sigma_x$, and $\gamma_2 = \sigma_z$, where $\sigma_{x,y,z}$ are the Pauli matrices. $\psi_{\pm,j}(k)$ represents fermion fields describing low-energy fermions on the antipodal patches of the Fermi surface (Fig. 1). These chiral fermions have different energy dispersions, represented as $k_x + k_y^2$ and $-k_x + k_y^2$ for $\psi_{+,j}(k)$ and $\psi_{-,j}(k)$, respectively. However, their energy dispersion can be represented with a single term $\delta_{\mathbf{k}} = k_x + k_y^2$ within the Nambu spinor representation. $j = 1, \cdots, N$ stands for the fermion flavor index. $\Phi(q)$ represents a scalar boson field for Ising-nematic order-parameter fluctuations or critical bosons. $g$ represents the Yukawa coupling between fermions and critical bosons.

The effective field theory in Eq. (1) exhibits two U(1) symmetries: (i) the vector symmetry with $\Psi_j(k) \to e^{i\theta_v \gamma_2}\Psi_j(k)$ and (ii) the axial symmetry with $\Psi_j(k) \to e^{i\theta_a}\Psi_j(k)$. It is essential to

recognize that the presence or absence of $\gamma_2$ in the vector and axial symmetry transformations, respectively, results from expressing the action in the Nambu spinor basis. The vector symmetry implies the conservation of the total fermion number density, denoted as $n = n_+ + n_-$. Here, $n_\pm$ represents the number density of each chiral fermion, defined as:

$$n_\pm = \sum_{j=1}^{N} \int \frac{dk_0 d^2\boldsymbol{k}}{(2\pi)^3} \left\langle \psi_{\pm,j}^\dagger(k)\psi_{\pm,j}(k) \right\rangle. \tag{3}$$

Conversely, the axial symmetry signifies the conservation of the difference between the two fermion number densities, denoted as $m = n_+ - n_-$.

We introduce two random potential terms for fermions in our effective action as follows [63]:

$$S_{\text{ran}} = \sum_{j=1}^{N} \int \frac{dk_0 d^2\boldsymbol{k} d^2\boldsymbol{q}}{(2\pi)^5} \left\{ v_f(\boldsymbol{q})\bar{\Psi}_j(k+q)\gamma_1\Psi_j(k) + v_b(\boldsymbol{q})\bar{\Psi}_j(k+q)\Psi_j^*(-k) \right\}. \tag{4}$$

Here, $v_f(\boldsymbol{q})$ and $v_b(\boldsymbol{q})$ denote forward and backward disorder scattering, respectively, wherein each term scatters fermions within the same patch or between opposite patches. Notably, $v_f(\boldsymbol{q})$ upholds both U(1) symmetries, conserving both $n$ and $m$, which correspond to separately preserving $n_+$ and $n_-$. However, $v_b(\boldsymbol{q})$ breaks the axial symmetry, conserving only $n$ but not $m$.

For disorder averaging, we assume Gaussian white-noise distributions for the random variables $v_{f/b}(\boldsymbol{r})$, specifically $\langle v_{f/b}(\boldsymbol{r})v_{f/b}(\boldsymbol{r}')\rangle = \delta(\boldsymbol{r}-\boldsymbol{r}')\Delta_{f/b}$, where $\Delta_{f/b}$ represents the variances of these distributions. Employing the replica trick [64, 65] to perform the disorder average for $S_{\text{ran}}$, we obtain the following disorder-averaged action:

$$
\begin{aligned}
S = &\sum_{a=0}^{R}\sum_{j=1}^{N} \int \frac{dk_0 d^2\boldsymbol{k}}{(2\pi)^3} \bar{\Psi}_j^a(k)\big(ik_0\gamma_0 + i\delta_{\boldsymbol{k}}\gamma_1\big)\Psi_j^a(k) + \frac{1}{2}\int \frac{dq_0 d^2\boldsymbol{q}}{(2\pi)^3}\Phi(-q)(q_0^2 + q_x^2 + q_y^2)\Phi(q) \\
&+ \sum_{a=0}^{R}\sum_{j=1}^{N} \int \frac{dk_0 d^2\boldsymbol{k} dq_0 d^2\boldsymbol{q}}{(2\pi)^6}\frac{ig}{\sqrt{N}}\Phi(q)\bar{\Psi}_j^a(k+q)\gamma_1\Psi_j^a(k) \\
&+ \sum_{a,b=0}^{R}\sum_{j=1}^{N} \int \frac{dk_0 d^2\boldsymbol{k} dk_0' d^2\boldsymbol{k}' d^2\boldsymbol{q}}{(2\pi)^8}\left\{ \frac{\Delta_f}{2}\bar{\Psi}_j^a(k+q)\gamma_1\Psi_j^a(k)\bar{\Psi}_j^b(k'-q)\gamma_1\Psi_j^b(k') \right. \\
&\left. + \frac{\Delta_b}{2}\bar{\Psi}_j^a(k+q)\Psi_j^{a*}(-k)\bar{\Psi}_j^{b*}(k'+q)\Psi_j^b(-k') \right\}. \tag{5}
\end{aligned}
$$

Here, $a, b = 0, \cdots, R$ denote the replica indices introduced for the replica trick. The disorder average transforms the random potential terms in Eq. (4) into the four-point elastic scattering terms, represented by $\Delta_f$ and $\Delta_b$.

## 2.2 Dimensional regularization

To establish a controllable RG framework, we adopt a dimensional-regularized theory [28] and tailor it to address our disorder problem. By extending the codimension of the Fermi surface from 1 to $d-1$, we adjust the fermion kinetic term to $\bar{\Psi}(k)(ik_0\gamma_0 + i\boldsymbol{k}_\perp \cdot \boldsymbol{\gamma}_\perp + i\delta_{\boldsymbol{k}}\gamma_1)\Psi(k)$, where $\boldsymbol{k}_\perp = (k_1, \cdots, k_{d-2})$ and $\boldsymbol{\gamma}_\perp = (\gamma_1, \cdots, \gamma_{d-2})$ are newly introduced momentum components and gamma matrices, respectively. All gamma matrices satisfy the Clifford algebra as $\gamma_i\gamma_j + \gamma_j\gamma_i = 2\delta_{ij}$ with $i, j = 0, \cdots, d-1$. The other components of Eq. (5) should be adjusted

accordingly. Consequently, the full action of the $(d+1)$-dimensional theory is expressed as:

$$S = \sum_{a=0}^{R}\sum_{j=1}^{N}\int \frac{dk_0 d^{d-2}\boldsymbol{k}_\perp d^2\boldsymbol{k}}{(2\pi)^{d+1}}\bar{\Psi}_j^a(k)\big(ik_0\gamma_0 + i\boldsymbol{k}_\perp\cdot\boldsymbol{\gamma}_\perp + i\delta_k\gamma_{d-1}\big)\Psi_j^a(k) + \frac{1}{2}\int \frac{dq_0 d^{d-2}\boldsymbol{q}_\perp d^2\boldsymbol{q}}{(2\pi)^{d+1}}\Phi(-q)q_y^2\Phi(q)$$

$$+ \sum_{a=0}^{R}\sum_{j=1}^{N}\int \frac{dk_0 d^{d-2}\boldsymbol{k}_\perp d^2\boldsymbol{k}dq_0 d^{d-2}\boldsymbol{q}_\perp d^2\boldsymbol{q}}{(2\pi)^{2(d+1)}}\frac{ig}{\sqrt{N}}\Phi(q)\bar{\Psi}_j^a(k+q)\gamma_{d-1}\Psi_j^a(k)$$

$$+ \sum_{a,b=0}^{R}\sum_{j=1}^{N}\int \frac{dk_0 d^{d-2}\boldsymbol{k}_\perp d^2\boldsymbol{k}dk_0' d^{d-2}\boldsymbol{k}_\perp' d^2\boldsymbol{k}'d^{d-2}\boldsymbol{q}_\perp d^2\boldsymbol{k}}{(2\pi)^{3d+2}}\Big\{\frac{\Delta_f}{2}\bar{\Psi}_j^a(k+q)\gamma_{d-1}\Psi_j^a(k)\bar{\Psi}_j^b(k'-q)\gamma_{d-1}\Psi_j^b(k')$$

$$+ \frac{\Delta_b}{2}\bar{\Psi}_j^a(k+q)\Psi_j^{*a}(-k)\bar{\Psi}_j^{*b}(k'+q)\Psi_j^b(-k')\Big\}, \tag{6}$$

where the irrelevant $q_0^2$ and $q_x^2$ terms are dropped in the bosonic action. Importantly, $\boldsymbol{k}_\perp$ is exchanged, but $k_0$ is not in the $\Delta_f$ and $\Delta_b$ terms as the disorder scattering is elastic. This anisotropic characteristic of the disorder scattering disrupts the formal $(d-1)$-dimensional symmetry within the vector space $(k_0, \boldsymbol{k}_\perp)$ as described in the clean theory [28], resulting in distinct rates of renormalization for $k_0$ and $\boldsymbol{k}_\perp$.

The quadratic part of the action in Eq. (6) is invariant under the following scaling transformation:

$$k_0 = \frac{k_0'}{b},$$

$$\boldsymbol{k}_\perp = \frac{\boldsymbol{k}_\perp'}{b},$$

$$k_x = \frac{k_x'}{b},$$

$$k_y = \frac{k_y'}{\sqrt{b}},$$

$$\Psi_j^a(k) = b^{\frac{2d+3}{4}}\Psi_j^{a\prime}(k'),$$

$$\Phi(q) = b^{\frac{2d+3}{4}}\Phi'(q'). \tag{7}$$

Under this scaling, the coupling constants undergo the following transformations:

$$g' = b^{\frac{5-2d}{4}}g,$$

$$\Delta_f' = b^{\frac{5-2d}{2}}\Delta_f,$$

$$\Delta_b' = b^{\frac{5-2d}{2}}\Delta_b. \tag{8}$$

It is crucial to note that all couplings become marginal at the upper critical dimension $d = 5/2$. Consequently, a perturbative RG analysis can be conducted by tuning $d$ as:

$$d = 5/2 - \epsilon, \tag{9}$$

where $\epsilon$ serves as a small parameter in the perturbative expansion [28]. Utilizing this expansion parameter, we investigate the scaling behavior of the theory in $d < 5/2$. Importantly, a single $\epsilon$ parameter suffices for both interaction and disorder [63] due to the anomalous scaling law $[k_x] = 2[k_y] = 1$ and $[E_f] = [E_b] = 1$ at the Ising-nematic QCP [28,66] ($[x]$ denotes the mass dimension of $x$). For general interacting disordered systems lacking such a law, a double epsilon expansion scheme is necessary [29–32, 32, 33, 33–39, 67, 68].

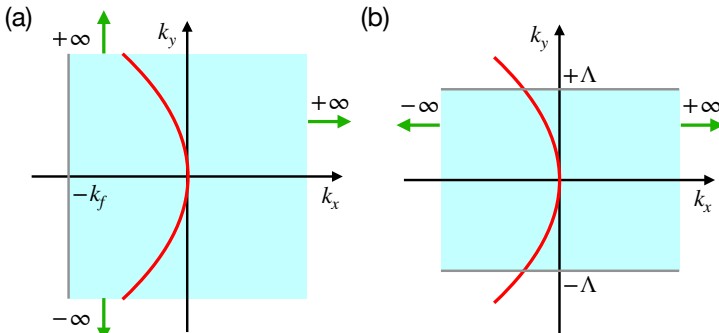

Figure 2: (a) Schematic illustration of a cutoff-regularization scheme employed in this study. While the integral region for $k_y$ extends to the infinite range $k_y \in (-\infty, \infty)$, that for $k_x$ extends to the semi-infinite range $k_x \in (-k_f, \infty)$. Here, $k_f$ represents a cutoff introduced for regularizing divergent momentum integrals arising from disorder scattering. (b) Schematic illustration of an alternative scheme, discussed in Sec. 4.1. While the integral region for $k_x$ extends to the infinite range $k_x \in (-\infty, \infty)$, that for $k_y$ extends to the finite range $k_y \in (-\Lambda, \Lambda)$. Here, $\Lambda$ represents a cutoff introduced for regularizing divergent momentum integrals. In each plot, the red line denotes a Fermi surface segment within the two-patch model.

## 2.3 Cutoff regularization for disorder scattering

The disorder scattering leads to integrals that necessitate additional cutoff regularization [Fig. 2(a)]. To illustrate this, we consider the one-loop fermion self-energy diagram resulting from the forward disorder scattering [Fig. 3(c)] [63]:

$$
\begin{aligned}
\Sigma_1(p) &= \frac{i\Delta_f}{N} \int \frac{d^{d-2}\boldsymbol{k}_\perp}{(2\pi)^{d-2}} \int_{-k_f}^{\infty} \frac{dk_x}{2\pi} \int_{-\infty}^{\infty} \frac{dk_y}{2\pi} \frac{-p_0\gamma_0 + (k_x + p_x + k_y^2)\gamma_{d-1}}{p_0^2 + \boldsymbol{k}_\perp^2 + (k_x + p_x + k_y^2)^2} \\
&= \frac{i\Delta_f}{N} \int_{-k_f}^{\infty} \frac{dk_x}{2\pi} \int_{-\infty}^{\infty} \frac{dk_y}{2\pi} \frac{\Gamma(2-\frac{d}{2})}{(4\pi)^{\frac{d-2}{2}}} \frac{-p_0\gamma_0 + (k_x + p_x + k_y^2)\gamma_{d-1}}{\left[p_0^2 + (k_x + p_x + k_y^2)^2\right]^{\frac{d-2}{2}}} .
\end{aligned}
\tag{10}
$$

Setting $k_f \to \infty$ from the outset makes the integral divergent for any $d$ since the integrand loses its dependence on $k_y$ upon integrating over $k_x$. In this scenario, the dimensional regularization fails, and epsilon poles responsible for renormalization cannot be isolated. On the other hand, adopting a finite value of $k_f$ keeps the dimensional regularization valid, and the epsilon poles can be determined from the expansion of $\Sigma_1(p)$ given as $\Sigma_1(p) \approx -iAp_0\gamma_0 - iBp_x\gamma_{d-1} - iC\gamma_{d-1} + \mathcal{O}(p^2)$. The coefficients $A$, $B$, and $C$ are explicitly given by:

$$
\begin{aligned}
A &= -\frac{\Delta_f}{N} \int_{-k_f}^{\infty} d\xi \; \nu(\xi) \frac{\Gamma(2-\frac{d}{2})}{(4\pi)^{\frac{d-2}{2}}\left(\xi^2 + p_0^2\right)^{2-\frac{d}{2}}} , \\
B &= -\frac{\Delta_f}{N} \int_{-k_f}^{\infty} d\xi \; \nu(\xi) \frac{\left[(3-d)\xi^2 - p_0^2\right]\Gamma(2-\frac{d}{2})}{(4\pi)^{\frac{d-2}{2}}\left(\xi^2 + p_0^2\right)^{3-\frac{d}{2}}} , \\
C &= \frac{\Delta_f}{N} \int_{-k_f}^{\infty} d\xi \; \nu(\xi) \frac{\Gamma(1-\frac{d}{2})}{(4\pi)^{\frac{d-2}{2}}\left(\xi^2 + p_0^2\right)^{2-\frac{d}{2}}} .
\end{aligned}
\tag{11}
$$

These expressions result from integrating out $\boldsymbol{k}_\perp$ and converting the $k_x$ and $k_y$ integrals into an energy integral over $\xi$ using a density of states given by $\nu(\xi) = \int_{-k_f}^{\infty} \frac{dk_x}{2\pi} \int_{-\infty}^{\infty} \frac{dk_y}{2\pi} \delta(\xi - \delta_{\boldsymbol{k}}) = \frac{\sqrt{\xi + k_f}}{2\pi^2}$. The epsilon poles can be obtained by expanding $A$, $B$, and $C$ with respect to $\epsilon$ as:

$$A = -\frac{\Delta_f}{N\epsilon} \frac{S\sqrt{2}}{4} + \mathcal{O}(1),$$
$$B = -\frac{\Delta_f}{N\epsilon} \frac{S\sqrt{2}}{8} + \mathcal{O}(1),$$
$$C = \frac{\Delta_f}{N\epsilon} \frac{S\sqrt{2}}{8} k_f + \mathcal{O}(1), \tag{12}$$

where $S = \frac{2}{(4\pi)^{5/4}\Gamma(5/4)}$. Importantly, the epsilon poles of $A$ and $B$, contributing to the beta functions, are independent of $k_f$. This indicates that the resulting beta functions and low-energy effective theory at the fixed point remain independent of the cutoff scale $k_f$. In other words, a UV-IR mixing does not occur in our regularization scheme. It is worth noting that the epsilon pole of $C$ proportional to $k_f$ does not renormalize the theory but merely shifts the chemical potential. This term should be eliminated with a counterterm [27], and then the theory remains cutoff-independent.

The absence of UV-IR mixing is attributed to the renormalizability of the theory at $d = 5/2$. While integrals formally display cutoff dependence, dimensional analysis dictates that epsilon poles manifest as dimensionless numerical constants due to the marginal nature of disorder scattering at $d = 5/2$ [69]. Consequently, the isolation of epsilon poles remains achievable regardless of the cutoff scale. However, caution is warranted in selecting a cutoff regularization scheme to avoid altering the upper critical dimension. Specifically, we observed that introducing a cutoff scale in the $k_y$ integral, $\int_{-\Lambda}^{\Lambda} dk_y$, modifies the upper critical dimension, leading to UV-IR mixing. Refer to Sec. 4.1 for details.

The other one-loop corrections from disorder scattering (*e.g.,* Fig. 3(f)) share the same structure as $\Sigma_1(p)$ and undergo similar treatment. However, certain two-loop corrections (*e.g.,* Fig. 3(j-k)) necessitate a distinct cutoff regularization scheme. Nevertheless, epsilon poles persist independently of the cutoff. Refer to Appendix C for further details.

Our regularization scheme is outlined as follows:

1. Introduce a cutoff in divergent momentum integrals arising from disorder scattering.

2. Compute the momentum integrals while maintaining a finite cutoff value.

3. Expand the resulting integrals using an epsilon parameter to extract epsilon poles.

4. Epsilon poles remain cutoff-independent if the cutoff regularization respects the theory's renormalizability.

It is noteworthy that we exclude cutoff regularization in the Yukawa coupling, as momentum integrals stemming from this coupling are convergent.

## 3 Renormalization group analysis

### 3.1 Renormalized action and beta functions

We adopt a field-theoretic RG approach where loop corrections are computed order by order in $\epsilon$ [69]. The divergent parts in the limit $\epsilon \to 0$ are absorbed into renormalization factors in

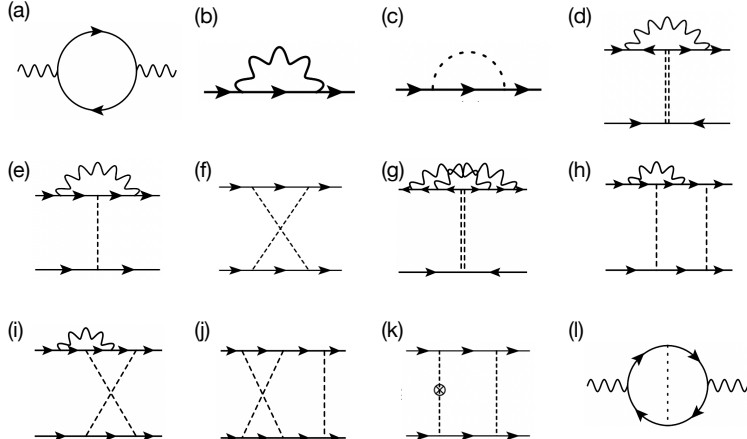

Figure 3: Selected Feynman diagrams for two-loop RG analysis. (a) One-loop self-energy corrections for bosons stemming from the Yukawa coupling. (b-c) One-loop self-energy corrections for fermions. (d) One-loop vertex corrections for backward disorder scattering. (e-f) One-loop vertex corrections for forward disorder scattering. (g) Two-loop vertex corrections for backward disorder scattering. (h-k) Two-loop vertex corrections for forward disorder scattering. (l) A two-loop boson-self energy correction leading to diffusive Landau damping. In all diagrams, the solid and wave lines stand for the fermion and boson propagators, respectively. The single and double dashed lines represent forward disorder scattering and backward disorder scattering, respectively. Refer to the Appendix for the full library of Feynman diagrams up to two-loop order.

the minimal subtraction scheme. The resulting renormalized action has the same form as the bare action in Eq. (6) while momenta and fields are renormalized as [28]:

$$k_0 = \mu^{-1}\frac{Z_2}{Z_0}k_{0,B},$$

$$\boldsymbol{k}_\perp = \mu^{-1}\frac{Z_2}{Z_1}\boldsymbol{k}_{\perp,B},$$

$$k_x = \mu^{-1}k_{x,B},$$

$$k_y = \mu^{-\frac{1}{2}}k_{y,B},$$

$$\Psi_j^a(k) = \mu^{\frac{2d+3}{4}}Z_2^{-\frac{1}{2}}\left(\frac{Z_2}{Z_0}\right)^{-\frac{1}{2}}\left(\frac{Z_2}{Z_1}\right)^{-\frac{d-2}{2}}\Psi_{j,B}^a(k),$$

$$\Phi(q) = \mu^{\frac{2d+3}{4}}Z_3^{-\frac{1}{2}}\left(\frac{Z_2}{Z_0}\right)^{-\frac{1}{2}}\left(\frac{Z_2}{Z_1}\right)^{-\frac{d-2}{2}}\Phi_B(q). \tag{13}$$

The coupling constants of the renormalized action are given by

$$g = \mu^{-\frac{\epsilon}{2}}Z_g^{-1}Z_2Z_3^{\frac{1}{2}}\left(\frac{Z_2}{Z_0}\right)^{-\frac{1}{2}}\left(\frac{Z_2}{Z_1}\right)^{-\frac{d-2}{2}}g_B,$$

$$\Delta_f = \mu^{-\epsilon}Z_{\Delta_f}^{-1}Z_2^2\left(\frac{Z_2}{Z_1}\right)^{-(d-2)}\Delta_{f,B},$$

$$\Delta_b = \mu^{-\epsilon}Z_{\Delta_b}^{-1}Z_2^2\left(\frac{Z_2}{Z_1}\right)^{-(d-2)}\Delta_{b,B}, \tag{14}$$

where $g$, $\Delta_f$, and $\Delta_b$ ($g_B$, $\Delta_{f,B}$, and $\Delta_{b,B}$) denote the renormalized (bare) coupling constants.

The RG flow of the theory is characterized by the beta functions: $\beta_g \equiv \frac{\partial g}{\partial \ln \mu}$, $\beta_{\Delta_f} \equiv \frac{\partial \Delta_f}{\partial \ln \mu}$, and $\beta_{\Delta_b} \equiv \frac{\partial \Delta_b}{\partial \ln \mu}$. Here, $\mu \to 0$ denotes the low-energy limit. Using the relationship between the bare and renormalized couplings in Eq. (14), we represent the beta functions as [63]:

$$\beta_g = g\left[ -\frac{\epsilon}{2}\bar{z} + \frac{1}{2}z + \frac{1}{4}\bar{z} - \frac{3}{4} + 2\gamma_\Psi - \gamma_g + \gamma_\Phi \right],$$

$$\beta_{\Delta_f} = \Delta_f\left[ -\epsilon\bar{z} + \frac{1}{2}\bar{z} - \frac{1}{2} + 4\gamma_\Psi - \gamma_{\Delta_f} \right],$$

$$\beta_{\Delta_b} = \Delta_b\left[ -\epsilon\bar{z} + \frac{1}{2}\bar{z} - \frac{1}{2} + 4\gamma_\Psi - \gamma_{\Delta_b} \right]. \tag{15}$$

Here, $z$ and $\bar{z}$ are the dynamical exponents, $\gamma_\Psi$ and $\gamma_\Phi$ are the anomalous dimensions of fields, and $\gamma_g$, $\gamma_{\Delta_f}$, and $\gamma_{\Delta_b}$ are the anomalous dimensions of couplings, respectively. These critical exponents are defined as

$$z = 1 + \frac{\partial \ln(Z_0/Z_2)}{\partial \ln \mu},$$

$$\bar{z} = 1 + \frac{\partial \ln(Z_1/Z_2)}{\partial \ln \mu},$$

$$\gamma_\Psi = \frac{1}{2}\frac{\partial \ln Z_2}{\partial \ln \mu},$$

$$\gamma_\Phi = \frac{1}{2}\frac{\partial \ln Z_3}{\partial \ln \mu},$$

$$\gamma_g = \frac{\partial \ln Z_g}{\partial \ln \mu},$$

$$\gamma_{\Delta_f} = \frac{\partial \ln Z_{\Delta_f}}{\partial \ln \mu},$$

$$\gamma_{\Delta_b} = \frac{\partial \ln Z_{\Delta_f}}{\partial \ln \mu}. \tag{16}$$

For the computation of the counterterms, we utilize the bare fermion propagator and the dressed boson propagator, which includes the Landau damping derived from the one-loop self-energy correction [Fig. 3(a)]. These propagators are expressed as:

$$G_0(k) = \frac{1}{i}\frac{k_0\gamma_0 + \boldsymbol{k}_\perp \cdot \boldsymbol{\gamma}_\perp + \delta_{\boldsymbol{k}}\gamma_{d-1}}{k_0^2 + \boldsymbol{k}_\perp^2 + \delta_{\boldsymbol{k}}^2},$$

$$D_1(q) = \frac{1}{q_y^2 + g^2 B_d \frac{|\boldsymbol{Q}|^{d-1}}{|q_y|}}. \tag{17}$$

Here, $B_d = \frac{\Gamma(\frac{3-d}{2})\Gamma(\frac{d}{2})^2}{2\pi(4\pi)^{(d-1)/2}\Gamma(d)}$, and $\Gamma(x)$ represents the gamma function. Refer to Appendix A for the computation of $D_1(q)$. Using these propagators is appropriate in the weak-disorder regime of our interest: $\Delta_f, \Delta_b \ll E_F$, where $E_F$ represents the Fermi energy.

We compute all renormalization factors of $Z_0, Z_1, Z_2, Z_3, Z_g, Z_{\Delta_f}$, and $Z_{\Delta_b}$ up to two-loop order [63]. Refer to the Appendix for calculation details. We insert them into Eq. (16) and solve the resulting equations order-by-order in $\epsilon$. As a result, we obtain the critical exponents

up to two-loop order as:

$$\bar{z} = \left[ 1 - 0.67\tilde{g} + 0.5\tilde{\Delta}_f + 0.5\tilde{\Delta}_b - 0.57\tilde{g}^2 - 21\tilde{\Delta}_f\sqrt{\frac{\tilde{g}}{N}} - 21\tilde{\Delta}_b\sqrt{\frac{\tilde{g}}{N}} \right]^{-1},$$

$$z = \bar{z}\left[ 1 + \tilde{\Delta}_f + \tilde{\Delta}_b - 20(\tilde{\Delta}_f + \tilde{\Delta}_b)\sqrt{\frac{\tilde{g}}{N}} \right],$$

$$\gamma_\Psi = \bar{z}\left[ 0.25\tilde{\Delta}_f + 0.25\tilde{\Delta}_b + 0.08\tilde{g}^2 \right],$$

$$\gamma_\Phi = 0,$$

$$\gamma_g = \gamma_\Psi,$$

$$\gamma_{\Delta_f} = 4\gamma_\Psi + \bar{z}\left[ -0.04\tilde{\Delta}_f + 0.75\frac{\tilde{\Delta}_b^2}{\tilde{\Delta}_f} - 3.5\tilde{g}\tilde{\Delta}_f + 5.5\tilde{\Delta}_f\sqrt{\frac{\tilde{g}}{N}} + 5.5\frac{\tilde{\Delta}_b^2}{\tilde{\Delta}_f}\sqrt{\frac{\tilde{g}}{N}} \right],$$

$$\gamma_{\Delta_b} = \bar{z}\left[ -4\tilde{g} - 0.14\tilde{\Delta}_f - 24\tilde{g}^2 - 2.3\tilde{g}\tilde{\Delta}_f - 11\tilde{\Delta}_b\sqrt{\frac{\tilde{g}}{N}} \right]. \tag{18}$$

Here, $\tilde{g}$, $\tilde{\Delta}_f$, and $\tilde{\Delta}_b$ are defined as:

$$\tilde{g} = \frac{g^{\frac{4}{3}}}{3^{\frac{2}{3}}\sqrt{6}\pi N}, \quad \tilde{\Delta}_f = \frac{\Delta_f}{2\pi^{\frac{5}{4}}\Gamma(\frac{1}{4})}, \quad \tilde{\Delta}_b = \frac{\Delta_b}{2\pi^{\frac{5}{4}}\Gamma(\frac{1}{4})}. \tag{19}$$

It is noteworthy that $\gamma_\Phi = 0$ is sustained up to the two-loop order while challenged in the third order, as noted in Ref. [70].

Substituting Eq. (18) into Eq. (15), we finally obtain the beta functions as:

$$\beta_{\tilde{g}} = \bar{z}\tilde{g}\left[ -0.67\epsilon + 0.67\tilde{g} + 0.17\tilde{\Delta}_f + 0.17\tilde{\Delta}_b + 0.57\tilde{g}^2 + 7.3\tilde{\Delta}_f\sqrt{\frac{\tilde{g}}{N}} + 7.3\tilde{\Delta}_b\sqrt{\frac{\tilde{g}}{N}} \right],$$

$$\beta_{\tilde{\Delta}_f} = \bar{z}\tilde{\Delta}_f\left[ -\epsilon + 0.33\tilde{g} - 0.21\tilde{\Delta}_f + 0.29\tilde{g}^2 + 3.5\tilde{g}\tilde{\Delta}_f + 5.0\tilde{\Delta}_f\sqrt{\frac{\tilde{g}}{N}} \right]$$
$$- \bar{z}\tilde{\Delta}_b\left[ 0.25\tilde{\Delta}_f + 0.75\tilde{\Delta}_b + 5.5\tilde{\Delta}_b\sqrt{\frac{\tilde{g}}{N}} - 11\tilde{\Delta}_f\sqrt{\frac{\tilde{g}}{N}} \right],$$

$$\beta_{\tilde{\Delta}_b} = \bar{z}\tilde{\Delta}_b\left[ -\epsilon + 4.33\tilde{g} + 0.89\tilde{\Delta}_f + 0.75\tilde{\Delta}_b + 24\tilde{g}^2 + 2.3\tilde{g}\tilde{\Delta}_f + 11\tilde{\Delta}_f\sqrt{\frac{\tilde{g}}{N}} + 22\tilde{\Delta}_b\sqrt{\frac{\tilde{g}}{N}} \right],$$

$$\bar{z} = \left[ 1 - 0.67\tilde{g} + 0.5\tilde{\Delta}_f + 0.5\tilde{\Delta}_b - 0.57\tilde{g}^2 - 21\tilde{\Delta}_f\sqrt{\frac{\tilde{g}}{N}} - 21\tilde{\Delta}_b\sqrt{\frac{\tilde{g}}{N}} \right]^{-1}. \tag{20}$$

Notably, the beta functions are expanded using an "effective" Yukawa coupling $\tilde{g} \sim g^{4/3}$, deviating from a typical factor $g^2$. This modification arises due to an IR enhancement factor of $g^{-2/3}$ resulting from Landau damping in the boson propagator [28]. Additionally, certain two-loop terms involving interaction and disorder in the beta functions exhibit a fractional power of $\tilde{g}$ as $\frac{\sqrt{\tilde{g}}}{\sqrt{N}}$, exhibiting a more pronounced IR enhancement factor of $g^{-4/3}$. As an illustration, consider the two-loop vertex correction in Fig. 3(i). After integrating out the fermion propagators, this vertex correction is expressed as $\delta\Delta_f \sim \frac{g^2\Delta_f^2}{N}\int dp \int dk_y \frac{f(p)}{|k_y|}\left(k_y^2 + g^2 B_d \frac{|K|^{d-1}}{|k_y|}\right)^{-1}$, where $p$ denotes internal momenta except for $k_y$. The extra factor of $|k_y|$ in the denominator introduces an additional factor of $g^{-2/3}$ during the integration over $k_y$, resulting in a total enhancement factor of $g^{-4/3}$. Consequently, $\delta\Delta_f$ acquires a fractional power of $\tilde{g}$, expressed as $\delta\Delta_f \sim \frac{g^{2/3}\Delta_f^2}{N} \sim \frac{\sqrt{\tilde{g}}}{\sqrt{N}}\Delta_f^2$.

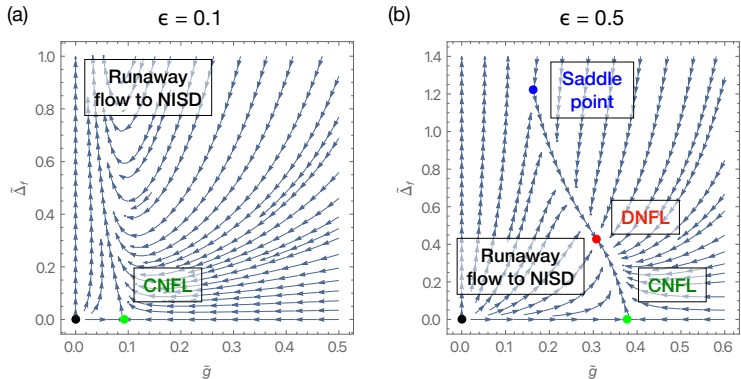

Figure 4: (a) RG flow diagram for $(\epsilon, N) = (0.1, \infty)$ in the $\tilde{g}$-$\tilde{\Delta}_f$ parameter space ($\tilde{\Delta}_b$ is set to zero). Here, $\epsilon = d_c - d$ represents the deviation of the system's actual dimension ($d$) from the upper critical dimension ($d_c = 5/2$), and $N$ represents the fermion's flavor number. The green dot at $(\tilde{g}, \tilde{\Delta}_f) = (0.093, 0)$ denotes the clean non-Fermi liquid (CNFL) fixed point [28], which becomes destabilized after the introduction of $\tilde{\Delta}_f$. The RG flow culminates in a non-interacting, strong disorder (NISD) fixed point at $(\tilde{g}, \tilde{\Delta}_f) = (0, \infty)$. (b) RG flow diagram for $(\epsilon, N) = (0.5, \infty)$. The red dot at $(\tilde{g}, \tilde{\Delta}_f) = (0.31, 0.43)$ represents a stable disordered non-Fermi liquid (DNFL) fixed point. The blue dot denotes a saddle point at $(\tilde{g}, \tilde{\Delta}_f) = (0.16, 1.22)$ separating the DNFL fixed point from the NISD fixed point. The CNFL fixed point (the green dot) is now located at $(\tilde{g}, \tilde{\Delta}_f) = (0.38, 0)$.

## 3.2 Disordered non-Fermi liquid fixed point

We commence our analysis by examining the beta functions in an infinite fermion flavor limit (*i.e.*, $N \to \infty$), where the beta functions take on simplified forms:

$$\beta_{\tilde{g}} = \bar{z}\tilde{g}\left[-0.67\epsilon + 0.67\tilde{g} + 0.17\tilde{\Delta}_f + 0.17\tilde{\Delta}_b + 0.57\tilde{g}^2\right],$$

$$\beta_{\tilde{\Delta}_f} = \bar{z}\tilde{\Delta}_f\left[-\epsilon + 0.33\tilde{g} - 0.21\tilde{\Delta}_f + 0.29\tilde{g}^2 + 3.5\tilde{g}\tilde{\Delta}_f\right] - \bar{z}\tilde{\Delta}_b\left[0.25\tilde{\Delta}_f + 0.75\tilde{\Delta}_b\right],$$

$$\beta_{\tilde{\Delta}_b} = \bar{z}\tilde{\Delta}_b\left[-\epsilon + 4.33\tilde{g} + 0.89\tilde{\Delta}_f + 0.75\tilde{\Delta}_b + 24\tilde{g}^2 + 2.3\tilde{g}\tilde{\Delta}_f\right],$$

$$\bar{z} = \left[1 - 0.67\tilde{g} + 0.5\tilde{\Delta}_f + 0.5\tilde{\Delta}_b - 0.57\tilde{g}^2\right]^{-1}. \tag{21}$$

To examine the fixed point structure of these equations, it is crucial to analyze two distinct cases separately: (i) the small $\epsilon$ case ($0 < \epsilon \leq \epsilon_c$) and (ii) the large $\epsilon$ case ($\epsilon_c < \epsilon \leq 0.5$). The threshold value $\epsilon_c = 0.40$ serves as a demarcation point, separating the two cases.

In the scenario of small $\epsilon$, the beta functions yield a single non-Gaussian fixed point, as illustrated in Fig. 4(a). This fixed point is expressed as:

$$\left(\tilde{g}^*, \tilde{\Delta}_f^*, \tilde{\Delta}_b^*\right) = \left(-0.58 + \sqrt{0.34 + 1.17\epsilon}, 0, 0\right), \tag{22}$$

which corresponds to the previously identified CNFL fixed point [28]. The variation of $\tilde{g}^*$ with respect to $\epsilon$ is illustrated by a green line in Fig. 5(a). To investigate the stability of this fixed

point, we utilize linearized beta functions:

$$\begin{pmatrix} \beta_{\tilde{g}} \\ \beta_{\tilde{\Delta}_f} \\ \beta_{\tilde{\Delta}_b} \end{pmatrix} = M \begin{pmatrix} \delta\tilde{g} \\ \delta\tilde{\Delta}_b \\ \delta\tilde{\Delta}_b \end{pmatrix} + \mathcal{O}(\delta\tilde{g}^2, \delta\tilde{\Delta}_f^2, \delta\tilde{\Delta}_b^2). \tag{23}$$

Here, $\delta\tilde{g}$, $\delta\tilde{\Delta}_f$, and $\delta\tilde{\Delta}_b$ represent the deviations of the coupling constants from their fixed point values, defined as follows:

$$\begin{aligned} \delta\tilde{g} &= \tilde{g} - \tilde{g}^*, \\ \delta\tilde{\Delta}_f &= \tilde{\Delta}_f - \tilde{\Delta}_f^*, \\ \delta\tilde{\Delta}_b &= \tilde{\Delta}_b - \tilde{\Delta}_b^*. \end{aligned} \tag{24}$$

The matrix $M$ incorporates derivatives of the beta functions to the coupling constants:

$$M = \begin{pmatrix} \frac{\partial\beta_{\tilde{g}}}{\partial\tilde{g}} & \frac{\partial\beta_{\tilde{g}}}{\partial\tilde{\Delta}_f} & \frac{\partial\beta_{\tilde{g}}}{\partial\tilde{\Delta}_b} \\ \frac{\partial\beta_{\tilde{\Delta}_f}}{\partial\tilde{g}} & \frac{\partial\beta_{\tilde{\Delta}_f}}{\partial\tilde{\Delta}_f} & \frac{\partial\beta_{\tilde{\Delta}_f}}{\partial\tilde{\Delta}_b} \\ \frac{\partial\beta_{\tilde{\Delta}_b}}{\partial\tilde{g}} & \frac{\partial\beta_{\tilde{\Delta}_b}}{\partial\tilde{\Delta}_f} & \frac{\partial\beta_{\tilde{\Delta}_b}}{\partial\tilde{\Delta}_b} \end{pmatrix} \Bigg|_{(\tilde{g},\tilde{\Delta}_f,\tilde{\Delta}_b)=(\tilde{g}^*,\tilde{\Delta}_f^*,\tilde{\Delta}_b^*)}. \tag{25}$$

Substituting Eqs. (21) and (22) into Eq. (25), we calculate the eigenvalues of $M$ at the CNFL fixed point as: $\frac{0.67\tilde{g}^*+1.14(\tilde{g}^*)^2}{1-0.67\tilde{g}^*-0.57(\tilde{g}^*)^2}$, $\frac{-\epsilon+0.33\tilde{g}^*+0.29(\tilde{g}^*)^2}{1-0.67\tilde{g}^*-0.57(\tilde{g}^*)^2}$, and $\frac{-\epsilon+4.33\tilde{g}^*+24(\tilde{g}^*)^2}{1-0.67\tilde{g}^*-0.57(\tilde{g}^*)^2}$, which govern the RG flow of $\delta\tilde{g}$, $\delta\tilde{\Delta}_f$, and $\delta\tilde{\Delta}_b$ in the vicinity of the fixed point. Here, the value of $\tilde{g}^*$ is specified in Eq. (22). The negativity of the second eigenvalue signifies the relevance of $\delta\tilde{\Delta}_f$, while $\delta\tilde{g}$ and $\delta\tilde{\Delta}_b$ are deemed irrelevant as indicated by their positive eigenvalues. Consequently, introducing $\delta\tilde{\Delta}_f$ destabilizes the CNFL fixed point, ultimately driving the theory towards an infinite-disorder regime, as depicted in the left top in Fig. 4(a):

$$\left(\tilde{g}^*, \tilde{\Delta}_f^*, \tilde{\Delta}_b^*\right) = \left(0, \infty, 0\right), \tag{26}$$

which we term a "non-interacting, strong disorder (NISD) fixed point." As a result, we deduce the absence of a stable fixed point in the small $\epsilon$ scenario.

In the large $\epsilon$ scenario, the beta functions yield three non-Gaussian fixed points, as illustrated in Fig. 4(b). One is the unstable CNFL fixed point given by Eq. (22). The other two are determined by solving the following cubic equation:

$$\tilde{g}^3 + 1.07\tilde{g}^2 - (0.10 + 1.16\epsilon)\tilde{g} + 0.15\epsilon = 0. \tag{27}$$

One of the two fixed points corresponds to a DNFL fixed point, which is given by:

$$\begin{aligned} \tilde{g}^* &= -\frac{1}{3}\left(1.07 + \xi C + \frac{A_0}{\xi C}\right), \\ \tilde{\Delta}_f^* &= 3.94\epsilon - 3.94\tilde{g}^* - 3.35(\tilde{g}^*)^2, \\ \tilde{\Delta}_b^* &= 0. \end{aligned} \tag{28}$$

Here, $\xi$, $A_0$ and $C$ are defined as follows:

$$\begin{aligned} \xi &= \frac{-1+\sqrt{3}i}{2}, \\ A_0 &= 1.44 + 3.48\epsilon, \\ A_1 &= 3.41 + 15.22\epsilon, \\ C &= \sqrt[3]{\frac{A_1 + \sqrt{A_1^2 - 4A_0^3}}{2}}. \end{aligned} \tag{29}$$

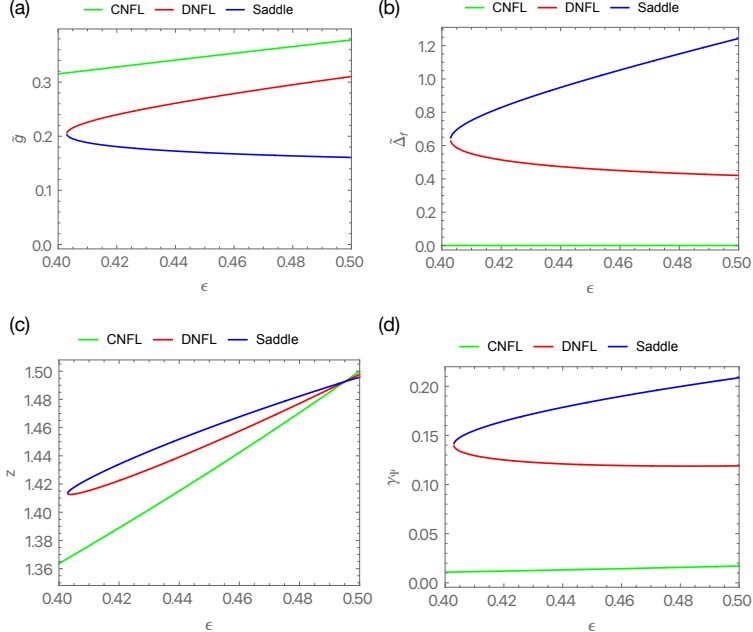

Figure 5: (a-b) Depiction of the values of $\tilde{g}$ and $\tilde{\Delta}_f$ at the three non-Gaussian fixed points, illustrated in Fig. 4(b), as a function of $\epsilon$. (c-d) Depiction of the values of the dynamical critical exponent ($z$) and the anomalous dimension of fermion fields ($\gamma_\Psi$) at the three fixed points as a function of $\epsilon$.

By substituting Eqs. (21) and (28) into Eq. (25), it is straightforward to show that all eigenvalues of $M$ have positive real parts, *i.e.*, all perturbations $\delta\tilde{g}$, $\delta\tilde{\Delta}_f$, and $\delta\tilde{\Delta}_b$ are deemed irrelevant at the fixed point, indicating that this fixed point is stable. The stable nature is also visible in the RG flow diagram, as depicted by the red dot in Fig. 4(b).

The variations of $\tilde{g}^*$ and $\tilde{\Delta}_f^*$ with respect to $\epsilon$ are illustrated by red lines in Fig. 5(a) and (b), respectively. Our findings reveal that $\tilde{g}^*$ increases as $\epsilon$ rises, while $\tilde{\Delta}_f^*$ displays an opposing decreasing trend. One possible explanation for this behavior is that the increase in $\epsilon$ leads to the growth of $\tilde{g}^*$, followed by a subsequent reduction in $\tilde{\Delta}_f^*$ due to an amplified screening effect within the term $3.5\tilde{g}\tilde{\Delta}_f$.

The other fixed point is found to be:

$$\tilde{g}^* = -\frac{1}{3}\left(1.07 + \xi^2 C + \frac{A_0}{\xi^2 C}\right),$$
$$\tilde{\Delta}_f^* = 3.94\epsilon - 3.94\tilde{g}^* - 3.35(\tilde{g}^*)^2,$$
$$\tilde{\Delta}_b^* = 0,$$

(30)

where $\xi$, $A_0$, and $C$ are given in Eq. (29). The variations of $\tilde{g}^*$ and $\tilde{\Delta}_f^*$ with respect to $\epsilon$ are illustrated by blue lines in Fig. 5(a) and (b), respectively. By substituting Eqs. (21) and (30) into Eq. (25), specifically for $\epsilon = 0.5$, we determine the eigenvalues of $M$ as $-0.11$, $0.45$, and $1.5$. The corresponding eigenvectors, or scaling fields, are found to be $-0.089\delta\tilde{g} + \delta\tilde{\Delta}_f$, $0.047\delta\tilde{g} + \delta\tilde{\Delta}_f$, and $\tilde{\Delta}_b$. Notably, the first scaling field is relevant, while the other two are irrelevant, indicating the saddle point nature of this fixed point. Consequently, we deduce that this fixed point represents a demarcation point between the DNFL fixed point from the NISD fixed point, as depicted by the blue dot in Fig. 4(b), embodying a critical surface separating these fixed points.

Based on these discoveries, we conclude that the DNFL fixed point, as presented in Eq. (28), governs the quantum critical behavior observed in 2D metallic systems near Ising-nematic QCPs. Furthermore, our investigations reveal that the previously identified CNFL fixed point, given in Eq. (22), loses stability in the presence of random potential disorder, limiting its significance to an ideal clean limit. Additionally, considering a large $\epsilon$ value (*i.e.*, $\epsilon \geq \epsilon_c$, which encompasses the physical value $\epsilon = 0.5$) proves crucial for comprehending the critical behavior at the QCPs, despite the formal classification of $\epsilon$ as a small expansion parameter.

### 3.3 Role of two-loop corrections

At the one-loop order, the beta functions, as presented in Eq. (20), are simplified as [63]:

$$
\beta_{\tilde{g}} = \bar{z}\tilde{g}\left[ -0.67\epsilon + 0.67\tilde{g} + 0.17\tilde{\Delta}_f + 0.17\tilde{\Delta}_b \right],
$$

$$
\beta_{\tilde{\Delta}_f} = \bar{z}\tilde{\Delta}_f\left[ -\epsilon + 0.33\tilde{g} - 0.21\tilde{\Delta}_f \right] - \bar{z}\tilde{\Delta}_b\left[ 0.25\tilde{\Delta}_f + 0.75\tilde{\Delta}_b \right],
$$

$$
\beta_{\tilde{\Delta}_b} = \bar{z}\tilde{\Delta}_b\left[ -\epsilon + 4.33\tilde{g} + 0.89\tilde{\Delta}_f + 0.75\tilde{\Delta}_b \right],
$$

$$
\bar{z} = \left[ 1 - 0.67\tilde{g} + 0.5\tilde{\Delta}_f + 0.5\tilde{\Delta}_b \right]^{-1}. \tag{31}
$$

These one-loop beta functions lack the DNFL fixed point for any $\epsilon$, instead exhibiting a runaway flow to the NISD fixed point. Thus, it is evident that two-loop corrections play a pivotal role in the emergence of the DNFL fixed point, highlighting the necessity of considering them for its identification.

To delve deeper into this aspect, we note that among the various two-loop order terms outlined in Eq. (21), the presence of $3.5\tilde{g}\tilde{\Delta}_f$ in $\beta_{\tilde{\Delta}_f}$ is crucial for the screening of $\tilde{\Delta}_f$, as its absence results in the disappearance of the DNFL fixed point. This pivotal term arises from two-loop order vertex corrections stemming from the Yukawa coupling, as illustrated in Fig. 3(h-i). In contrast, the contribution from the one-loop correction, depicted in Fig. 3(e), does not impact the beta functions due to cancellation with fermion self-energy corrections (Fig. 3(b)), as dictated by the Ward identity. Consequently, the two-loop corrections represent the leading screening effect for $\tilde{\Delta}_f$ within loop expansions.

For the screening term $3.5\tilde{g}\tilde{\Delta}_f$ to stabilize the DNFL fixed point, an additional condition of $\epsilon > \epsilon_c = 0.40$ must be met. To elucidate this, let's suppose the system at the CNFL fixed point, where the value of $\tilde{g}$ is given by $\tilde{g}^* = -0.58 + \sqrt{0.34 + 1.17\epsilon}$. For the screening term to dominate over the one-loop antiscreening term of $-0.21\tilde{\Delta}_f$, and thus provide an overall screening effect, $\epsilon$ must be large enough to satisfy $3.5(-0.58 + \sqrt{0.34 + 1.17\epsilon}) > 0.21$. If this condition is met, $\tilde{\Delta}_f$ can have a fixed point value: $\tilde{\Delta}_f^* = \frac{\epsilon - 0.33\tilde{g}^* - 0.29(\tilde{g}^*)^2}{-0.21 + 3.5\tilde{g}^*}$, as derived from setting $\beta_{\tilde{\Delta}_f} = 0$. However, due to the additional screening effect from $0.17\tilde{\Delta}_f^*$ in $\beta_{\tilde{g}}$, the value of $\tilde{g}^*$ is lowered from its CNFL fixed point value. This reduction in $\tilde{g}^*$ results in further adjustments to $\tilde{\Delta}_f^*$, creating a feedback loop. If this adjustment between $\tilde{g}^*$ and $\tilde{\Delta}_f^*$ can bring $\beta_{\tilde{g}} = 0$, the DNFL fixed point could be stable. Otherwise, the $0.17\tilde{\Delta}_f^*$ term in $\beta_{\tilde{g}}$ could drive $\tilde{g}^*$ to vanish, leading to a runaway flow towards the NISD fixed point. It turns out that with $\epsilon > \epsilon_c$, the necessary adjustments can be achieved, stabilizing the DNFL fixed point.

In contrast, $\tilde{\Delta}_b$ begins to acquire the screening effect from the one-loop order correction, presented in Fig. 3(d). The two-loop corrections, such as Fig. 3(g), primarily enhance this screening effect. Consequently, the RG flow to $\tilde{\Delta}_b = 0$ appears consistently in both one-loop and two-loop order analyses.

### 3.4 Physical quantities at fixed points

#### 3.4.1 Critical exponents

In the limit $N \to \infty$, the critical exponents $z$ and $\gamma_\Psi$, as presented in Eq. (18), exhibit simplified forms:

$$z = \frac{1 + \tilde{\Delta}_f + \tilde{\Delta}_b}{1 - 0.67\tilde{g} + 0.5\tilde{\Delta}_f + 0.5\tilde{\Delta}_b - 0.57\tilde{g}^2}\,,$$
$$\gamma_\Psi = \frac{0.25\tilde{\Delta}_f + 0.25\tilde{\Delta}_b + 0.08\tilde{g}^2}{1 - 0.67\tilde{g} + 0.5\tilde{\Delta}_f + 0.5\tilde{\Delta}_b - 0.57\tilde{g}^2}\,. \tag{32}$$

Upon substituting Eq. (22) into Eq. (32), we derive the critical exponents at the CNFL fixed point:

$$z = \frac{3}{3 - 2\epsilon}\,,$$
$$\gamma_\Psi = \frac{0.16 + 0.28\epsilon - 0.28\sqrt{0.34 + 1.17\epsilon}}{3 - 2\epsilon}\,. \tag{33}$$

These expressions are illustrated by green lines in Fig. 5(c) and (d), respectively. For $d = 2$ or $\epsilon = 0.5$, these values simplify to:

$$z = 1.5\,,$$
$$\gamma_\Psi = 0.017\,. \tag{34}$$

The critical exponents at the DNFL fixed point can be obtained by substituting Eq. (28) into Eq. (32), although the resulting expressions are too intricate to be presented. Their values are depicted by red lines in Fig. 5(c) and (d), respectively. For $d = 2$ or $\epsilon = 0.5$, these values simplify to:

$$z = 1.5\,,$$
$$\gamma_\Psi = 0.13\,. \tag{35}$$

Notably, $\gamma_\Psi = 0.13$ at the DNFL fixed point significantly exceeds $\gamma_\Psi = 0.017$ at the CNFL fixed point. This discrepancy arises from the substantial correction contributed by the forward scattering $\tilde{\Delta}_f$ at the DNFL fixed point.

#### 3.4.2 Fermion's density of states

We compute the fermion's density of states resorting to the following formula [7]:

$$N(\omega) = -\frac{1}{\pi} \int \frac{d^2\boldsymbol{k}}{(2\pi)^d} \text{Im}\Big[\text{tr}\big\{G(ik_0 \to \omega + i0^+, \boldsymbol{k})\big\}\Big]\,, \tag{36}$$

where $G(k)$ stands for the full fermion's Green function. The scaling behavior of $G(k)$ is described by the following scaling function:

$$G(k, \mu, \boldsymbol{F}) = \frac{1}{\mu^{2\gamma_\Psi}|\delta_{\boldsymbol{k}}|^{1-2\gamma_\Psi}} g(k_0/|\delta_{\boldsymbol{k}}|^z)\,, \tag{37}$$

which can be obtained by solving the Callan-Symanzik equation

$$\Big[k \cdot \boldsymbol{\nabla}_k - \boldsymbol{\beta_F} \cdot \boldsymbol{\nabla}_F + 1 - 2\gamma_\Psi\Big]G(k, \mu, \boldsymbol{F}) = 0\,,$$

Table 1: Critical exponents of the Ising-nematic quantum criticality in two-dimensional metals. The DNFL and CNFL fixed points primarily differ in the exponent $a$, which describes the pseudogap-like behavior of the fermion's density of states, as defined in Eq. (39). $\alpha$, $\beta$, $\gamma$, and $\delta$ represent the critical exponents for thermodynamic quantities defined in Eq. (43). $\nu$ and $z$ are the correlation length and dynamical exponents, respectively. For comparison, the exponents for the fermi liquid (FL) phase are also provided.

|      | $a$   | $\alpha$ | $\beta$ | $\gamma$ | $\delta$ | $\nu$ | $z$ |
|------|-------|----------|---------|----------|----------|-------|-----|
| DNFL | 0.17  | $-1/2$   | 3/4     | 1        | 7/3      | 1     | 3/2 |
| CNFL | 0.023 | $-1/2$   | 3/4     | 1        | 7/3      | 1     | 3/2 |
| FL   | 0     | -        | -       | -        | -        | -     | 1   |

where $k \cdot \nabla_k \equiv z k_0 \frac{\partial}{\partial k_0} + \bar{z} \boldsymbol{k}_\perp \cdot \nabla_{\boldsymbol{k}_\perp} + \delta_{\boldsymbol{k}} \frac{\partial}{\partial \delta_{\boldsymbol{k}}}$, $\boldsymbol{F} \equiv (\tilde{g}, \tilde{\Delta}_f, \tilde{\Delta}_b)$ and $\nabla_{\boldsymbol{F}} \equiv \left( \frac{2}{3} \frac{\partial}{\partial \tilde{g}}, \frac{\partial}{\partial \tilde{\Delta}_f}, \frac{\partial}{\partial \tilde{\Delta}_b} \right)$. Refer to the Appendix E for the derivation. Substituting Eq. (37) into Eq. (36), we obtain

$$N(\omega) \sim \int_{-\infty}^{\infty} dk_x \int_{-\Lambda}^{\Lambda} dk_y \frac{1}{|\delta_{\boldsymbol{k}}|^{1-2\gamma_\Psi}} g\left( \frac{|\omega|}{|\delta_{\boldsymbol{k}}|^z} \right) \sim |\omega|^a, \tag{38}$$

where the exponent $a$ is given by

$$a = \frac{2\gamma_\Psi}{z}. \tag{39}$$

Note that the $k_y$-integral should be regularized with a cutoff $\Lambda$ so that it does not contribute to the scaling [7].

We evaluate the exponent $a$ by utilizing the values of $z$ and $\gamma_\Psi$ in Eqs. (34) and (35) for the CNFL and DNFL fixed points, respectively. At the CNFL fixed point, we obtain $a = 0.023$, which is almost indistinguishable from that of an ordinary non-interacting fermion gas, $a = 0$. On the other hand, at the DNFL fixed point, we obtain $a = 0.17$, which is anomalously large due to the sizable correction from $\gamma_\Psi = 0.13$. As a result, the fermion's density of states is substantially suppressed near the Fermi energy as $N(\omega) \sim |\omega|^{0.17}$ at the DNFL fixed point.

### 3.4.3 Thermodynamic quantities

We consider the following additional coupling terms [63]:

$$\delta S = \int d^{d+1}x \left[ r\Phi^2(x) - h\Phi(x) - hN(x) \right], \tag{40}$$

where $r$ is the tuning parameter for the quantum phase transition, $h$ is an external field, and $N(x) = \bar{\Psi}(x)\gamma_1\Psi(x)$. Note that $h$ is coupled to both boson field $\Phi(x)$ and fermion field $\Psi(x)$ since they have the same symmetries [71].

Considering $\delta S$, we find the homogeneity relation of a free energy density $f \equiv -(T/V)\ln \int \mathcal{D}\Psi \mathcal{D}\Phi e^{-S}$ as

$$f(r,h) = b^{-D} f(r b^{1/\nu}, h b^{y_h}, T b^z), \tag{41}$$

where $b$ is the scaling parameter that scales a system size $L$ as $L \to bL$ or temperature $T$ as $T \to b^z T$. Here, $D$ is the effective scaling dimension of the space-time given as

$$D = z + (d-2)\bar{z} + 1. \tag{42}$$

When counting $D$, we should ignore momentum coordinate $k_y$ since it becomes redundant when the whole Fermi surface is considered [28]. $\nu = [r]$ is the correlation length exponent. $y_h = [h]$ represents the scaling dimension of $h$. We find $y_h$ as $y_h = \frac{1}{2}(D+1) - \gamma_\Phi$ from the coupling term for $\Phi$ or $y_h = 1 - 2\gamma_\Psi$ from that for $\Psi$. It turns out that the former has a larger value than the latter at the DNFL fixed point. This indicates that the former determines the leading critical behavior, as explicit calculations confirm. Refer to the Appendix E for further details. Therefore, we conclude $y_h = \frac{1}{2}(D+1) - \gamma_\Phi$.

From Eq. (41), we find thermodynamic quantities showing critical behaviors as

$$
\begin{aligned}
c_v &\equiv -\frac{\partial^2 f}{\partial r^2} \sim |r|^{-\alpha}, \\
m &\equiv -\frac{\partial f}{\partial h}\bigg|_{h \to 0} \sim (-r)^\beta, \\
\chi &\equiv \frac{\partial^2 f}{\partial h^2}\bigg|_{h \to 0} \sim |r|^{-\gamma}, \\
h &\propto |m|^\delta,
\end{aligned}
\tag{43}
$$

where $c_v$ is the specific heat, $m$ is the Ising-nematic order parameter, and $\chi$ is the susceptibility. The exponents are given by

$$
\begin{aligned}
\alpha &= 2 - D\nu, \\
\beta &= \frac{\nu}{2}(D - 1 + 2\gamma_\Phi), \\
\gamma &= (1 - 2\gamma_\Phi)\nu, \\
\delta &= \frac{D + 1 + 2\gamma_\Phi}{D - 1 - 2\gamma_\Phi}.
\end{aligned}
\tag{44}
$$

We evaluate the exponents by focusing on $d = 2$. Up to two-loop order, we find $D = z + 1 = 5/2$, $\nu = 1$, and $\gamma_\Phi = 0$. Substituting them into Eq. (44), we obtain $\alpha = -1/2$, $\beta = 3/4$, $\gamma = 1$, and $\delta = 7/3$. The calculated critical exponents are summarized in Table 1.

## 3.5 DNFL fixed point at a finite $N$

We expand our RG analysis to finite values of $N$. Figure 6 showcases our numerical computation results obtained by solving Eq. (20) numerically. Our findings reveal the persistence of the DNFL fixed point for $N < N_c$, where $N_c$ denotes a threshold value. Beyond this threshold, the DNFL fixed point destabilizes, and the RG flow exhibits a runaway flow toward the NISD fixed point. The threshold value $N_c$ tends to increase with $\epsilon$, as delineated by the red lines in each panel of Fig. 6, separating the DNFL and NISD regions.

Within the DNFL region, we observe that for a given $N$, the value of $\tilde{g}$ at the DNFL fixed point increases with $\epsilon$ [Fig. 6(a)], while the value of $\tilde{\Delta}_f$ shows an opposing decreasing trend [Fig. 6(b)]. These trends align with those observed in the infinite-$N$ case, as illustrated in Fig. 5(a-b). Furthermore, for a given $\epsilon$, the values of $\tilde{g}$ and $\tilde{\Delta}_f$ exhibit opposing trends of increase and decrease, respectively, as $N$ increases. One possible explanation for this behavior is that the increase in $N$ amplifies the screening effect within the term $7.3\tilde{\Delta}_f\sqrt{\tilde{g}/N}$ for $\tilde{g}$, leading to a reduction in $\tilde{g}$. This reduction, in turn, amplifies $\tilde{\Delta}_f$ by weakening the screening term $3.5\tilde{g}\tilde{\Delta}_f$, leading to an increase in $\tilde{\Delta}_f$.

Additionally, we compute the values of $z$ and $\gamma_\Psi$ at the DNFL fixed point, as presented in Fig. 6(c-d). Our findings reveal that the value of $z$ increases with an increase in $\epsilon$ while remaining largely unaffected by $N$ [Fig. 6(c)]. Conversely, the value of $\gamma_\Psi$ shows increasing trends as $N$ increases, while remaining largely unaffected by $\epsilon$ [Fig. 6(d)]. Notably, these

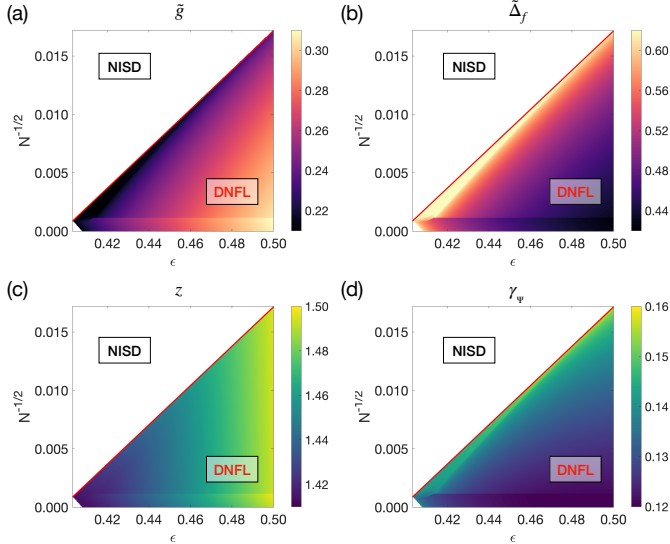

Figure 6: (a-d) Depiction of the values of $\tilde{g}$, $\tilde{\Delta}_f$, $z$, and $\gamma_{\Psi}$ at the DNFL fixed point as a function of $(\epsilon, N^{-1/2})$. Each panel utilizes a color scale to represent the corresponding quantity, with the empty region denoting the absence of the DNFL fixed point, characterized instead by a runaway RG flow toward the NISD fixed point. The red line, marking $N^{-1/2} = 0.168\epsilon - 0.0669$, delineates the boundary separating these two regions.

variations mirror those of $\tilde{g}$ and $\tilde{\Delta}_f$, respectively. These trends suggest that $\tilde{g}$ and $\tilde{\Delta}_f$ are primary factors in determining the values of $z$ and $\gamma_{\Psi}$, respectively, through the relationship presented in Eq. (18).

## 3.6 Stability of DNFL fixed point

### 3.6.1 Higher-order corrections

It's important to examine the persistence of the DNFL fixed point against higher-order corrections since its existence may not be guaranteed by taking a small $\epsilon$ limit. This contrasts with the CNFL fixed point, where such a limit can make higher-order corrections arbitrarily small, ensuring its persistence. However, the existence of the DNFL fixed point relies on the condition $\epsilon > \epsilon_c$, so the small $\epsilon$ limit cannot be taken in this case. Consequently, one must verify the robustness of the DNFL fixed point by explicitly calculating pertinent corrections in each order. Here, we explore the stability of the DNFL fixed point against third-loop-order corrections based on a scenario inferred from our two-loop-order results.

We begin by examining possible three-loop-order corrections computed using the fermion and boson propagators presented in Eq. (17). We assume that three-loop fermion self-energy corrections consist exclusively of terms proportional to $\tilde{g}^3$. Terms involving a mix of $\tilde{g}$ and $\tilde{\Delta}_f$ are expected to exhibit strong IR enhancement factors and are therefore disregarded in the large $N$ limit (see the Appendix C.2 for further discussion). Three-loop-order vertex corrections for forward scattering can be expressed as $\tilde{g}\tilde{\Delta}_f^2$. In both cases, pure disorder contributions, $\tilde{\Delta}_f^3$, are expected to lack epsilon poles (see the Appendices C.2 and D.1 for more information). Three-loop-order vertex corrections for the Yukawa coupling are disregarded due to their cancellation with the fermion self-energy contribution.

The boson propagator undergoes alterations due to the two-loop self-energy correction [Fig. 3(l)] as follows: $D_2(q) = \left[ q_y^2 + g^2 B_d \frac{|q|^{d-1}}{|q_y|} - \Pi_2(q) \right]^{-1}$, where $\Pi_2(q) = -g^2 \tilde{\Delta}_f \tilde{B}_d \frac{|q|^{2d-3}}{|q_y|^2}$

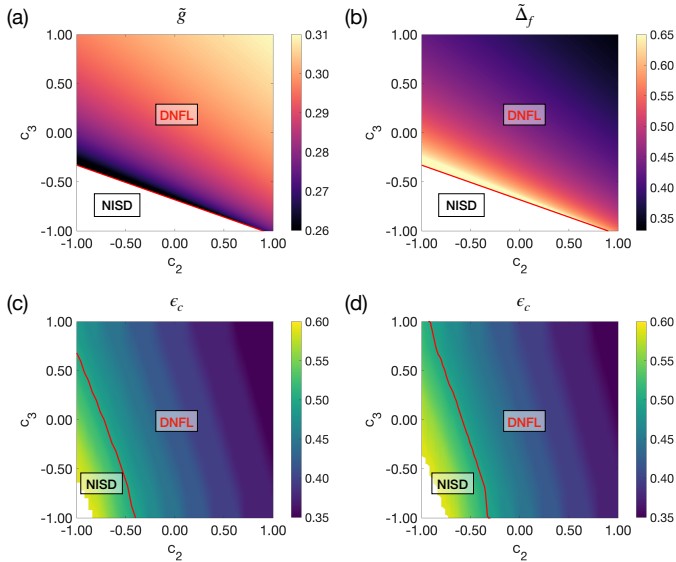

Figure 7: (a-b) Depiction of the values of $\tilde{g}$, $\tilde{\Delta}_f$ at the DNFL fixed point as a function of $(c_2, c_3)$, for $\epsilon = 0.5$ and $c_1 = 0.5$. Here, $(c_2, c_3)$ are the coefficients of the three-loop order corrections in the beta functions presented in Eq. (45). Each panel utilizes a color scale to represent the corresponding quantity, with the empty region denoting the absence of the DNFL fixed point, characterized instead by a runaway RG flow toward the NISD fixed point. The red line, marking $c_3 = -0.3526c_2 - 0.6826$, delineates the boundary separating these two regions. (c-d) The minimum threshold value $\epsilon_c$ to maintain the DNFL fixed point, plotted against $(c_2, c_3)$. $c_1 = 0$ and $c_1 = 1$ are utilized for (c) and (d), respectively. The color scale indicates the value of $\epsilon_c$ in each panel. The red line delineates the boundary where $\epsilon_c$ exceeds the physical value of $\epsilon = 0.5$, indicating the absence of the DNFL fixed point within the physical reality. The empty region denotes the absence of the DNFL fixed point for any value of $\epsilon$.

$(\tilde{B}_d \simeq 0.05)$ corresponds to diffusive Landau damping [3]. In higher-loop analysis, this modification might give rise to additional corrections not captured by loop expansions using the propagators presented in Eq. (17). To assess this effect, we expand $D_2(q)$ with respect to $\Pi_2(q)$ as $D_2(q) = D_1(q) \sum_{n=0}^{\infty} \left[ -D_1(q) g^2 \tilde{\Delta}_f \tilde{B}_d \frac{|q|^{2d-3}}{|q_y|^2} \right]^n$. The scaling analysis tells us that all higher-order terms in the expansion have the same superficial degree of divergence as the zeroth-order term. This indicates that loop corrections can still be regularized using dimensional regularization. For example, using $D_2(q)$, we find the first-order fermion self-energy correction from the Yukawa coupling as $\Sigma_1 = \tilde{g} \frac{-i \boldsymbol{P} \cdot \boldsymbol{\gamma}}{\epsilon} \sum_{n=0}^{\infty} a_n \left[ \frac{\tilde{\Delta}_f}{N^{1/2} \tilde{g}^{1/2}} \right]^n$, where $a_n$ is a numerical coefficient independent of the coupling constants. Note that the expansion parameter has a negative power of $N$. As a result, this correction can be dropped by taking the large $N$ limit, at least, in the low but intermediate temperature scale.

Based on the above observations, we posit that the beta functions for $\tilde{g}$ and $\tilde{\Delta}_f$ at the third order in the infinite $N$ limit are represented as follows:

$$\beta_{\tilde{g}}^{3\text{-loop}} = \frac{\beta_{\tilde{g}}^{2\text{-loop}}}{1 - \bar{z}^{2\text{-loop}} c_1 \tilde{g}^3} + \frac{\bar{z}^{2\text{-loop}}}{1 - \bar{z}^{2\text{-loop}} c_1 \tilde{g}^3} \left[ c_1 \tilde{g}^4 \right],$$

$$\beta_{\tilde{\Delta}_f}^{3\text{-loop}} = \frac{\beta_{\tilde{\Delta}_f}^{2\text{-loop}}}{1 - \bar{z}^{2\text{-loop}} c_1 \tilde{g}^3} + \frac{\bar{z}^{2\text{-loop}}}{1 - \bar{z}^{2\text{-loop}} c_1 \tilde{g}^3} \left[ 0.5 c_1 \tilde{\Delta}_f \tilde{g}^3 + c_2 \tilde{\Delta}_f^2 \tilde{g}^2 + c_3 \tilde{\Delta}_f^3 \tilde{g} \right]. \tag{45}$$

Here, $\beta_{\tilde{g}}^{\text{2-loop}}$ and $\beta_{\tilde{\Delta}_f}^{\text{2-loop}}$ ($\beta_{\tilde{g}}^{\text{3-loop}}$ and $\beta_{\tilde{\Delta}_f}^{\text{3-loop}}$) denote the two-loop (three-loop) beta functions, and $\bar{z}^{\text{2-loop}}$ denotes the dynamical exponent $\bar{z}$ found in the two-loop order. The two-loop beta functions and dynamical exponent are taken from Eqs. (21) and (32) with setting $\tilde{\Delta}_b = 0$. In principle, one can determine the coefficients $c_1$, $c_2$, and $c_3$ through three-loop-order calculations. Here, we investigate the stability of the DNFL fixed point within a weak-perturbation scenario: $|c_1|, |c_2|, |c_3| < 1$.

Figures 7(a-b) illustrate how $\tilde{g}^*$ and $\tilde{\Delta}_f^*$ vary as $(c_2, c_3)$ changes, based on the conditions $\beta_{\tilde{g}}^{\text{3-loop}} = \beta_{\tilde{\Delta}_f}^{\text{3-loop}} = 0$. Our findings reveal that when $c_2$ and $c_3$ are positive, $\tilde{\Delta}_f^*$ decreases as the magnitude of either $c_2$ or $c_3$ increases, likely due to the enhanced screening effect on $\tilde{\Delta}_f^*$. When this change of $\tilde{\Delta}_f^*$ is feed-backed to $\beta_{\tilde{g}}^{\text{2-loop}}$, $\tilde{g}^*$ displays an increasing trend due to the reduction in its screening term $0.17\tilde{\Delta}_f^*$. Conversely, when $c_2$ and $c_3$ are negative, the rise of their magnitude indicates the increase of $\tilde{\Delta}_f^*$ and subsequent reduction in $\tilde{g}^*$. Furthermore, when $|c_2|$ or $|c_3|$ becomes too large in the negative range, the DNFL fixed point is destabilized instead showing runaway flow to the NISD fixed point, as illustrated by white devoid regions (Fig. 7(a-b)).

Figures 7(c-d) illustrate how the DNFL fixed point becomes destabilized through changes in $\epsilon_c$, in response to variations in $(c_2, c_3)$. Our results show that positive values of $c_2$ and $c_3$ tend to decrease $\epsilon_c$, while negative values lead to an increasing trend in $\epsilon_c$. When $|c_2|$ or $|c_3|$ becomes too large in the negative range, $\epsilon_c$ can exceed the physical value, $\epsilon_{\text{ph}} = 0.5$. In this regime, achieving a balance between $\tilde{\Delta}_f^*$ and $\tilde{g}^*$, as detailed in Sec. 3.3, becomes unattainable for any $\epsilon \le \epsilon_{\text{ph}}$. Consequently, the DNFL fixed point disappears, indicating a runaway flow toward the NISD fixed point. It's worth mentioning that additional screening terms, such as a positive $c_1$ in $\beta_{\tilde{\Delta}_f}^{\text{3-loop}}$, can broaden the stability range for the DNFL fixed point, evident from comparing Figures 7(c) and (d) obtained from different values of $c_1$.

We anticipate that quartic or higher-order corrections in $\beta_{\tilde{\Delta}_f}$ could have a similar impact. Their screening or antiscreening nature suggests a corresponding increase or decrease in $\epsilon_c$, particularly when the magnitudes of these corrections are not excessively large. The DNFL fixed point is likely to remain stable if the overall effect of these corrections is screening. However, if their effect is antiscreening, it must be weak enough to keep $\epsilon_c \le \epsilon_{\text{ph}}$. The stability of the DNFL fixed point seems plausible given the relatively low fixed point values of the coupling parameters—specifically, $\tilde{g}^* \approx 0.3$ and $\tilde{\Delta}_f^* \approx 0.4$ for $\epsilon = 0.5$. If these coupling values ensure that higher-order corrections are smaller than the two-loop screening term, the DNFL fixed point may withstand these additional antiscreening perturbations.

### 3.6.2 Interpatch disorder scattering

Up to now, our focus has centered on the two-patch model, which incorporates two antipodal segments among the entire Fermi surface, as depicted in Fig. 1. While this minimal model naturally extends the previous clean model [6] to account for random potential disorder, broadening our approach to encompass the entire Fermi surface becomes crucial for understanding physical phenomena involving its entirety, such as Cooper pairing [72]. Hence, we now turn our attention to an extended multi-patch model [72], characterized by a Lagrangian density given by:

$$\mathcal{L} = \sum_{\alpha=1}^{N_p} \mathcal{L}_\alpha + \sum_{\alpha,\beta=1(\alpha<\beta)}^{N_p} \mathcal{L}_{\alpha\beta}^{\text{dis}}. \tag{46}$$

Here, the indices $\alpha, \beta = 1, \cdots, N_p$ denote $N_p$ pairs of antipodal patches across the Fermi surface [Fig. 7]. The first term in Eq. (46) represents the original Lagrangian, as presented in Eq. (1),

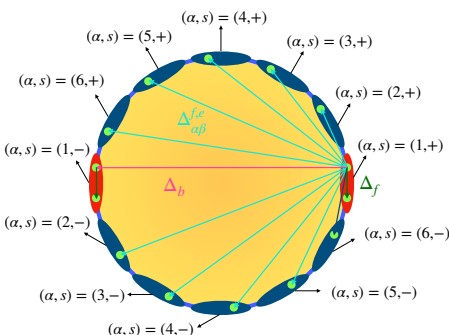

Figure 8: Schematic illustration of an extended multiple-patch model with interpatch disorder scattering. This model comprises $N_p$ pairs of two antipodal patches (*e.g.*, $N_p = 6$ in this illustration), including the original patches (depicted in red) and additional segments stemming from the entire Fermi surface (depicted in blue). The green and magenta arrows represent the original forward ($\Delta_f$) and backward disorder scattering ($\Delta_b$), respectively, while the cyan arrows depict interpatch disorder scattering ($\Delta_{\alpha\beta}^{f,e}$).

which is replicated for each pair of patches denoted by $\alpha$. The second term represents a newly introduced "interpatch disorder scattering" term that mixes fermions from different patches ($\alpha \neq \beta$):

$$
\begin{aligned}
\mathcal{L}_{\alpha\beta}^{\text{dis}} = &-\sum_{a,b=0}^{R}\sum_{j=1}^{N}\sum_{s,s'=\pm}\frac{\Delta_{\alpha\beta}^{f}}{2N}\psi_{\alpha,s,j}^{a,\dagger}(k+q)\psi_{\alpha,s,j}^{a}(k)\psi_{\beta,s',j}^{b,\dagger}(k'-q)\psi_{\beta,s',j}^{b}(k') \\
&-\sum_{a,b=0}^{R}\sum_{j=1}^{N}\sum_{s=\pm}\frac{\Delta_{\alpha\beta}^{e}}{2N}\psi_{\alpha,s,j}^{a,\dagger}(k+q)\psi_{\beta,s,j}^{a}(k)\psi_{\beta,\bar{s},j}^{b,\dagger}(k'-q)\psi_{\alpha,\bar{s},j}^{b}(k') \\
&-\sum_{a,b=0}^{R}\sum_{j=1}^{N}\sum_{s=\pm}\frac{\Delta_{\alpha\beta}^{e}}{2N}\psi_{\alpha,\bar{s},j}^{a,\dagger}(k+q)\psi_{\beta,s,j}^{a}(k)\psi_{\beta,s,j}^{b,\dagger}(k'-q)\psi_{\alpha,\bar{s},j}^{b}(k').
\end{aligned}
\tag{47}
$$

Here, $\Delta_{\alpha\beta}^{f}$ shift fermions within their respective patch, while $\Delta_{\alpha\beta}^{e}$ transfer fermions from one to another. These two terms conserve the sum of Fermi momenta of fermions, making their influence more significant compared to other nonconserving terms for low energy fermions near the Fermi surface [27].

We investigate the stability of the DNFL fixed point concerning the introduction of interpatch disorder scattering. Utilizing Eq. (15), we formally express the beta function for $\Delta_{\alpha\beta}^{f,e}$ as follows:

$$
\beta_{\Delta_{\alpha\beta}^{f,e}} = \Delta_{\alpha\beta}^{f,e}\left[-\epsilon\bar{z} + \frac{1}{2}(\bar{z}-1) + 4\gamma_{\Psi} - \gamma_{\Delta_{\alpha\beta}^{f,e}}\right],
\tag{48}
$$

where the critical exponents $\bar{z}$, $\gamma_{\Psi}$, and $\gamma_{\Delta_{\alpha\beta}^{f,e}}$ are defined in Eq. (16). Evaluating these exponents generally poses challenges as relevant Feynman diagrams intricately involve fermion propagators from different patches, not representable in global momentum coordinates. However, one-loop self-energy diagrams are manageable despite the challenge, as all propagators are confined within a single patch. Here, we calculate $\bar{z}$ and $\gamma_{\Psi}$ considering these one-loop self-energy diagrams. The contributions from $\tilde{g}$, $\tilde{\Delta}_f$, and $\tilde{\Delta}_b$ are provided in Eq. (18). The relevant diagrams for $\Delta_{\alpha\beta}^{f,e}$ resemble Fig. 3(c), leading to the following evaluations up to one-loop

order in the $N \to \infty$ limit:

$$\bar{z} = \left[ 1 - 0.67\tilde{g} + 0.5\tilde{\Delta}_f + 0.5\tilde{\Delta}_b + 0.5\tilde{\Delta} + \mathcal{O}\left(\tilde{g}^2, \tilde{\Delta}^2\right) \right]^{-1},$$
$$\gamma_{\Psi} = 0.25\bar{z}\left(\tilde{\Delta}_f + \tilde{\Delta}_b + \tilde{\Delta}_{\alpha\beta}\right) + \mathcal{O}\left(\tilde{g}^2, \tilde{\Delta}^2\right), \tag{49}$$

where we denote $\tilde{\Delta} = \frac{1}{2\pi^{5/4}\Gamma(\frac{1}{4})} \sum_{\beta \neq \alpha} \left(\Delta_{\alpha\beta}^f + \Delta_{\alpha\beta}^e\right)$. Utilizing this result, we derive $\beta_{\Delta_{\alpha\beta}^{f,e}}$ as:

$$\beta_{\Delta_{\alpha\beta}^{f,e}} \simeq F\Delta_{\alpha\beta}^{f,e} + 0.75\tilde{\Delta}_{\alpha\beta}\Delta_{\alpha\beta}^{f,e} - \gamma_{\Delta_{\alpha\beta}^{f,e}}\Delta_{\alpha\beta}^{f,e}, \tag{50}$$

where $F$, the coefficient of the linear term, is given by $F = \bar{z}(-\epsilon + 0.33\tilde{g} + 0.75\tilde{\Delta}_f + 0.75\tilde{\Delta}_b)$. At the CNFL fixed point, $F$ remains large (*e.g.*, $F \approx -0.33$ for $\epsilon = 0.40$ and $F \approx -0.43$ for $\epsilon = 0.5$), indicating the strong coupling nature of $\Delta_{\alpha\beta}^{f,e}$. However, at the DNFL fixed point, its sign reverses or weakens significantly (*e.g.*, $F \approx 0.12$ for $\epsilon = 0.40$ or $F \approx -0.021$ for $\epsilon = 0.5$), attributed to a large contribution from the anomalous dimension of fermion fields ($4\gamma_{\Psi} \sim \tilde{\Delta}_f \approx 0.46$). This suggests the potential irrelevance of $\Delta_{\alpha\beta}^{f,e}$ at the DNFL fixed point, maintaining the stability of the DNFL fixed point against their introduction. Nonetheless, as $\gamma_{\Delta_{\alpha\beta}^{f,e}}$ includes additional linear-order contributions concerning $\tilde{g}$ and $\tilde{\Delta}_f$, the analysis remains inconclusive. Identifying the relevance of the interpatch disorder scattering, necessitating the evaluation of $\gamma_{\Delta_{\alpha\beta}^{f,e}}$ and potentially higher-order RG analysis, represents a significant future research direction.

## 4 Discussion

### 4.1 Alternative cutoff regularization scheme

One may employ the following alternative cutoff regularization:

$$\Sigma_1(p) = \frac{i\Delta_f}{N} \int \frac{d^{d-2}\boldsymbol{k}_\perp}{(2\pi)^{d-2}} \int_{-\infty}^{\infty} \frac{dk_x}{2\pi} \int_{-\Lambda}^{\Lambda} \frac{dk_y}{2\pi} \frac{-p_0\gamma_0 + (k_x + p_x + k_y^2)\gamma_{d-1}}{p_0^2 + |\boldsymbol{k}_\perp|^2 + (k_x + p_x + k_y^2)^2}. \tag{51}$$

In this scenario, we derive $B = \frac{\tilde{\Delta}_f}{N} \frac{2\Lambda\Gamma(\frac{3-d}{2})}{(4\pi)^{\frac{d-2}{2}}[p_0^2]^{\frac{3-d}{2}}}$ and $A = C = 0$. The upper critical dimension for the disorder scattering is now $d_c = 3$, distinct from $d_c = 5/2$ for the Yukawa coupling. Consequently, to determine the $\epsilon$-pole of $B$, we need to introduce a double epsilon expansion scheme [38], where $d = 3 - \epsilon - \epsilon_\tau$ and both $\epsilon$ and $\epsilon_\tau$ are regarded as small expansion parameters. Using this scheme, we find $B = \frac{\tilde{\Delta}_f}{N(\epsilon+\epsilon_\tau)} \frac{2\Lambda}{\sqrt{\pi}} + \mathcal{O}(1)$. The $\epsilon$-pole now depends on the UV-cutoff $\Lambda$, resulting in UV-IR mixing as observed in the prior studies of the spin-density-wave QCP problem [9, 10]. Therefore, our original regularization scheme offers a more robust framework for capturing the scaling behavior of the system compared to this alternative scheme without UV-IR mixing, remaining insensitive to cutoff dependencies or microscopic details.

### 4.2 Random mass disorder for critical bosons

In this study, our focus has been on the inclusion of random potential terms for fermions. However, it is important to recognize that random terms for bosons could also be significant

and warrant consideration [73]. To shed light on how such terms can be incorporated into our framework, we examine the following random $T_c$ disorder [38]:

$$-\frac{\Gamma}{2}\sum_{a,b=1}^{R}\int\frac{dk_0 d^2\boldsymbol{k}\,dk_0' d^2\boldsymbol{k}'d^2\boldsymbol{q}}{(2\pi)^8}\Phi^a(k+q)\Phi^a(k)\Phi^b(k'-q)\Phi^b(k').$$

This term leads to a first-order boson self-energy correction of the form:

$$\Gamma\int\frac{d^{d-2}\boldsymbol{q}_\perp dq_x dq_y}{(2\pi)^d}\frac{1}{q_y^2+g^2 B_d|\boldsymbol{q}|^{d-1}/|q_y|}\sim\Gamma\left(\frac{5-2d}{3}\right)\int\frac{dq_x}{2\pi}.$$

Notably, the integral for $q_x$ cannot be regularized since the remaining integral is independent of $q_x$. One possible solution is to reintroduce the omitted $q_x^2$ term in the boson propagator. In this scenario, the self-energy becomes $\Gamma\int\frac{d^{d-2}\boldsymbol{q}_\perp dq_x dq_y}{(2\pi)^d}\frac{1}{q_x^2+q_y^2+g^2 B_d|\boldsymbol{q}|^{d-1}/|q_y|}\sim\Gamma(\frac{4-d}{3})$, indicating that this correction is UV-finite near $d\approx d_c=5/2$. However, retaining the $q_x^2$ term might potentially interfere with the anomalous scaling law described in Eq. (7) at the Ising-nematic QCP. In consideration of this possibility, we tentatively conclude that the influence of random terms on bosons may not be thoroughly investigated within the limitations of our dimensional regularization scheme.

## 4.3 Extension to other quantum phase transitions

Our RG framework is potentially applicable to other metallic quantum critical systems, characterized by an order parameter with zero center-of-mass momentum and critical fluctuations coupled to a finite density of fermions via a Yukawa coupling. In these systems, the two-patch model description, combined with a parabolic dispersion, is appropriate, and the dimensional regularization presented in Eqs. (7) and (8) remains valid. Notable examples include itinerant ferromagnetic quantum phase transitions [3], U(1) spin liquids [74–78], and the half-filled Landau level [79–83].

As an illustration, consider the case of the U(1) spin liquid [74–77]. In this scenario, the Yukawa coupling term in the action of Eq. (6) requires modification:

$$\frac{ig}{\sqrt{N}}\Phi(q)\bar{\Psi}_j^a(k+q)\gamma_{d-1}\Psi_j^a(k)\to\frac{ig}{\sqrt{N}}\Phi(q)\bar{\Psi}_j^a(k+q)\gamma_0\Psi_j^a(k),$$

while the other components remain unchanged [7]. The transition from $\gamma_{d-1}$ to $\gamma_0$ in the vertex alters the sign of the primary screening term $3.5\tilde{g}\tilde{\Delta}_f$ in $\beta_{\tilde{\Delta}_f}$. Notably, the sign alteration invalidates the screening of $\tilde{\Delta}_f$ through this term. Consequently, we speculate that the RG flow may exhibit a runaway flow to the strong disorder regime, and a DNFL fixed point might not manifest in this case, at least within the scope of the two-loop order.

# 5 Conclusion

We have investigated the impact of random potential disorder for fermions on the scaling behavior of the two-patch model for two-dimensional Ising-nematic quantum critical points. Employing a controllable renormalization group theory, we systematically incorporate quantum corrections stemming from the random potential and the Yukawa coupling between electrons and bosonic order-parameter fluctuations through a perturbative epsilon expansion. Extending our analysis beyond the conventional one-loop level to the two-loop order, we have

unveiled a stable disordered non-Fermi liquid fixed point for the two-patch model and computed critical exponents up to the two-loop order. Our investigation sheds light on the scaling characteristics of two-dimensional metallic quantum critical points in the presence of random potential disorder. Furthermore, our findings highlight the essential role of higher-order loop corrections in elucidating the intricate interplay between quantum criticality and quenched randomness in two dimensions.

# Acknowledgments

We would like to thank Iksu Jang, Jaeho Han, Jinho Yang, Chushun Tian, and Sung Sik Lee for sharing their insights.

**Funding information** K.-M.K. was supported by the Institute for Basic Science in the Republic of Korea through the project IBS-R024-D1. K.-S.K. was supported by the Ministry of Education, Science, and Technology (RS-2024-00337134, NRF-2021R1A2C1006453, and NRF-2021R1A4A3029839) of the National Research Foundation of Korea (NRF) and by TJ Park Science Fellowship of the POSCO TJ Park Foundation.

**Note added** After completing our investigations, we received correspondence from S.-S. Lee indicating that a power-law self-energy correction due to disorder scattering might be more significant than the logarithmic corrections considered in our analysis. We found that this correction does indeed appear in our dimensional regularization scheme. If this correction plays a substantial role, the dynamics of disordered fermions at lower energy levels might differ from what we described, indicating that our results are more applicable to intermediate temperature ranges. To gain a more accurate understanding of the low-energy behavior, it will be essential to conduct additional RG analyses that include these power-law corrections and then compare those results with our current findings. This future research could offer deeper insights into the nature of two-dimensional disordered nematic quantum criticality.

# A One-loop self-energy corrections

Table 2: Feynman diagrams for one-loop self-energy corrections. Here, $A_0$, $A_1$, and $A_2$ represent the coefficient of the $\epsilon$ poles computed from the corresponding Feynman diagrams (see Eq. (E.10) for the definitions). $\Pi_1(q)$ represents the Landau damping term for the dressed boson propagator.

| Diagram No. | BS1-1 | FS1-1 | FS2-2 | FS1-3 |
|---|---|---|---|---|
| Feynman Diagram | | | | |
| Renormalization factors | $\Pi_1(q) = -g^2 B_d \frac{\|\mathbf{Q}\|^{d-1}}{\|q_y\|}$ | $A_0 = -\tilde{g},$ $A_1 = -\tilde{g}$ | $A_0 = -\tilde{\Delta}_f,$ $A_2 = -\frac{1}{2}\tilde{\Delta}_f$ | $A_0 = -\tilde{\Delta}_b,$ $A_2 = -\frac{1}{2}\tilde{\Delta}_b$ |

## A.1 Boson self-energy

### A.1.1 Feynman diagram BS1-1

The boson self-energy correction in Table 2 BS1-1 is given by

$$\Pi_1(q) = -g^2 \int \frac{d^{d+1}k}{(2\pi)^{d+1}} \text{tr}\big[\gamma_{d-1} G_0(k+q)\gamma_{d-1} G_0(k)\big] = 2g^2 \int \frac{d^{d+1}k}{(2\pi)^{d+1}} \frac{\delta_{\mathbf{k+q}}\delta_{\mathbf{k}} - (\mathbf{K+Q})\cdot\mathbf{K}}{\big[\delta_{\mathbf{k+q}}^2 + (\mathbf{K+Q})^2\big]\big[\delta_{\mathbf{k}}^2 + \mathbf{K}^2\big]}.$$

Integrating over $k_x$ and $k_y$, we obtain $\Pi_1(q)$ as

$$\Pi_1(q) = g^2 \int \frac{d\mathbf{K}dk_y}{(2\pi)^d} \frac{(|\mathbf{K+Q}| + |\mathbf{K}|)\big(1 - \frac{(\mathbf{K+Q})\cdot\mathbf{K}}{|\mathbf{K+Q}||\mathbf{K}|}\big)}{(2k_y q_y)^2 + (|\mathbf{K+Q}| + |\mathbf{K}|)^2} = \frac{g^2}{4|q_y|} \int \frac{d\mathbf{K}}{(2\pi)^{d-1}} \left(1 - \frac{(\mathbf{K+Q})\cdot\mathbf{K}}{|\mathbf{K+Q}||\mathbf{K}|}\right).$$

Using the Feynman parametrization method [69], we obtain

$$\Pi_1(q) = \frac{g^2}{4\pi|q_y|} \int_0^1 dx \int \frac{d\mathbf{K}}{(2\pi)^{d-1}} \frac{-2[x(1-x)]^{\frac{1}{2}}\tilde{\mathbf{K}}^2}{\tilde{\mathbf{K}}^2 + x(1-x)\mathbf{Q}^2},$$

where $\tilde{\mathbf{K}} = \mathbf{K} + x\mathbf{Q}$. Integrating over $\mathbf{K}$, we obtain

$$\Pi_1(q) = -\frac{g^2|\mathbf{Q}|^{d-1}\Gamma(\frac{3-d}{2})}{2\pi|q_y|(4\pi)^{\frac{d-1}{2}}} \int_0^1 dx[x(1-x)]^{\frac{d-2}{2}}.$$

Integrating over $x$, we finally obtain

$$\Pi_1(q) = -g^2 B_d \frac{|\mathbf{Q}|^{d-1}}{|q_y|}, \qquad B_d = \frac{\Gamma(\frac{3-d}{2})\Gamma(\frac{d}{2})^2}{2\pi(4\pi)^{(d-1)/2}\Gamma(d)}.$$

## A.2 Fermion self-energy

### A.2.1 Feynman diagram FS1-1

The fermion self-energy correction in Table 2 FS1-1 is given by

$$\Sigma(1) = -\frac{g^2}{N} \int \frac{d^{d+1}k}{(2\pi)^{d+1}} \gamma_{d-1} G_0(p+k)\gamma_{d-1} D_1(k) = \frac{ig^2}{N} \int \frac{d^{d+1}k}{(2\pi)^{d+1}} \frac{-(\mathbf{P+K})\cdot\mathbf{\Gamma} + \delta_{\mathbf{p+k}}\gamma_{d-1}}{\delta_{\mathbf{p+k}}^2 + (\mathbf{P+K})^2} D_1(k).$$

Integrating over $k_x$ and $k_y$, we obtain $\Sigma(1)$ as

$$\Sigma(1) = \frac{ig^2}{2N} \int \frac{d\mathbf{K}dk_y}{(2\pi)^d} \frac{-(\mathbf{P+K})\cdot\mathbf{\Gamma}}{|\mathbf{P+K}|\big[k_y^2 + g^2 B_d \frac{|\mathbf{K}|^{d-1}}{|k_y|}\big]} = \frac{ig^2}{3\sqrt{3}N} \int \frac{d\mathbf{K}}{(2\pi)^{d-1}} \frac{-(\mathbf{P+K})\cdot\mathbf{\Gamma}}{|\mathbf{P+K}|\big[g^2 B_d|\mathbf{K}|^{d-1}\big]^{1/3}}.$$

Using the Feynman parametrization method, we obtain

$$\Sigma(1) = \frac{ig^{4/3}}{3\sqrt{3}B_d^{1/3}N} \int_0^1 dx \frac{x^{-\frac{1}{2}}(1-x)^{\frac{d-7}{6}}\Gamma(\frac{d+2}{6})}{\Gamma(\frac{1}{2})\Gamma(\frac{d-1}{6})} \int \frac{d\mathbf{K}}{(2\pi)^{d-1}} \frac{-(1-x)(\mathbf{P}\cdot\mathbf{\Gamma})}{\big[(\mathbf{K}+x\mathbf{P}) + x(1-x)\mathbf{P}^2\big]^{\frac{d+2}{6}}}.$$

Integrating over $\mathbf{K}$ and $x$, we obtain

$$\Sigma(1) = -\frac{ig^{4/3}\Gamma(\frac{5-2d}{6})(\mathbf{P}\cdot\mathbf{\Gamma})}{3\sqrt{3}B_d^{1/3}N|\mathbf{P}|^{\frac{5-2d}{3}}} \int_0^1 dx \frac{x^{\frac{d-4}{3}}(1-x)^{\frac{d-2}{2}}}{(4\pi)^{\frac{d-1}{2}}\Gamma(\frac{1}{2})\Gamma(\frac{d-1}{6})} = -\frac{iS'g^{4/3}}{6\sqrt{3}B^{1/3}N} \frac{\mathbf{P}\cdot\mathbf{\Gamma}}{\epsilon} + \mathcal{O}(1),$$

where $S' = \frac{2}{(4\pi)^{3/4}\Gamma(3/4)}$ and $B = \lim_{d\to 5/2} B_d$. Defining $\tilde{g} = \frac{S'g^{4/3}}{6\sqrt{3}B^{1/3}N}$, we finally obtain

$$\Sigma(1) = -\frac{\tilde{g}}{\epsilon}(i\mathbf{P}\cdot\mathbf{\Gamma}).$$

### A.2.2 Feynman diagram FS1-2

The fermion self-energy correction in Table 2 FS1-2 is given by

$$\Sigma(2) = -\Delta_f \int \frac{d^{d+1}k}{(2\pi)^d} \delta(k_0)\gamma_{d-1} G_0(p+k)\gamma_{d-1} = i\Delta_f \int \frac{d^d k}{(2\pi)^d} \frac{-p_0\gamma_0 - (\mathbf{p}_\perp + \mathbf{k}_\perp)\cdot\gamma_\perp + \delta_{\mathbf{p+k}}\gamma_{d-1}}{p_0^2 + (\mathbf{p}_\perp + \mathbf{k}_\perp)^2 + \delta_{\mathbf{p+k}}^2},$$

where $d^d k \equiv d\mathbf{k}_\perp dk_x dk_y$. To find renormalization factors, we expand $\Sigma(2)$ for $p$ as

$$\Sigma(2) = i\Sigma_0 + \Sigma_a(ip_0\gamma_0) + \Sigma_b(i\mathbf{p}_\perp \cdot \gamma_\perp) + \Sigma_c(i\delta_{\mathbf{p}}\gamma_{d-1}) + \mathcal{O}(p^2),$$

where $\Sigma_0$, $\Sigma_a$, $\Sigma_b$, and $\Sigma_c$ are, respectively, given by

$$\Sigma_0 = \Delta_f \int \frac{d^d k}{(2\pi)^d} \frac{-\mathbf{k}_\perp \cdot \gamma_\perp + \delta_{\mathbf{k}}\gamma_{d-1}}{\mathbf{k}_\perp^2 + \delta_{\mathbf{k}}^2 + p_0^2},$$

$$\Sigma_a = -\Delta_f \int \frac{d^d k}{(2\pi)^d} \frac{1}{\mathbf{k}_\perp^2 + \delta_{\mathbf{k}}^2 + p_0^2},$$

$$\Sigma_b = -\Delta_f \int \frac{d^d k}{(2\pi)^d} \frac{-2k_{\perp,i}^2 + \mathbf{k}_\perp^2 + \delta_{\mathbf{k}}^2 + p_0^2}{\left[\mathbf{k}_\perp^2 + \delta_{\mathbf{k}}^2 + p_0^2\right]^2},$$

$$\Sigma_c = -\Delta_f \int \frac{d^d k}{(2\pi)^d} \frac{-\mathbf{k}_\perp^2 + \delta_{\mathbf{k}}^2 - p_0^2}{\left[\mathbf{k}_\perp^2 + \delta_{\mathbf{k}}^2 + p_0^2\right]^2}.$$

Integrating over $\mathbf{k}_\perp$, we obtain

$$\Sigma_0 = \Delta_f \int \frac{dk_x dk_y}{(2\pi)^2} \frac{\delta_{\mathbf{k}}\gamma_{d-1}\Gamma(2-\frac{d}{2})}{(4\pi)^{\frac{d-2}{2}}\left[\delta_{\mathbf{k}} + p_0^2\right]^{2-\frac{d}{2}}},$$

$$\Sigma_a = -\Delta_f \int \frac{dk_x dk_y}{(2\pi)^2} \frac{\Gamma(2-\frac{d}{2})}{(4\pi)^{\frac{d-2}{2}}\left[\delta_{\mathbf{k}}^2 + p_0^2\right]^{2-\frac{d}{2}}},$$

$$\Sigma_c = -\Delta_f \int \frac{dk_x dk_y}{(2\pi)^2} \frac{\left(\frac{6-2d}{4-d}\delta_{\mathbf{k}}^2 - \frac{2}{4-d}p_0^2\right)\Gamma(3-\frac{d}{2})}{(4\pi)^{\frac{d-2}{2}}\left[\delta_{\mathbf{k}}^2 + p_0^2\right]^{3-\frac{d}{2}}},$$

where $\Sigma_b$ vanishes.

It turns out that these integrals diverge when integrated over $k_x, k_y \in (-\infty, \infty)$. For example, $\Sigma_a$ is calculated as

$$\int_{-\infty}^{\infty} \frac{dk_x dk_y}{(2\pi)^2} \frac{\Gamma(2-\frac{d}{2})}{\left[\delta_{\mathbf{k}}^2 + p_0^2\right]^{2-\frac{d}{2}}} = \int_{-\infty}^{\infty} \frac{dk_y}{2\pi} \frac{\Gamma(\frac{3-d}{2})}{(4\pi)^{\frac{1}{2}}|p_0|^{3-d}},$$

which integral trivially diverges since the integrand is independent of $k_y$. Thus, the dimensional regularization fails in this case. Integrating over $k_y$ first does not help, either. The problem here is that there are infinitely many points of $(k_x, k_y)$ in the integral region for the contour $\delta_{\mathbf{k}} = c$, where $c$ is a constant including $c = 0$.

Note that this is an artifact of the patch theory. If the whole Fermi surface had been taken into account, such divergence would have not arisen. In this respect, we regularize the integral for $\Sigma_a$ by introducing a cutoff scale as $k_x \in (-k_f, \infty)$. Then, $\Sigma_a$ becomes

$$\int_{-k_f}^{\infty} \frac{dk_x}{2\pi} \int_{-\infty}^{\infty} \frac{dk_y}{2\pi} \frac{\Gamma(2-\frac{d}{2})}{\left[\delta_{\mathbf{k}}^2 + p_0^2\right]^{2-\frac{d}{2}}} = \frac{\sqrt{\pi}\Gamma(2-\frac{d}{2})\Gamma(\frac{5}{2}-d)\phi_d\left(\frac{|p_0|}{|k_f|}\right)}{4\pi^2\Gamma(4-d)(-k_f)^\epsilon},$$

where $\phi_d(x) \equiv {}_2F_1\left(\frac{5-2d}{4}, \frac{7-2d}{4}, \frac{5-d}{2}, -x^2\right)$ is a hypergeometric function of $x$. Expanding this expression with $\epsilon$, we find an $\epsilon$ pole as $\frac{\sqrt{2}}{2\pi\Gamma(\frac{1}{4})\epsilon} - \frac{\sqrt{2}}{2\pi\Gamma(\frac{1}{4})}\ln(-|p_0|/|k_f|) + \cdots$. The finite part still diverges in the limit of $k_f \to \infty$ but an $\epsilon$ pole can be extracted out regardless of $k_f$.

Then, the problem is whether we can find singular corrections corresponding to $\epsilon$ poles regardless of $k_f$ or not in general. For this matter, we consider a general expression for the integral of $\int \frac{dk_x dk_y}{(2\pi)^2} f(\delta_{\mathbf{k}})$, where the integrand depends on $\mathbf{k}$ only with $\delta_{\mathbf{k}}$. Otherwise, there would be no divergence associated with $\mathbf{k}$. Converting the momentum integral into an energy integral, we obtain $\int_{-k_f}^{\infty} d\xi\, \nu(\xi; k_f) f(\xi)$, where the density of states is

$$\nu(\xi; k_f) = \int_{-k_f}^{\infty} \frac{dk_x}{2\pi} \int_{-\infty}^{\infty} \frac{dk_y}{2\pi} \delta(\xi - \delta_{\mathbf{k}}) = \frac{1}{2\pi^2}\sqrt{\xi + k_f}\,.$$

We split the integral into three parts as follows

$$\int_{-k_f}^{\infty} d\xi\, \nu(\xi; k_f) f(\xi) = \int_0^{\infty} d\xi\, \nu(\xi; k_f = 0) f(\xi) + \int_0^{\infty} d\xi[\nu(\xi; k_f) - \nu(\xi; k_f = 0)] f(\xi) + \int_{-k_f}^{0} d\xi\, \nu(\xi; k_f) f(\xi)$$

$$= \frac{1}{2\pi^2} \int_0^{\infty} d\xi\, \sqrt{\xi} f(\xi) + \frac{k_f}{2\pi^2} \int_0^{\infty} d\xi\, \frac{f(\xi)}{\sqrt{\xi + k_f} + \sqrt{\xi}} + \frac{1}{2\pi^2} \int_{-k_f}^{0} d\xi\, \sqrt{\xi + k_f} f(\xi). \quad \text{(A.1)}$$

Power counting tells that only the first term is singular if $f(\xi)$ has an $\xi$-power lower than $-1/2$. In fact, most of loop corrections except for $\Sigma_0$ satisfy this condition because we are performing the renormalization group analysis around the upper critical dimension. For example, we consider $\Sigma_a$ where we have $f(\xi) \sim \xi^{-\frac{3}{2}-\epsilon}$. In this case, the first term, $\int^{\infty} d\xi\, \xi^{-1-\epsilon}$, is singular in the $\epsilon \to 0$ limit while the second term, $\int^{\infty} d\xi\, \xi^{-2-\epsilon}$, is not. As a result, we may find an $\epsilon$ pole by writing the integral as

$$\int_{-k_f}^{\infty} \frac{dk_x}{2\pi} \int_{-\infty}^{\infty} \frac{dk_y}{2\pi} f(\delta_{\mathbf{k}}) = \int_0^{\infty} \frac{d\xi}{2\pi^2} \sqrt{\xi} f(\xi) + \mathcal{O}(1), \quad \text{(A.2)}$$

where the finite part of $\mathcal{O}(1)$ depends on $k_f$ and may diverge in the limit of $k_f \to \infty$.

Using Eq. (A.2), we find

$$\Sigma_0 = \Delta_f \int_{-k_f}^{\infty} \frac{d\xi}{2\pi^2} \sqrt{\xi + k_f}\, \frac{\xi \gamma_{d-1} \Gamma(2-\frac{d}{2})}{(4\pi)^{\frac{d-2}{2}}[\xi^2 + p_0^2]^{2-\frac{d}{2}}}\,,$$

$$\Sigma_a = -\Delta_f \int_0^{\infty} \frac{d\xi}{2\pi^2} \sqrt{\xi}\, \frac{\Gamma(2-\frac{d}{2})}{(4\pi)^{\frac{d-2}{2}}[\xi^2 + p_0^2]^{2-\frac{d}{2}}}\,,$$

$$\Sigma_c = -\Delta_f \int_0^{\infty} \frac{d\xi}{2\pi^2} \sqrt{\xi}\, \frac{\left(\frac{6-2d}{4-d}\xi^2 - \frac{2}{4-d}p_0^2\right)\Gamma(3-\frac{d}{2})}{(4\pi)^{\frac{d-2}{2}}[\xi^2 + p_0^2]^{3-\frac{d}{2}}}\,,$$

where we have not used Eq. (A.2) for $\Sigma_0$ because it gets a singular correction from not only the first term but also the second term in Eq. (A.1). Integrating over $\epsilon$, we have

$$\Sigma_0 = \Delta_f \frac{(k_f \gamma_{d-1})\phi_d'\left(\frac{|p_0|}{|k_f|}\right)}{\pi(4\pi)^{\frac{d}{2}}(-k_f)^{\frac{5-2d}{2}}} = \frac{\Delta_f}{\epsilon} \frac{S\sqrt{2}}{8}(k_f \gamma_{d-1}) + \mathcal{O}(1),$$

$$\Sigma_a = -\Delta_f \frac{\Gamma(\frac{3}{4})\Gamma(\frac{5-2d}{4})}{\pi(4\pi)^{\frac{d}{2}}|p_0|^{\frac{5-2d}{2}}} = -\frac{\Delta_f}{\epsilon} \frac{S\sqrt{2}}{4} + \mathcal{O}(1),$$

$$\Sigma_c = -\Delta_f \frac{\Gamma(\frac{3}{4})\Gamma(\frac{5-2d}{4})}{2\pi(4\pi)^{\frac{d}{2}}|p_0|^{\frac{5-2d}{2}}} = -\frac{\Delta_f}{\epsilon} \frac{S\sqrt{2}}{8} + \mathcal{O}(1),$$

where $\phi'_d(x) = \frac{\Gamma(\frac{1}{2})\Gamma(\frac{3}{2}-d)}{\Gamma(3-d)}{}_2F_1\left(\frac{3-2d}{4},\frac{5-2d}{4},\frac{3-d}{2},-x^2\right)$ and $S = \frac{2}{(4\pi)^{5/4}\Gamma(5/4)}$. Defining $\tilde{\Delta}_f = \frac{\sqrt{2}S\Delta_f}{4}$, we finally obtain

$$\Sigma(2) = -\frac{\tilde{\Delta}_f}{\epsilon}(ip_0\gamma_0) - \frac{\tilde{\Delta}_f}{2\epsilon}(i\delta_{\mathbf{p}}\gamma_{d-1}) + \frac{\tilde{\Delta}_f}{2\epsilon}(ik_f\gamma_{d-1}).$$

### A.2.3 Feynman diagram FS1-3

The fermion self-energy correction in Table 2 FS1-3 is given by

$$\Sigma(3) = -\Delta_b \int \frac{d^{d+1}k}{(2\pi)^d}\delta(k_0)G_0^*(k-p) = i\Delta_b \int \frac{d^d k}{(2\pi)^d}\frac{-p_0\gamma_0 - (\mathbf{k}_\perp - \mathbf{p}_\perp)\cdot\gamma_\perp - \delta_{\mathbf{k}-\mathbf{p}}\gamma_{d-1}}{(\mathbf{k}_\perp - \mathbf{p}_\perp)^2 + \delta^2_{\mathbf{k}-\mathbf{p}} + p_0^2}.$$

To find renormalization factors, we expand $\Sigma(3)$ for $p$ as

$$\Sigma(3) = i\Sigma_0 + \Sigma_a(ip_0\gamma_0) + \Sigma_b(i\mathbf{p}_\perp\cdot\gamma_\perp) + \Sigma_c(i\delta_{\mathbf{p}}\gamma_{d-1}) + \mathcal{O}(p^2),$$

where $\Sigma_0$, $\Sigma_a$, $\Sigma_b$, and $\Sigma_c$ are, respectively, given by

$$\Sigma_0 = \Delta_b \int \frac{d^d k}{(2\pi)^d}\frac{-\mathbf{k}_\perp\cdot\gamma_\perp - \delta_{\mathbf{k}}\gamma_{d-1}}{\mathbf{k}_\perp^2 + p_0^2 - \delta_{\mathbf{k}}^2},$$

$$\Sigma_a = -\Delta_b \int \frac{d^d k}{(2\pi)^d}\frac{1}{\mathbf{k}_\perp^2 + p_0^2 + \delta_{\mathbf{k}}^2},$$

$$\Sigma_b = \Delta_b \int \frac{d^d k}{(2\pi)^d}\frac{-2k_{\perp,i}^2 + \mathbf{k}_\perp^2 + \delta_{\mathbf{k}}^2 + p_0^2}{[\mathbf{k}_\perp^2 + p_0^2 + \delta_{\mathbf{k}}^2]^2},$$

$$\Sigma_c = -\Delta_b \int \frac{d^d k}{(2\pi)^d}\frac{-\mathbf{k}_\perp^2 + \delta_{\mathbf{k}}^2 - p_0^2}{[\mathbf{k}_\perp^2 + p_0^2 + \delta_{\mathbf{k}}^2]^2}.$$

The above expressions are similar to those of $\Sigma(2)$. As a result, we obtain

$$\Sigma(3) = -\frac{\tilde{\Delta}_b}{\epsilon}(ip_0\gamma_0) - \frac{\tilde{\Delta}_b}{2\epsilon}(i\delta_{\mathbf{p}}\gamma_{d-1}) - \frac{\tilde{\Delta}_b}{2\epsilon}(ik_f\gamma_{d-1}),$$

where $\tilde{\Delta}_b = \frac{\sqrt{2}S\Delta_b}{4}$.

## B One-loop vertex corrections

### B.1 Forward Scattering

#### B.1.1 Feynman diagram FV1-1

The vertex correction in Table 3 FV1-1 is given as

$$\mathcal{M}(1) = \Delta_f^2 \int \frac{d^{d+1}k}{(2\pi)^d}\delta(k_0)\gamma_{d-1}G_0(k+p_1)\gamma_{d-1}\otimes\gamma_{d-1}G_0(-k+p_2)\gamma_{d-1}$$

$$= -\Delta_f^2 \int \frac{d^{d+1}k}{(2\pi)^d}\delta(k_0)\frac{\mathcal{N}}{\mathcal{D}},$$

Table 3: Feynman diagrams for one-loop self-energy corrections. Here, $A_g$, $A_{\Delta_f}$, and $A_{\Delta_f}$ represent the coefficient of the $\epsilon$ poles computed from the corresponding Feynman diagrams (see Eq. (E.10) for the definition).

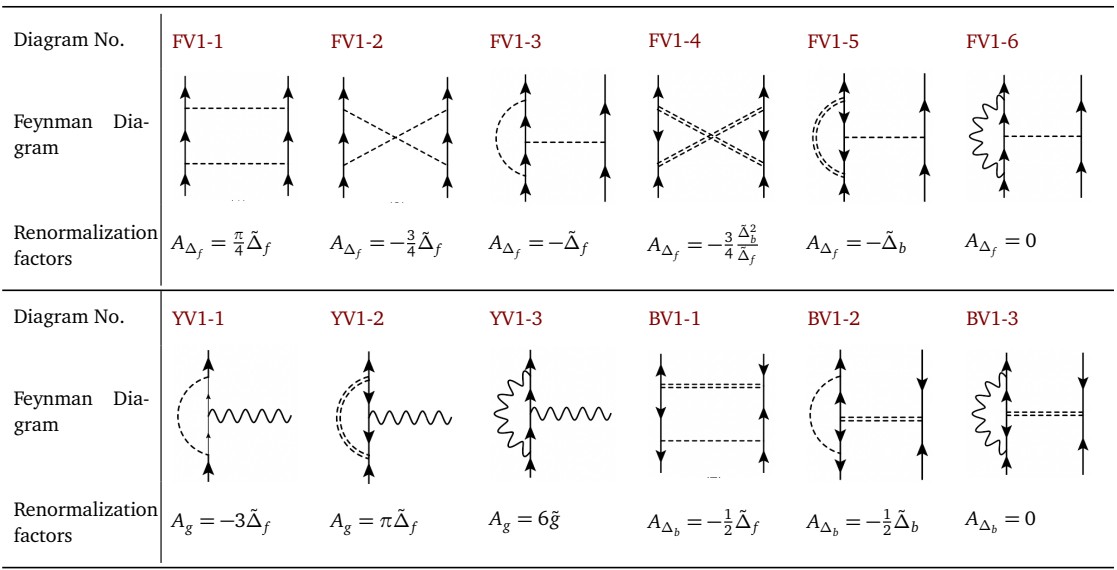

| Diagram No. | FV1-1 | FV1-2 | FV1-3 | FV1-4 | FV1-5 | FV1-6 |
|---|---|---|---|---|---|---|
| Feynman Diagram | | | | | | |
| Renormalization factors | $A_{\Delta_f} = \frac{\pi}{4}\tilde{\Delta}_f$ | $A_{\Delta_f} = -\frac{3}{4}\tilde{\Delta}_f$ | $A_{\Delta_f} = -\tilde{\Delta}_f$ | $A_{\Delta_f} = -\frac{3}{4}\frac{\tilde{\Delta}_b^2}{\tilde{\Delta}_f}$ | $A_{\Delta_f} = -\tilde{\Delta}_b$ | $A_{\Delta_f} = 0$ |
| Diagram No. | YV1-1 | YV1-2 | YV1-3 | BV1-1 | BV1-2 | BV1-3 |
| Feynman Diagram | | | | | | |
| Renormalization factors | $A_g = -3\tilde{\Delta}_f$ | $A_g = \pi\tilde{\Delta}_f$ | $A_g = 6\tilde{g}$ | $A_{\Delta_b} = -\frac{1}{2}\tilde{\Delta}_f$ | $A_{\Delta_b} = -\frac{1}{2}\tilde{\Delta}_b$ | $A_{\Delta_b} = 0$ |

where $\mathcal{D}$ and $\mathcal{N}$ are given by

$$\mathcal{D} = \left[(\mathbf{K}+\mathbf{P}_1)^2 + \delta_{\mathbf{k}+\mathbf{p}_1}^2\right]\left[(\mathbf{K}-\mathbf{P}_2)^2 + \delta_{-\mathbf{k}+\mathbf{p}_2}^2\right],$$
$$\mathcal{N} = \delta_{\mathbf{k}+\mathbf{p}_1}\gamma_{d-1} \otimes \delta_{-\mathbf{k}+\mathbf{p}_2}\gamma_{d-1} - (\mathbf{K}+\mathbf{P}_1)\cdot\gamma \otimes (\mathbf{K}-\mathbf{P}_2)\cdot\gamma$$
$$- (\mathbf{K}+\mathbf{P}_1)\cdot\gamma \otimes \delta_{-\mathbf{k}+\mathbf{p}_2}\gamma_{d-1} + \delta_{\mathbf{k}+\mathbf{p}_1}\gamma_{d-1} \otimes (\mathbf{K}-\mathbf{P}_2)\cdot\gamma.$$

In the numerator, there are four terms whose matrices are given by $\gamma_{d-1} \otimes \gamma_{d-1}$, $\gamma_i \otimes \gamma_i$, $\gamma_i \otimes \gamma_{d-1}$, and $\gamma_{d-1} \otimes \gamma_i$ with $i = 1, \cdots, d-2$. The first two would diverge while the latter two would vanish after being integrated over $\mathbf{K}$. Among the two non-vanishing terms, the term for $\gamma_{d-1} \otimes \gamma_{d-1}$ gives a renormalization factor for $\Delta_f$. On the other hand, the term for $\gamma_i \otimes \gamma_i$ is an artifact stemming from the generalization of the dimension from $d = 2$ to general $d$, and it should be eliminated by a counterterm. From now on, we focus on the term $\gamma_{d-1} \otimes \gamma_{d-1}$ giving the renormalization factor.

For future use, we define the following quantity:

$$\delta\Delta_f(a) \equiv \lim_{\{p_i\}\to 0} \frac{1}{4}\mathrm{tr}\left[\mathcal{M}(a)\gamma_{d-1} \otimes \gamma_{d-1}\right], \tag{B.1}$$

where $\{p_i\}$ denote external momenta such as $p_1, p_2$ in $\mathcal{M}(1)$. This quantity is directly related to a renormalization factor, so we just call it a "renormalization factor".

Using Eq. (B.1), we find the renormalization factor $\delta\Delta_f(1)$ as

$$\delta\Delta_f(1) = -\Delta_f^2 \int \frac{d\mathbf{k}_\perp}{(2\pi)^{d-2}} \int_{-\infty}^{\infty} \frac{dk_x}{2\pi} \int_{-\infty}^{\infty} \frac{dk_y}{2\pi} \frac{(k_x + k_y^2)(-k_x + k_y^2)}{\left[(k_x + k_y^2)^2 + \mathbf{k}_\perp^2\right]\left[(-k_x + k_y^2)^2 + \mathbf{k}_\perp^2\right]}.$$

Scaling variables as $k_x \to |\mathbf{k}_\perp| k_x$ and $k_y \to \sqrt{|\mathbf{k}_\perp|} k_y$, we obtain

$$\delta\Delta_f(1) = -S_{d-2}\Delta_f^2 \int_{p_0}^{\infty} dk_\perp k_\perp^{d-\frac{7}{2}} \int_{-\infty}^{\infty} \frac{dk_x}{2\pi} \int_{-\infty}^{\infty} \frac{dk_y}{2\pi} \frac{(k_x + k_y^2)(-k_x + k_y^2)}{\left[(k_x + k_y^2)^2 + 1\right]\left[(-k_x + k_y^2)^2 + 1\right]},$$

where $S_{d-2} = 2/((4\pi)^{\frac{d-2}{2}}\Gamma(\frac{d-2}{2}))$. We point out that $p_0$ should be introduced as a lower cutoff for the infrared convergence. We find an $\epsilon$ pole from the $k_\perp$-integral as $\int_{p_0}^\infty dk_\perp k_\perp^{d-7/2} = \frac{1}{\epsilon} + \mathcal{O}(1)$. The remaining integral is done as

$$\int_{-\infty}^\infty \frac{dx}{2\pi} \int_{-\infty}^\infty \frac{dy}{2\pi} \frac{(x+y^2)(-x+y^2)}{\left[(x+y^2)^2+1\right]\left[(-x+y^2)^2+1\right]} = -\frac{\sqrt{2}}{16}.$$

As a result, we obtain

$$\delta\Delta_f(1) = \frac{\pi\Delta_f\tilde{\Delta}_f}{4\epsilon}.$$

### B.1.2 Feynman diagram FV1-2

From the vertex correction in Table 3 FV1-2, we find the renormalization factor $\delta\Delta_f(2)$ as

$$\delta\Delta_f(2) = -\Delta_f^2 \int \frac{d\mathbf{k}_\perp}{(2\pi)^{d-2}} \int \frac{dk_x dk_y}{(2\pi)^2} \frac{\delta_\mathbf{k}^2}{\left[\delta_\mathbf{k}^2 + \mathbf{k}_\perp^2\right]^2}.$$

We encounter the same divergence as with $\Sigma(2)$. Regularizing the $k_x$-integral with $k_f$, we obtain

$$\delta\Delta_f(2) = -\Delta_f^2 \int \frac{d\mathbf{k}_\perp}{(2\pi)^{d-2}} \int_{-k_f}^\infty \frac{dk_x}{2\pi} \int_{-\infty}^\infty \frac{dk_y}{2\pi} \frac{(k_x+k_y^2)^2}{\left[(k_x+k_y^2)^2 + \mathbf{k}_\perp^2\right]^2}.$$

To find an $\epsilon$ pole, we may set $k_f = 0$ as proven in Eq. (A.2). Scaling variables as $k_x \to |\mathbf{k}_\perp|k_x$ and $k_y \to \sqrt{|\mathbf{k}_\perp|}k_y$, we have

$$\delta\Delta_f(2) = -S_{d-2}\Delta_f^2 \int_{p_0}^\infty dk_\perp k_\perp^{d-\frac{7}{2}} \int_0^\infty \frac{dk_x}{2\pi} \int_{-\infty}^\infty \frac{dk_y}{2\pi} \frac{(k_x+k_y^2)^2}{\left[(k_x+k_y^2)^2 + 1\right]^2} + \mathcal{O}(1).$$

We find an $\epsilon$ pole from the $k_\perp$-integral as $\int_{p_0}^\infty dk_\perp k_\perp^{d-7/2} = \frac{1}{\epsilon} + \mathcal{O}(1)$. The remaining integral is done as

$$\int_0^\infty \frac{dx}{2\pi} \int_{-\infty}^\infty \frac{dy}{2\pi} \frac{(x+y^2)^2}{\left[(x+y^2)^2+1\right]^2} = \frac{3\sqrt{2}}{16\pi}.$$

As a result, we obtain

$$\delta\Delta_f(2) = -\frac{3\Delta_f\tilde{\Delta}_f}{4\epsilon}.$$

### B.1.3 Feynman diagram FV1-3

From the vertex correction in Table 3 FV1-3, we find the renormalization factor $\delta\Delta_f(3)$ as

$$\delta\Delta_f(3) = -2\Delta_f^2 \int \frac{d\mathbf{k}_\perp}{(2\pi)^{d-2}} \int_{-k_f}^\infty \frac{dk_x}{2\pi} \int_{-\infty}^\infty \frac{dk_y}{2\pi} \frac{(k_x+k_y^2)^2 - \mathbf{k}_\perp^2}{\left[(k_x+k_y^2)^2 + \mathbf{k}_\perp^2\right]^2}.$$

Setting $k_f = 0$ and scaling variables as $k_x \to |\mathbf{k}_\perp|k_x$ and $k_y \to \sqrt{|\mathbf{k}_\perp|}k_y$, we have

$$\delta\Delta_f(3) = -2S_{d-2}\Delta_f^2 \int_{p_0}^\infty dk_\perp k_\perp^{d-\frac{7}{2}} \int_0^\infty \frac{dk_x}{2\pi} \int_{-\infty}^\infty \frac{dk_y}{2\pi} \frac{(k_x+k_y^2)^2 - 1}{\left[(k_x+k_y^2)^2 + 1\right]^2}.$$

We find an $\epsilon$ pole from the $k_\perp$-integral as $\int_{p_0}^\infty dk_\perp k_\perp^{d-7/2} = \frac{1}{\epsilon} + \mathcal{O}(1)$. The remaining integral is done as

$$\int_0^\infty \frac{dx}{2\pi} \int_{-\infty}^\infty \frac{dy}{2\pi} \frac{(x+y^2)^2 - 1}{\left[(x+y^2)^2 + 1\right]^2} = \frac{\sqrt{2}}{8\pi}.$$

As a result, we obtain

$$\delta\Delta_f(3) = -\frac{\Delta_f \tilde{\Delta}_f}{\epsilon}.$$

### B.1.4   Feynman diagram FV1-4

From the vertex correction in Table 3 FV1-4, we find the renormalization factor $\delta\Delta_f(4)$ as

$$\delta\Delta_f(4) = -\Delta_b^2 \int \frac{d\mathbf{k}_\perp}{(2\pi)^{d-2}} \int_{-k_f}^\infty \frac{dk_x}{2\pi} \int_{-\infty}^\infty \frac{dk_y}{2\pi} \frac{(k_x + k_y^2)^2}{\left[(k_x + k_y^2)^2 + \mathbf{k}_\perp^2\right]^2}.$$

The integration is the same with $\delta\Delta_f(2)$. As a result, we obtain

$$\delta\Delta_f(4) = -\frac{3\Delta_b \tilde{\Delta}_b}{4\epsilon}.$$

### B.1.5   Feynman diagram FV1-5

From the vertex correction in Table 3 FV1-5, we find the renormalization factor $\delta\Delta_f(5)$ as

$$\delta\Delta_f(5) = -2\Delta_f \Delta_b \int \frac{d\mathbf{k}_\perp}{(2\pi)^{d-2}} \int_{-k_f}^\infty \frac{dk_x}{2\pi} \int_{-\infty}^\infty \frac{dk_y}{2\pi} \frac{(k_x + k_y^2)^2 - \mathbf{k}_\perp^2}{\left[(k_x + k_y^2)^2 + \mathbf{k}_\perp^2\right]^2}.$$

The integration is the same with $\delta\Delta_f(3)$. As a result, we obtain

$$\delta\Delta_f(5) = -\frac{\Delta_f \tilde{\Delta}_b}{\epsilon}.$$

### B.1.6   Feynman diagram FV1-6

From the vertex correction in Table 3 FV1-6, we find the renormalization factor $\delta\Delta_f(6)$ as

$$\delta\Delta_f(6) = -\frac{2g^2 \Delta_f}{N} \int \frac{d\mathbf{K}}{(2\pi)^{d-1}} \int_{-\infty}^\infty \frac{dk_x}{2\pi} \int_{-\infty}^\infty \frac{dk_y}{2\pi} \frac{(k_x + k_y^2)^2 - \mathbf{K}^2}{\left[(k_x + k_y^2)^2 + \mathbf{K}^2\right]^2 \left[k_y^2 + g^2 B_d \frac{|\mathbf{K}|^{d-1}}{|k_y|}\right]}.$$

Shifting $k_x \to k_x - k_y^2$ and scaling variables as $k_x \to |\mathbf{K}|k_x$ and $k_y \to [g^2 B_d |\mathbf{K}|^{d-1}]^{1/3} k_y$, we have

$$\delta\Delta_f(6) = -\frac{2S_{d-1} g^{4/3} \Delta_f}{B_d^{1/3} N} \int_{|\mathbf{p}|}^\infty dK K^{\frac{2d-8}{3}} \int_{-\infty}^\infty \frac{dk_x}{2\pi} \int_{-\infty}^\infty \frac{dk_y}{2\pi} \frac{k_x^2 - 1}{\left[k_x^2 + 1\right]^2 \left[k_y^2 + 1/|k_y|\right]},$$

where $S_{d-1} = 2/((4\pi)^{d-1}\Gamma(\frac{d-1}{2}))$. Integrated over $k_x$, this correction vanishes due to the following identity: $\int_{-\infty}^\infty dx \frac{x^2-1}{(x^2+1)^2} = 0$. As a result, we obtain

$$\delta\Delta_f(6) = 0.$$

## B.2 Backward Scattering

### B.2.1 Feynman diagram BV1-1

The vertex correction in Table 3 BV1-1 is

$$\mathcal{M}(7) = 4\Delta_b\Delta_f \int \frac{d^{d+1}k}{(2\pi)^d} \delta(k_0) G_0(k-p_1)\gamma_{d-1} \otimes G_0(k-p_3)\gamma_{d-1}.$$

Similarly with Eq. (B.1), we define

$$\delta\Delta_b(a) \equiv \lim_{\{p_i\}\to 0} \frac{1}{4}\text{tr}\Big[\mathcal{M}(a)I_{2\times 2} \otimes I_{2\times 2}\Big]. \tag{B.2}$$

Using this formula, we find the renormalization factor $\delta\Delta_b(7)$ as

$$\delta\Delta_b(7) = -4\Delta_b\Delta_f \int \frac{d\mathbf{k}_\perp}{(2\pi)^{d-2}} \int_{-k_f}^\infty \frac{dk_x}{2\pi} \int_{-\infty}^\infty \frac{dk_y}{2\pi} \frac{(k_x+k_y^2)^2}{\Big[(k_x+k_y^2)^2 + \mathbf{k}_\perp^2\Big]^2}.$$

The integration is the same with $\delta\Delta_f(2)$. As a result, we obtain

$$\delta\Delta_b(7) = -\frac{3\Delta_b\tilde{\Delta}_f}{\epsilon}.$$

### B.2.2 Feynman diagram BV1-2

From the vertex correction in Table 3 BV1-2, we find the renormalization factor $\delta\Delta_b(8)$ as

$$\delta\Delta_b(8) = -2\Delta_b\Delta_f \int \frac{d\mathbf{k}_\perp}{(2\pi)^{d-2}} \int_{-\infty}^\infty \frac{dk_x}{2\pi} \int_{-\infty}^\infty \frac{dk_y}{2\pi} \frac{(k_x+k_y^2)(-k_x+k_y^2)-\mathbf{k}_\perp^2}{\Big[(k_x+k_y^2)^2+\mathbf{k}_\perp^2\Big]\Big[(-k_x+k_y^2)^2+\mathbf{k}_\perp^2\Big]}.$$

Scaling variables as $k_x \to |\mathbf{k}_\perp|k_x$ and $k_y \to \sqrt{|\mathbf{k}_\perp|}k_y$, we have

$$\delta\Delta_b(8) = 2S_{d-2}\Delta_b\Delta_f \int_{p_0}^\infty dk_\perp k_\perp^{d-\frac{7}{2}} \int_{-\infty}^\infty \frac{dk_x}{2\pi} \int_{-\infty}^\infty \frac{dk_y}{2\pi} \frac{(k_x+k_y^2)(k_x-k_y^2)+1}{\Big[(k_x+k_y^2)^2+1\Big]\Big[(k_x-k_y^2)^2+1\Big]}.$$

We find an $\epsilon$ pole from the $k_\perp$ integral as $\int_{p_0}^\infty dk_\perp k_\perp^{d-7/2} = \frac{1}{\epsilon}$. The remaining integral can be done to give

$$\int_{-\infty}^\infty \frac{dx}{2\pi} \int_{-\infty}^\infty \frac{dy}{2\pi} \frac{(x+y^2)(x-y^2)+1}{\Big[(x+y^2)^2+1\Big]\Big[(x-y^2)^2+1\Big]} = \frac{\sqrt{2}}{8}.$$

As a result, we obtain

$$\delta\Delta_b(8) = \frac{\pi\Delta_b\tilde{\Delta}_f}{\epsilon}.$$

### B.2.3 Feynman diagram BV1-3

From the vertex correction in Table 3 BV1-3, we find the renormalization factor $\delta\Delta_b(9)$ as

$$\delta\Delta_b(9) = -\frac{2g^2\Delta_b}{N} \int \frac{d\mathbf{K}}{(2\pi)^{d-1}} \int_{-\infty}^\infty \frac{dk_x}{2\pi} \int_{-\infty}^\infty \frac{dk_y}{2\pi} \frac{(k_x+k_y^2)(-k_x+k_y^2)-\mathbf{K}^2}{\Big[(k_x+k_y^2)^2+\mathbf{K}^2\Big]\Big[(-k_x+k_y^2)^2+\mathbf{K}^2\Big]} \frac{1}{\Big[k_y^2+g^2B_d\frac{|\mathbf{K}|^{d-1}}{|k_y|}\Big]}.$$

Scaling variables as $k_x \to |\mathbf{K}|k_x$ and $k_y \to [g^2 B_d |\mathbf{K}|^{d-1}]^{1/3} k_y$, we have

$$\delta\Delta_b(9) = \frac{2S_{d-1}g^{4/3}\Delta_b}{B_d^{1/3}N} \int_{|\mathbf{p}|}^{\infty} dK K^{\frac{2d-8}{3}} \int_{-\infty}^{\infty} \frac{dk_x}{2\pi} \int_{-\infty}^{\infty} \frac{dk_y}{2\pi} \frac{(k_x + C_{|\mathbf{K}|}k_y^2)(k_x - C_{|\mathbf{K}|}k_y^2) + 1}{\left[(k_x + C_{|\mathbf{K}|}k_y^2)^2 + 1\right]^2 \left[k_y^2 + 1/|k_y|\right]},$$

where $C_{|\mathbf{K}|} = [g^2 B_d |\mathbf{K}|^{d-1}]^{2/3}/|\mathbf{K}|$. Since $C_{|\mathbf{K}|}$ is proportional to $g^{4/3}$, it remains to be small as long as the coupling $e$ is small.

Expanding this expression in terms of $C_{|\mathbf{K}|}$, we have

$$\delta\Delta_b(9) = \frac{2S_{d-1}g^{4/3}\Delta_b}{B_d^{1/3}N} \int_{|\mathbf{p}|}^{\infty} dK K^{\frac{2d-8}{3}} \int_{-\infty}^{\infty} \frac{dk_x}{2\pi} \int_{-\infty}^{\infty} \frac{dk_y}{2\pi} \left[ \frac{1}{\left[k_x^2 + 1\right]\left[k_y^2 + 1/|k_y|\right]} - \frac{(k_x^2 - 3)C_{|\mathbf{K}|}^2 k_y^4}{\left[k_x^2 + 1\right]^3 \left[k_y^2 + 1/|k_y|\right]} \right],$$

up to $\mathcal{O}(C_{|\mathbf{K}|}^4)$ terms. The second term is proportional to $(g^{4/3})^3$, so it is comparable to three-loop corrections. Dropping this term, we have

$$\delta\Delta_b(9) = \frac{2S_{d-1}g^{4/3}\Delta_b}{B_d^{1/3}N} \int_{|\mathbf{p}|}^{\infty} dK K^{\frac{2d-8}{3}} \int_{-\infty}^{\infty} \frac{dk_x}{2\pi} \int_{-\infty}^{\infty} \frac{dk_y}{2\pi} \frac{1}{\left[k_x^2 + 1\right]\left[k_y^2 + 1/|k_y|\right]}.$$

We find an $\epsilon$ pole from the K integral as $\int_{|\mathbf{p}|}^{\infty} dK K^{\frac{2d-8}{3}} = \frac{3}{2\epsilon}$. The remaining integral is done as

$$\int_{-\infty}^{\infty} \frac{dx}{2\pi} \int_{-\infty}^{\infty} \frac{dy}{2\pi} \frac{1}{\left[x^2 + 1\right]\left[y^2 + 1/|y|\right]} = \frac{1}{3\sqrt{3}}.$$

As a result, we obtain

$$\delta\Delta_b(9) = \frac{6\Delta_b \tilde{g}}{\epsilon}.$$

### B.3 Yukawa coupling

#### B.3.1 Feynman diagram YV1-1

The vertex correction in Table 3 YV1-1 is

$$\mathcal{M}(10) = \frac{ig\Delta_f}{\sqrt{N}} \int \frac{d^{d+1}k}{(2\pi)^d} \delta(k_0)\gamma_{d-1}G_0(k+p_1)\gamma_{d-1}G_0(k+p_2)\gamma_{d-1}.$$

Similarly with Eq. (B.1), we define

$$i\delta g(a) \equiv \lim_{\{p_i\}\to 0} \frac{1}{2}\mathrm{tr}\left[\mathcal{M}(a)\gamma_{d-1}\right]. \tag{B.3}$$

Using Eq. (B.3), we find the renormalization factor $\delta g(10)$ as

$$\delta g(10) = -\frac{g\Delta_f}{\sqrt{N}} \int \frac{d\mathbf{k}_\perp}{(2\pi)^{d-2}} \int_{-k_f}^{\infty} \frac{dk_x}{2\pi} \int_{-\infty}^{\infty} \frac{dk_y}{2\pi} \frac{(k_x + k_y^2)^2 - \mathbf{k}_\perp^2}{\left[(k_x + k_y^2)^2 + \mathbf{k}_\perp^2\right]^2}.$$

The integration is the same with $\delta\Delta_f(3)$. As a result, we obtain

$$\delta g(10) = -\frac{g}{\sqrt{N}} \frac{\tilde{\Delta}_f}{2\epsilon}.$$

### B.3.2 Feynman diagram YV1-2

From the vertex correction in Table 3 YV1-2, we find the renormalization factor $\delta g(11)$ as

$$\delta g(11) = -\frac{g\Delta_b}{\sqrt{N}} \int \frac{d\mathbf{k}_\perp}{(2\pi)^{d-2}} \int_{-k_f}^{\infty} \frac{dk_x}{2\pi} \int_{-\infty}^{\infty} \frac{dk_y}{2\pi} \frac{(k_x + k_y^2)^2 - \mathbf{k}_\perp^2}{\left[(k_x + k_y^2)^2 + \mathbf{k}_\perp^2\right]^2}.$$

The integration is the same with $\delta\Delta_f(3)$. As a result, we obtain

$$\delta g(11) = -\frac{g}{\sqrt{N}} \frac{\tilde{\Delta}_b}{2\epsilon}.$$

### B.3.3 Feynman diagram YV1-3

From the vertex correction in Table 3 YV1-3, we find the renormalization factor $\delta g(12)$ as

$$\delta g(12) = -\frac{g^3}{N^{3/2}} \int \frac{d\mathbf{K}}{(2\pi)^{d-1}} \int_{-\infty}^{\infty} \frac{dk_x}{2\pi} \int_{-\infty}^{\infty} \frac{dk_y}{2\pi} \frac{(k_x + k_y^2)^2 - \mathbf{K}^2}{\left[(k_x + k_y^2)^2 + \mathbf{K}^2\right]^2 \left[k_y^2 + g^2 B_d \frac{|\mathbf{K}|^{d-1}}{|k_y|}\right]}.$$

Shifting $k_x \to k_x - k_y^2$ and scaling variables as $k_x \to |\mathbf{K}|k_x$ and $k_y \to [g^2 B_d |\mathbf{K}|^{d-1}]^{1/3}k_y$, we have

$$\delta g(12) = -\frac{S_{d-1} g^{7/3}}{B_d^{1/3} N^{3/2}} \int_{|\mathbf{p}|}^{\infty} dK K^{\frac{2d-11}{3}} \int_{-\infty}^{\infty} \frac{dk_x}{2\pi} \int_{-\infty}^{\infty} \frac{dk_y}{2\pi} \frac{k_x^2 - 1}{\left[k_x^2 + 1\right]^2 \left[k_y^2 + 1/|k_y|\right]}.$$

Integrated over $k_x$, this vanishes. As a result, we obtain

$$\delta g(12) = 0.$$

## C  Two-loop self-energy corrections

### C.1  Boson Self-energy

### C.1.1  Feynman diagram BS2-1

The boson self-energy in Table 4 BS2-1 is given by

$$\Pi_2(1) = -\frac{g^4 \mu^{2\epsilon}}{N} \int \frac{d^{d+1}k d^{d+1}l}{(2\pi)^{2d+2}} \text{tr}\left[\gamma_{d-1} G_0(k+q)\gamma_{d-1} G_0(k)\gamma_{d-1} G_0(l)\gamma_{d-1} G_0(l+q)\right] D_1(k-l)$$

$$= -\frac{2g^4 \mu^{2\epsilon}}{N} \int \frac{d^{d+1}k d^{d+1}l}{(2\pi)^{2d+2}} \frac{\mathcal{N}}{\mathcal{D}} D_1(k-l),$$

where $\mathcal{D}$ and $\mathcal{N}$ are

$$\mathcal{D} = \left[(\mathbf{K}+\mathbf{Q})^2 + \delta_{k+q}^2\right]\left[\mathbf{K}^2 + \delta_k^2\right]\left[\mathbf{L}^2 + \delta_l^2\right]\left[(\mathbf{L}+\mathbf{Q})^2 + \delta_{l+q}^2\right], \tag{C.1a}$$

$$\mathcal{N} = \left[\delta_k \delta_{k+q} - \mathbf{K}\cdot(\mathbf{K}+\mathbf{Q})\right]\left[\delta_l \delta_{l+q} - \mathbf{L}\cdot(\mathbf{L}+\mathbf{Q})\right] \tag{C.1b}$$

$$- \left[\delta_k \delta_{l+q} + \mathbf{K}\cdot(\mathbf{L}+\mathbf{Q})\right]\left[\delta_l \delta_{k+q} + \mathbf{L}\cdot(\mathbf{K}+\mathbf{Q})\right] + \left[\delta_k \delta_l - \mathbf{K}\cdot\mathbf{L}\right]\left[\delta_{k+q}\delta_{l+q} - (\mathbf{K}+\mathbf{Q})\cdot(\mathbf{L}+\mathbf{Q})\right].$$

Integrating over $k_x$, we have

$$\Pi_2(1) = -\frac{g^4 \mu^{2\epsilon}}{N} \int \frac{d^{d+1}k d\mathbf{L} dl_y}{(2\pi)^{2d+1}} \frac{\mathcal{N}_1}{\mathcal{D}_1} D_1(k-l),$$

Table 4: Feynman diagrams for one-loop self-energy corrections. Here, $A_0$, $A_1$, and $A_2$ represent the coefficient of the $\epsilon$ poles computed from the corresponding Feynman diagrams (see Eq. (E.10) for the definition). $\Pi_2(q)$ represents two-loop corrections of Landau damping for the dressed boson propagator.

| Diagram No. | FS2-1 | FS2-2 | FS2-3 | FS2-4 |
|---|---|---|---|---|
| Feynman Diagram | | | | |
| Renormalization factors | $A_0 = -0.3361\tilde{g}^2,$ $A_1 = -0.3361\tilde{g}^2$ $A_2 = -0.1131\tilde{g}^2$ | $A_0 = -0.00006139\tilde{\Delta}_f^2,$ $A_1 = -0.001490\tilde{\Delta}_f^2$ | $A_0 = -0.4461\tilde{\Delta}_f\sqrt{\frac{\tilde{g}}{N}},$ $A_1 = -15.75\tilde{\Delta}_f\sqrt{\frac{\tilde{g}}{N}}$ | $A_0 = -0.0001228\tilde{\Delta}_f\tilde{\Delta}_b,$ $A_1 = -0.002980\tilde{\Delta}_f\tilde{\Delta}_b$ |
| Diagram No. | FS2-5 | BS2-1 | BS2-2 | BS2-3 |
| Feynman Diagram | | | | |
| Renormalization factors | $A_0 = -0.4461\tilde{\Delta}_b\sqrt{\frac{\tilde{g}}{N}},$ $A_1 = -15.75\tilde{\Delta}_b\sqrt{\frac{\tilde{g}}{N}}$ | $\Pi_2(q) = (0.6427\tilde{g})g^2\mu^\epsilon B_d\frac{|\mathbf{Q}|^{d-1}}{|q_y|}$ | $\Pi_2(q) = -0.05025g^2\tilde{\Delta}_f\mu^{2\epsilon}\frac{|\mathbf{Q}|^{2d-3}}{|q_y|^2}$ | $\Pi_2(q) = -0.05025g^2\tilde{\Delta}_b\mu^{2\epsilon}\frac{|\mathbf{Q}|^{2d-3}}{|q_y|^2}$ |

where $\mathcal{D}_1$ and $\mathcal{N}_1$ are given by

$$\mathcal{D}_1 = \left[(2k_y q_y + \delta_{\mathbf{q}})^2 + (|\mathbf{K}| + |\mathbf{K}+\mathbf{Q}|)^2\right]\left[\delta_{\mathbf{l}}^2 + \mathbf{L}^2\right]\left[\delta_{\mathbf{l}+\mathbf{q}}^2 + (\mathbf{L}+\mathbf{Q})^2\right],$$

$$\mathcal{N}_1 = (|\mathbf{K}| + |\mathbf{K}+\mathbf{Q}|)\left[\left(1 - \frac{\mathbf{K}\cdot(\mathbf{K}+\mathbf{Q})}{|\mathbf{K}||\mathbf{K}+\mathbf{Q}|}\right)\delta_{\mathbf{l}}\delta_{\mathbf{l}+\mathbf{q}} - \mathbf{L}\cdot(\mathbf{L}+\mathbf{Q}) + \frac{\mathbf{K}\cdot(\mathbf{K}+\mathbf{Q})\mathbf{L}\cdot(\mathbf{L}+\mathbf{Q})}{|\mathbf{K}||\mathbf{K}+\mathbf{Q}|}\right.$$
$$\left. - \frac{\mathbf{L}\cdot(\mathbf{K}+\mathbf{Q})\mathbf{K}\cdot(\mathbf{L}+\mathbf{Q})}{|\mathbf{K}||\mathbf{K}+\mathbf{Q}|} + \frac{\mathbf{K}\cdot\mathbf{L}(\mathbf{K}+\mathbf{Q})\cdot(\mathbf{L}+\mathbf{Q})}{|\mathbf{K}||\mathbf{K}+\mathbf{Q}|}\right]$$
$$+ (2k_y q_y + \delta_{\mathbf{q}})\left[\delta_{\mathbf{l}+\mathbf{q}}\left(\frac{\mathbf{L}\cdot(\mathbf{K}+\mathbf{Q})}{|\mathbf{K}+\mathbf{Q}|} - \frac{\mathbf{K}\cdot\mathbf{L}}{|\mathbf{K}|}\right) + \delta_{\mathbf{l}}\left(\frac{(\mathbf{K}+\mathbf{Q})\cdot(\mathbf{L}+\mathbf{Q})}{|\mathbf{K}+\mathbf{Q}|} - \frac{\mathbf{K}\cdot(\mathbf{L}+\mathbf{Q})}{|\mathbf{K}|}\right)\right].$$

Integrating over $l_x$, we obtain

$$\Pi_2(1) = -\frac{g^4\mu^{2\epsilon}}{2N}\int\frac{d\mathbf{K}dk_y d\mathbf{L}dl_y}{(2\pi)^{2d}}\frac{\mathcal{N}_2}{\mathcal{D}_2}\frac{1}{(k_y - l_y)^2 + g^2\mu^\epsilon B_d\frac{|\mathbf{K}-\mathbf{L}|^{d-1}}{|k_y - l_y|}},$$

where $\mathcal{D}_2$ and $\mathcal{N}_2$ are given by

$$\mathcal{D}_2 = \left[(2k_y q_y + \delta_{\mathbf{q}})^2 + (|\mathbf{K}| + |\mathbf{K}+\mathbf{Q}|)^2\right]\left[(2l_y q_y + \delta_{\mathbf{q}})^2 + (|\mathbf{L}| + |\mathbf{L}+\mathbf{Q}|)^2\right],$$

$$\mathcal{N}_2 = (|\mathbf{K}| + |\mathbf{K}+\mathbf{Q}|)(|\mathbf{L}| + |\mathbf{L}+\mathbf{Q}|)\left[\left(1 - \frac{\mathbf{K}\cdot(\mathbf{K}+\mathbf{Q})}{|\mathbf{K}||\mathbf{K}+\mathbf{Q}|}\right) + \frac{\mathbf{K}\cdot(\mathbf{K}+\mathbf{Q})\mathbf{L}\cdot(\mathbf{L}+\mathbf{Q})}{|\mathbf{K}||\mathbf{K}+\mathbf{Q}||\mathbf{L}||\mathbf{L}+\mathbf{Q}|}\right.$$
$$\left. - \frac{\mathbf{L}\cdot(\mathbf{K}+\mathbf{Q})\mathbf{K}\cdot(\mathbf{L}+\mathbf{Q})}{|\mathbf{K}||\mathbf{K}+\mathbf{Q}||\mathbf{L}||\mathbf{L}+\mathbf{Q}|} + \frac{\mathbf{K}\cdot\mathbf{L}(\mathbf{K}+\mathbf{Q})\cdot(\mathbf{L}+\mathbf{Q})}{|\mathbf{K}||\mathbf{K}+\mathbf{Q}||\mathbf{L}||\mathbf{L}+\mathbf{Q}|} - \frac{\mathbf{L}\cdot(\mathbf{L}+\mathbf{Q})}{|\mathbf{L}||\mathbf{L}+\mathbf{Q}|}\right]$$
$$+ (2k_y q_y + \delta_{\mathbf{q}})(2l_y q_y + \delta_{\mathbf{q}})\left[\frac{\mathbf{L}\cdot(\mathbf{K}+\mathbf{Q})}{|\mathbf{L}||\mathbf{K}+\mathbf{Q}|} - \frac{\mathbf{K}\cdot\mathbf{L}}{|\mathbf{K}||\mathbf{L}|} - \frac{(\mathbf{K}+\mathbf{Q})\cdot(\mathbf{L}+\mathbf{Q})}{|\mathbf{K}+\mathbf{Q}||\mathbf{L}+\mathbf{Q}|} + \frac{\mathbf{K}\cdot(\mathbf{L}+\mathbf{Q})}{|\mathbf{K}||\mathbf{L}+\mathbf{Q}|}\right].$$

Shifting $l_y$ as $l_y \to l_y + k_y$ and integrating over $k_y$, we have

$$\Pi_2(1) = -\frac{g^4\mu^{2\epsilon}}{8N}\int\frac{d\mathbf{K}d\mathbf{L}dl_y}{(2\pi)^{2d-1}}\frac{\mathcal{N}_3}{\mathcal{D}_3}\frac{1}{l_y^2 + g^2\mu^\epsilon B_d\frac{|\mathbf{K}-\mathbf{L}|^{d-1}}{|l_y|}},$$

where $\mathcal{D}_3$ and $\mathcal{N}_3$ are given by

$$\mathcal{D}_3 = |q_y| \Big[ (2l_y q_y)^2 + (|\mathbf{K}| + |\mathbf{K} + \mathbf{Q}| + |\mathbf{L}| + |\mathbf{L} + \mathbf{Q}|)^2 \Big], \tag{C.2a}$$

$$\mathcal{N}_3 = (|\mathbf{K}| + |\mathbf{K} + \mathbf{Q}| + |\mathbf{L}| + |\mathbf{L} + \mathbf{Q}|) \Bigg[ \left( 1 - \frac{\mathbf{K} \cdot (\mathbf{K} + \mathbf{Q})}{|\mathbf{K}||\mathbf{K} + \mathbf{Q}|} \right) \left( 1 - \frac{\mathbf{L} \cdot (\mathbf{L} + \mathbf{Q})}{|\mathbf{L}||\mathbf{L} + \mathbf{Q}|} \right) \tag{C.2b}$$

$$- \left( 1 - \frac{\mathbf{K} \cdot (\mathbf{L} + \mathbf{Q})}{|\mathbf{K}||\mathbf{L} + \mathbf{Q}|} \right) \left( 1 - \frac{\mathbf{L} \cdot (\mathbf{K} + \mathbf{Q})}{|\mathbf{L}||\mathbf{K} + \mathbf{Q}|} \right) + \left( 1 - \frac{\mathbf{K} \cdot \mathbf{L}}{|\mathbf{K}||\mathbf{L}|} \right) \left( 1 - \frac{(\mathbf{K} + \mathbf{Q}) \cdot (\mathbf{L} + \mathbf{Q})}{|\mathbf{K} + \mathbf{Q}||\mathbf{L} + \mathbf{Q}|} \right) \Bigg].$$

We may neglect the $l_y q_y$ term in the fermionic part since it would give rise to subleading terms in $g$. Integrating over $l_y$, we obtain

$$\Pi_2(1) = -\frac{g^4 \mu^{2\epsilon}}{12\sqrt{3}N} \int \frac{d\mathbf{K} d\mathbf{L}}{(2\pi)^{2d-2}} \frac{\mathcal{N}_4}{\mathcal{D}_4},$$

where $\mathcal{D}_4$ and $\mathcal{N}_4$ are given by

$$\mathcal{D}_4 = |q_y| \Big[ g^2 \mu^\epsilon B_d |\mathbf{K} - \mathbf{L}|^{d-1} \Big]^{1/3} (|\mathbf{K}| + |\mathbf{K} + \mathbf{Q}| + |\mathbf{L}| + |\mathbf{L} + \mathbf{Q}|),$$

$$\mathcal{N}_4 = \left( 1 - \frac{\mathbf{K} \cdot (\mathbf{K} + \mathbf{Q})}{|\mathbf{K}||\mathbf{K} + \mathbf{Q}|} \right) \left( 1 - \frac{\mathbf{L} \cdot (\mathbf{L} + \mathbf{Q})}{|\mathbf{L}||\mathbf{L} + \mathbf{Q}|} \right) - \left( 1 - \frac{\mathbf{K} \cdot (\mathbf{L} + \mathbf{Q})}{|\mathbf{K}||\mathbf{L} + \mathbf{Q}|} \right) \left( 1 - \frac{\mathbf{L} \cdot (\mathbf{K} + \mathbf{Q})}{|\mathbf{L}||\mathbf{K} + \mathbf{Q}|} \right)$$

$$+ \left( 1 - \frac{\mathbf{K} \cdot \mathbf{L}}{|\mathbf{K}||\mathbf{L}|} \right) \left( 1 - \frac{(\mathbf{K} + \mathbf{Q}) \cdot (\mathbf{L} + \mathbf{Q})}{|\mathbf{K} + \mathbf{Q}||\mathbf{L} + \mathbf{Q}|} \right).$$

Introducing coordinates of $\mathbf{K} \cdot \mathbf{Q} = K|\mathbf{Q}| \cos\theta_k$, $\mathbf{L} \cdot \mathbf{Q} = L|\mathbf{Q}| \cos\theta_l$, and $\mathbf{K} \cdot \mathbf{L} = KL \cos\theta_{kl}$, where $K = |\mathbf{K}|$, $L = |\mathbf{L}|$, and $\cos\theta_{kl} = \cos\theta_k \cos\theta_l + \sin\theta_k \sin\theta_l \cos\phi_l$, and changing variables as $K = |\mathbf{Q}|k$ and $L = |\mathbf{Q}|l$, we have

$$\Pi_2(1) = -\frac{g^{10/3} \mu^\epsilon |\mathbf{Q}|^{d-1} (\mu/|\mathbf{Q}|)^{\frac{2\epsilon}{3}}}{12\sqrt{3}|q_y|B_d^{1/3}N} \frac{4}{(4\pi)^{d-1}\pi\sqrt{\pi}\Gamma(\frac{d-2}{2})\Gamma(\frac{d-3}{2})} \int_0^\infty dk\, k^{d-2} \int_0^\infty dl\, l^{d-2} \int_0^\pi d\theta_k \int_0^\pi d\theta_l \int_0^\pi d\phi_l$$

$$\times \frac{\sin^{d-3}\theta_k \sin^{d-3}\theta_l \sin^{d-4}\phi_l}{(k + \eta_1 + l + \eta_2)[k^2 + l^2 - 2kl\cos\theta_{kl}]^{\frac{d-1}{6}}} \Bigg[ \left( 1 - \frac{k + \cos\theta_k}{\eta_1} \right) \left( 1 - \frac{l + \cos\theta_l}{\eta_2} \right)$$

$$- \left( 1 - \frac{l\cos\theta_{kl} + \cos\theta_k}{\eta_2} \right) \left( 1 - \frac{k\cos\theta_{kl} + \cos\theta_l}{\eta_1} \right) + (1 - \cos\theta_{kl}) \left( 1 - \frac{kl\cos\theta_{kl} + k\cos\theta_k + l\cos\theta_l + 1}{\eta_1 \eta_2} \right) \Bigg],$$

where $\eta_1 = \sqrt{k^2 + 1 + 2k\cos\theta_k}$ and $\eta_2 = \sqrt{l^2 + 1 + 2l\cos\theta_l}$. The remaining integrals can be done numerically to give

$$\int_0^\infty dk \int_0^\infty dl \int_0^\pi d\theta_k \int_0^\pi d\theta_l \int_0^\pi d\phi_l \frac{\sqrt{kl}\sin^{-\frac{1}{2}}\theta_k \sin^{-\frac{1}{2}}\theta_l \sin^{d-4}\phi_l}{(k + \eta_1 + l + \eta_2)[k^2 + l^2 - 2kl\cos\theta_{kl}]^{\frac{1}{4}}} \Bigg[ \left( 1 - \frac{k + \cos\theta_k}{\eta_1} \right) \left( 1 - \frac{l + \cos\theta_l}{\eta_2} \right)$$

$$- \left( 1 - \frac{l\cos\theta_{kl} + \cos\theta_k}{\eta_2} \right) \left( 1 - \frac{k\cos\theta_{kl} + \cos\theta_l}{\eta_1} \right) + (1 - \cos\theta_{kl}) \left( 1 - \frac{kl\cos\theta_{kl} + k\cos\theta_k + l\cos\theta_l + 1}{\eta_1 \eta_2} \right) \Bigg]$$

$$= \frac{\sqrt{\pi}\Gamma(\frac{d-3}{2})}{\Gamma(\frac{d-2}{2})}(-7.723).$$

As a result, we obtain

$$\Pi_2(1) = -g^2 \mu^\epsilon (c\tilde{g}) B_d \frac{|\mathbf{Q}|^{d-1}}{|q_y|}, \qquad c = -0.6427. \tag{C.3}$$

### C.1.2  Feynman diagram BS2-2

The boson self-energy in Table 4 BS2-2 is expressed as

$$\Pi_2(2) = -g^2 \Delta_f \mu^{2\epsilon} \int \frac{d^{d+1}k\, d^{d+1}l}{(2\pi)^{2d+1}} \delta(k_0 - l_0) \mathrm{tr}\big[ \gamma_{d-1} G_0(k+q) \gamma_{d-1} G_0(k) \gamma_{d-1} G_0(l) \gamma_{d-1} G_0(l+q) \big]$$

$$= -2g^2 \Delta_f \mu^{2\epsilon} \int \frac{d^{d+1}k\, d^{d+1}l}{(2\pi)^{2d+1}} \delta(k_0 - l_0) \frac{\mathcal{N}}{\mathcal{D}},$$

where $\mathcal{D}$ and $\mathcal{N}$ are given in Eq. (C.1). Integrating over $k_x$, $l_x$, and $l_y$, where the integration is the same with $\Pi_2(1)$, we have

$$\Pi_2(2) = -\frac{g^2 \Delta_f \mu^{2\epsilon}}{8} \int \frac{d\mathbf{K} d\mathbf{L} dl_y}{(2\pi)^{2d-1}} \delta(k_0 - l_0) \frac{\mathcal{N}_3}{\mathcal{D}_3},$$

where $\mathcal{D}_3$ and $\mathcal{N}_3$ are given by

$$\mathcal{D}_3 = |q_y| \left[ (2l_y q_y)^2 + (|\mathbf{K}| + |\mathbf{K}+\mathbf{Q}| + |\mathbf{L}| + |\mathbf{L}+\mathbf{Q}|)^2 \right],$$

$$\mathcal{N}_3 = (|\mathbf{K}| + |\mathbf{K}+\mathbf{Q}| + |\mathbf{L}| + |\mathbf{L}+\mathbf{Q}|) \left[ \left( 1 - \frac{\mathbf{K}\cdot(\mathbf{K}+\mathbf{Q})}{|\mathbf{K}||\mathbf{K}+\mathbf{Q}|} \right) \left( 1 - \frac{\mathbf{L}\cdot(\mathbf{L}+\mathbf{Q})}{|\mathbf{L}||\mathbf{L}+\mathbf{Q}|} \right) \right.$$
$$\left. - \left( 1 - \frac{\mathbf{K}\cdot(\mathbf{L}+\mathbf{Q})}{|\mathbf{K}||\mathbf{L}+\mathbf{Q}|} \right) \left( 1 - \frac{\mathbf{L}\cdot(\mathbf{K}+\mathbf{Q})}{|\mathbf{L}||\mathbf{K}+\mathbf{Q}|} \right) + \left( 1 - \frac{\mathbf{K}\cdot\mathbf{L}}{|\mathbf{K}||\mathbf{L}|} \right) \left( 1 - \frac{(\mathbf{K}+\mathbf{Q})\cdot(\mathbf{L}+\mathbf{Q})}{|\mathbf{K}+\mathbf{Q}||\mathbf{L}+\mathbf{Q}|} \right) \right].$$

Integrating over $l_y$, we obtain

$$\Pi_2(2) = -\frac{g^2 \Delta_f \mu^{2\epsilon}}{32|q_y|^2} \int \frac{d\mathbf{K} d\mathbf{L}}{(2\pi)^{2d-3}} \delta(k_0 - l_0) \left[ \left( 1 - \frac{\mathbf{K}\cdot(\mathbf{K}+\mathbf{Q})}{|\mathbf{K}||\mathbf{K}+\mathbf{Q}|} \right) \left( 1 - \frac{\mathbf{L}\cdot(\mathbf{L}+\mathbf{Q})}{|\mathbf{L}||\mathbf{L}+\mathbf{Q}|} \right) \right.$$
$$\left. - \left( 1 - \frac{\mathbf{K}\cdot(\mathbf{L}+\mathbf{Q})}{|\mathbf{K}||\mathbf{L}+\mathbf{Q}|} \right) \left( 1 - \frac{\mathbf{L}\cdot(\mathbf{K}+\mathbf{Q})}{|\mathbf{L}||\mathbf{K}+\mathbf{Q}|} \right) + \left( 1 - \frac{\mathbf{K}\cdot\mathbf{L}}{|\mathbf{K}||\mathbf{L}|} \right) \left( 1 - \frac{(\mathbf{K}+\mathbf{Q})\cdot(\mathbf{L}+\mathbf{Q})}{|\mathbf{K}+\mathbf{Q}||\mathbf{L}+\mathbf{Q}|} \right) \right].$$

The second line is odd in $\mathbf{K}$ and $\mathbf{L}$, so it vanishes.

Integrating over $l_0$, we have

$$\Pi_2(2) = -\frac{g^2 \Delta_f \mu^{2\epsilon}}{32|q_y|^2} \int_{-\infty}^{\infty} \frac{dk_0}{2\pi} \int \frac{d\mathbf{k}_\perp}{(2\pi)^{d-2}} \left( 1 - \frac{\mathbf{k}_\perp \cdot (\mathbf{k}_\perp + \mathbf{q}_\perp) + k_0(k_0 + q_0)}{\sqrt{\mathbf{k}_\perp^2 + k_0^2} \sqrt{(\mathbf{k}_\perp + \mathbf{q}_\perp)^2 + (k_0 + q_0)^2}} \right)$$
$$\times \int \frac{d\mathbf{l}_\perp}{(2\pi)^{d-2}} \left( 1 - \frac{\mathbf{l}_\perp \cdot (\mathbf{l}_\perp + \mathbf{q}_\perp) + k_0(k_0 + q_0)}{\sqrt{\mathbf{l}_\perp^2 + k_0^2} \sqrt{(\mathbf{l}_\perp + \mathbf{q}_\perp)^2 + (k_0 + q_0)^2}} \right).$$

Using the Feynman parametrization method, we have

$$\int_0^1 dx \frac{[x(1-x)]^{-1/2}}{\pi} \int \frac{d\mathbf{k}_\perp}{(2\pi)^{d-2}} \frac{-2x(1-x)\mathbf{Q}^2}{\tilde{\mathbf{k}}_\perp^2 + (k_0 + xq_0)^2 + x(1-x)\mathbf{Q}^2} = \int_0^1 dx \frac{-2[x(1-x)]^{1/2}\mathbf{Q}^2 \Gamma(\frac{4-d}{2})}{\Pi_2 (4\pi)^{(d-2)/2} \left[ (k_0 + xq_0)^2 + x(1-x)\mathbf{Q}^2 \right]^{\frac{4-d}{2}}},$$

where $\tilde{\mathbf{k}}_\perp = \tilde{\mathbf{k}}_\perp + x\mathbf{q}_\perp$. The integration for $\mathbf{l}_\perp$ is the same with that for $\mathbf{k}_\perp$. Then, we obtain

$$\Pi_2(2) = -\frac{2g^2 \Delta_f \mu^{2\epsilon} |\mathbf{Q}|^4}{(4\pi)^d |q_y|^2} \int_{-\infty}^{\infty} \frac{dk_0}{2\pi} \int_0^1 dx \int_0^1 dy \frac{[x(1-x)]^{1/2}[y(1-y)]^{1/2} \Gamma(\frac{4-d}{2})^2}{\left[ (k_0 + xq_0)^2 + x(1-x)\mathbf{Q}^2 \right]^{\frac{4-d}{2}} \left[ (k_0 + yq_0)^2 + y(1-y)\mathbf{Q}^2 \right]^{\frac{4-d}{2}}}.$$

Using the Feynman parametrization method, we obtain

$$\Pi_2(2) = -\frac{2g^2 \Delta_f \mu^{2\epsilon} |\mathbf{Q}|^4}{(4\pi)^d |q_y|^2} \int_{-\infty}^{\infty} \frac{dk_0}{2\pi} \int_0^1 dx \int_0^1 dy \int_0^1 dz \frac{[z(1-z)]^{(2-d)/2}[x(1-x)]^{1/2}[y(1-y)]^{1/2} \Gamma(4-d)}{\left[ \tilde{k}_0^2 + (zx(1-x) + (1-z)y(1-y))\mathbf{Q}^2 + z(1-z)(x-y)^2 q_0^2 \right]^{4-d}},$$

where $\tilde{k}_0 = k_0 + (zx + (1-z)y)q_0$. Integrating over $k_0$, we obtain

$$\Pi_2(2) = -\frac{2g^2 \Delta_f \mu^{2\epsilon} |\mathbf{Q}|^4}{(4\pi)^{d+1/2} |q_y|^2} \int_0^1 dx \int_0^1 dy \int_0^1 dz \frac{[z(1-z)]^{(2-d)/2}[x(1-x)]^{1/2}[y(1-y)]^{1/2} \Gamma(7/2-d)}{\left[ (zx(1-x) + (1-z)y(1-y))\mathbf{Q}^2 + z(1-z)(x-y)^2 q_0^2 \right]^{7/2-d}}.$$

The momentum factor can be found as $(\mu^{2\epsilon} |\mathbf{Q}|^4 / |q_y|^2) |\mathbf{Q}|^{2d-7} = \mu^{2\epsilon} |\mathbf{Q}|^{2d-3} / |q_y|^2$. The remaining integral can be done to give

$$\int_0^1 dx \int_0^1 dy \int_0^1 dz \frac{[z(1-z)]^{-1/4}[x(1-x)]^{1/2}[y(1-y)]^{1/2}}{zx(1-x) + (1-z)y(1-y)} = 1.644.$$

As a result, we obtain

$$\Pi_2(2) = -g^2 \tilde{\Delta}_f \mu^{2\epsilon} \tilde{B}_d \frac{|\mathbf{Q}|^{2d-3}}{|q_y|^2}, \qquad \tilde{B}_d = 0.05025.$$

### C.1.3 Feynman diagram BS2-3

The boson self-energy in Table 4 BS2-3 is expressed as

$$\Pi_2(3) = g^2\Delta_b\mu^{2\epsilon}\int\frac{d^{d+1}kd^{d+1}l}{(2\pi)^{2d+1}}\delta(k_0+l_0)\text{tr}\big[G_0^*(-k-q)\gamma_{d-1}G_0^*(-k)G_0(l)\gamma_{d-1}G_0(l+q)\big]$$

$$= -2g^2\Delta_b\mu^{2\epsilon}\int\frac{d^{d+1}kd^{d+1}l}{(2\pi)^{2d+1}}\delta(k_0+l_0)\frac{\mathcal{N}}{\mathcal{D}}\,,$$

where $\mathcal{D}$ and $\mathcal{N}$ are given in Eq. (C.1). The integration is the same with $\Pi_2(2)$. As a result, we obtain

$$\Pi_2(3) = -g^2\tilde{\Delta}_b\mu^{\epsilon}\tilde{B}_d\frac{|\mathbf{Q}|^{2d-3}}{|q_y|^2}\,,\qquad \tilde{B}_d = 0.05025\,. \tag{C.4}$$

## C.2 Fermion self-energy

In the two-loop order, there are two kinds of diagrams for fermion self-energy corrections: rainbow diagrams and crossed diagrams. The rainbow diagrams are represented as $\Sigma_r \sim G_0(p+k)G_0(p+l)G_0(p+k)$, where $p$ is external momentum, and $k$ and $l$ are loop momenta. For brevity, gamma matrices and boson propagators have been omitted. Since the loop momenta are "decoupled", the integrations for $k$ and $l$ are separately divergent. As a result, the integral has only a double pole and a simple pole proportional to $\ln p^2$, where the former is irrelevant for renormalization and the latter, called nonlocal divergence, is completely canceled by one loop counterterms. In other words, there is no simple pole, which contributes to the beta functions. We are allowed to drop the rainbow diagrams. From now on, we only focus on the crossed diagrams.

### C.2.1 Feynman diagram FS2-1

The fermion self-energy correction in Table 4 FS2-1 is expressed as

$$\Sigma(1) = \frac{g^4}{N^2}\int\frac{d^{d+1}kd^{d+1}l}{(2\pi)^{2d+2}}\gamma_{d-1}G_0(k+p)\gamma_{d-1}G_0(k+l+p)\gamma_{d-1}G_0(l+p)\gamma_{d-1}D_1(k)D_1(l)$$

$$= \frac{ig^4}{N^2}\int\frac{d^{d+1}kd^{d+1}l}{(2\pi)^{2d+2}}\frac{\mathcal{N}}{\mathcal{D}}D_1(k)D_1(l)\,,$$

where $\mathcal{D}$ and $\mathcal{N}$ are given by

$$\mathcal{D} = \big[(\mathbf{K}+\mathbf{P})^2 + \delta_{\mathbf{k+p}}^2\big]\big[(\mathbf{K}+\mathbf{L}+\mathbf{P})^2 + \delta_{\mathbf{k+l+p}}^2\big]\big[(\mathbf{L}+\mathbf{P})^2 + \delta_{\mathbf{l+p}}^2\big], \tag{C.5a}$$

$$\mathcal{N} = \big[(\mathbf{K}+\mathbf{P})\cdot\mathbf{\Gamma}(\mathbf{K}+\mathbf{L}+\mathbf{P})\cdot\mathbf{\Gamma}(\mathbf{L}+\mathbf{P})\cdot\mathbf{\Gamma} - (\mathbf{K}+\mathbf{P})\cdot\mathbf{\Gamma}\delta_{\mathbf{k+l+p}}\delta_{\mathbf{l+p}} - (\mathbf{K}+\mathbf{L}+\mathbf{P})\cdot\mathbf{\Gamma}\delta_{\mathbf{k+p}}\delta_{\mathbf{l+p}} \tag{C.5b}$$

$$-(\mathbf{L}+\mathbf{P})\cdot\mathbf{\Gamma}\delta_{\mathbf{k+p}}\delta_{\mathbf{k+l+p}}\big] + \gamma_{d-1}\big[-(\mathbf{K}+\mathbf{P})\cdot\mathbf{\Gamma}(\mathbf{K}+\mathbf{L}+\mathbf{P})\cdot\mathbf{\Gamma}\delta_{\mathbf{l+p}} - (\mathbf{K}+\mathbf{L}+\mathbf{P})\cdot\mathbf{\Gamma}(\mathbf{L}+\mathbf{P})\cdot\mathbf{\Gamma}\delta_{\mathbf{k+p}}$$

$$-(\mathbf{K}+\mathbf{P})\cdot\mathbf{\Gamma}(\mathbf{L}+\mathbf{P})\cdot\mathbf{\Gamma}\delta_{\mathbf{k+l+p}} + \delta_{\mathbf{k+p}}\delta_{\mathbf{k+l+p}}\delta_{\mathbf{l+p}}\big].$$

Integrating over $k_x$, we have

$$\Sigma(1) = \frac{ig^4}{N^2}\int\frac{d\mathbf{K}dk_yd^{d+1}l}{(2\pi)^{2d+1}}\frac{\mathcal{N}_1}{\mathcal{D}_1}D_1(k)D_1(l)\,,$$

where $\mathcal{D}_1$ and $\mathcal{N}_1$ are given by

$$\mathcal{D}_1 = 2|\mathbf{K}+\mathbf{P}||\mathbf{K}+\mathbf{L}+\mathbf{P}|\big[(|\mathbf{K}+\mathbf{P}|+|\mathbf{K}+\mathbf{L}+\mathbf{P}|)^2 + (\delta_{l+\mathbf{p}} + 2l_y k_y - \delta_{\mathbf{p}})^2\big]\big[(\mathbf{L}+\mathbf{P})^2 + \delta_{l+\mathbf{p}}^2\big],$$

$$\begin{aligned}
\mathcal{N}_1 = &\Big[(|\mathbf{K}+\mathbf{P}|+|\mathbf{K}+\mathbf{L}+\mathbf{P}|)\big\{(\mathbf{K}+\mathbf{P})\cdot\mathbf{\Gamma}(\mathbf{K}+\mathbf{L}+\mathbf{P})\cdot\mathbf{\Gamma}(\mathbf{L}+\mathbf{P})\cdot\mathbf{\Gamma} - |\mathbf{K}+\mathbf{P}||\mathbf{K}+\mathbf{L}+\mathbf{P}|(\mathbf{L}+\mathbf{P})\cdot\mathbf{\Gamma}\big\} \\
&- (\delta_{l+\mathbf{p}} + 2l_y k_y - \delta_{\mathbf{p}})\delta_{l+\mathbf{p}}\big\{|\mathbf{K}+\mathbf{L}+\mathbf{P}|(\mathbf{K}+\mathbf{P})\cdot\mathbf{\Gamma} - |\mathbf{K}+\mathbf{P}|(\mathbf{K}+\mathbf{L}+\mathbf{P})\cdot\mathbf{\Gamma}\big\}\Big] \\
&+ \gamma_{d-1}\Big[\delta_{l+\mathbf{p}}(|\mathbf{K}+\mathbf{P}|+|\mathbf{K}+\mathbf{L}+\mathbf{P}|)\big\{-(\mathbf{K}+\mathbf{P})\cdot\mathbf{\Gamma}(\mathbf{K}+\mathbf{L}+\mathbf{P})\cdot\mathbf{\Gamma} + |\mathbf{K}+\mathbf{P}||\mathbf{K}+\mathbf{L}+\mathbf{P}|\big\} \\
&- (\delta_{l+\mathbf{p}} + 2l_y k_y - \delta_{\mathbf{p}})\big\{|\mathbf{K}+\mathbf{L}+\mathbf{P}|(\mathbf{K}+\mathbf{P})\cdot\mathbf{\Gamma}(\mathbf{L}+\mathbf{P})\cdot\mathbf{\Gamma} - |\mathbf{K}+\mathbf{P}|(\mathbf{K}+\mathbf{L}+\mathbf{P})\cdot\mathbf{\Gamma}(\mathbf{L}+\mathbf{P})\cdot\mathbf{\Gamma}\big\}\Big].
\end{aligned}$$

Integrating over $l_x$, we obtain

$$\Sigma(1) = \frac{ig^4}{N^2}\int\frac{d\mathbf{K}dk_y d\mathbf{L}dl_y}{(2\pi)^{2d}}\frac{\mathcal{N}_2}{\mathcal{D}_2}D_1(k)D_1(l),$$

where $\mathcal{D}_2$ and $\mathcal{N}_2$ are given by

$$\mathcal{D}_2 = 4|\mathbf{K}+\mathbf{P}||\mathbf{K}+\mathbf{L}+\mathbf{P}||\mathbf{L}+\mathbf{P}|\big[(|\mathbf{K}+\mathbf{P}|+|\mathbf{K}+\mathbf{L}+\mathbf{P}|+|\mathbf{L}+\mathbf{P}|)^2 + (2l_y k_y - \delta_{\mathbf{p}})^2\big],$$

$$\begin{aligned}
\mathcal{N}_2 = &(|\mathbf{K}+\mathbf{P}|+|\mathbf{K}+\mathbf{L}+\mathbf{P}|+|\mathbf{L}+\mathbf{P}|)\Big[(\mathbf{K}+\mathbf{P})\cdot\mathbf{\Gamma}(\mathbf{K}+\mathbf{L}+\mathbf{P})\cdot\mathbf{\Gamma}(\mathbf{L}+\mathbf{P})\cdot\mathbf{\Gamma} \\
&- |\mathbf{K}+\mathbf{P}||\mathbf{K}+\mathbf{L}+\mathbf{P}|(\mathbf{L}+\mathbf{P})\cdot\mathbf{\Gamma} - |\mathbf{K}+\mathbf{L}+\mathbf{P}||\mathbf{L}+\mathbf{P}|(\mathbf{K}+\mathbf{P})\cdot\mathbf{\Gamma} + |\mathbf{K}+\mathbf{P}||\mathbf{L}+\mathbf{P}|(\mathbf{K}+\mathbf{L}+\mathbf{P})\cdot\mathbf{\Gamma}\Big] \\
&+ (2l_y k_y - \delta_{\mathbf{p}})\gamma_{d-1}\Big[|\mathbf{K}+\mathbf{P}|(\mathbf{K}+\mathbf{L}+\mathbf{P})\cdot\mathbf{\Gamma}(\mathbf{L}+\mathbf{P})\cdot\mathbf{\Gamma} - |\mathbf{K}+\mathbf{L}+\mathbf{P}|(\mathbf{K}+\mathbf{P})\cdot\mathbf{\Gamma}(\mathbf{L}+\mathbf{P})\cdot\mathbf{\Gamma} \\
&+ |\mathbf{L}+\mathbf{P}|(\mathbf{K}+\mathbf{P})\cdot\mathbf{\Gamma}(\mathbf{K}+\mathbf{L}+\mathbf{P})\cdot\mathbf{\Gamma} - |\mathbf{L}+\mathbf{P}||\mathbf{K}+\mathbf{P}||\mathbf{K}+\mathbf{L}+\mathbf{P}|\Big].
\end{aligned}$$

We rewrite this expression as $\Sigma(1) = \Sigma_A + \Sigma_B$, where $\Sigma_A$ and $\Sigma_B$ are given by

$$\Sigma_A = \frac{ig^4}{4N^2}\int\frac{d\mathbf{K}dk_y d\mathbf{L}dl_y}{(2\pi)^{2d}}\frac{|K_1|+|K_2|+|K_3|}{(2k_y l_y - \delta_{\mathbf{p}})^2 + (|K_1|+|K_2|+|K_3|)^2}\left[\frac{K_1 K_2 K_3}{|K_1||K_2||K_3|} - \frac{K_1}{|K_1|} + \frac{K_2}{|K_2|} - \frac{K_3}{|K_3|}\right]D_1(k)D_1(l),$$

$$\Sigma_B = \frac{ig^4}{4N^2}\int\frac{d\mathbf{K}dk_y d\mathbf{L}dl_y}{(2\pi)^{2d}}\frac{(2k_y l_y - \delta_{\mathbf{p}})\gamma_{d-1}}{(2k_y l_y - \delta_{\mathbf{p}})^2 + (|K_1|+|K_2|+|K_3|)^2}\left[\frac{K_2 K_3}{|K_2||K_3|} - \frac{K_1 K_3}{|K_1||K_3|} + \frac{K_1 K_2}{|K_1||K_2|} - 1\right]D_1(k)D_1(l),$$

where we introduced simplified notations as

$$K_1 = (\mathbf{K}+\mathbf{P})\cdot\mathbf{\Gamma}, \qquad K_2 = (\mathbf{K}+\mathbf{L}+\mathbf{P})\cdot\mathbf{\Gamma}, \qquad K_3 = (\mathbf{L}+\mathbf{P})\cdot\mathbf{\Gamma}, \tag{C.6a}$$

$$|K_1| = |\mathbf{K}+\mathbf{P}|, \qquad |K_2| = |\mathbf{K}+\mathbf{L}+\mathbf{P}|, \qquad |K_3| = |\mathbf{L}+\mathbf{P}|. \tag{C.6b}$$

We calculate $\Sigma_A$ first. Integrating over $k_y$ and $l_y$, we have

$$\Sigma_A = \frac{ig^{8/3}}{27B_d^{2/3}N^2}\int\frac{d\mathbf{K}d\mathbf{L}}{(2\pi)^{2d-2}}\frac{1}{(|K_1|+|K_2|+|K_3|)(|\mathbf{K}||\mathbf{L}|)^{(d-1)/3}}\left[\frac{K_1 K_2 K_3}{|K_1||K_2||K_3|} - \frac{K_1}{|K_1|} + \frac{K_2}{|K_2|} - \frac{K_3}{|K_3|}\right],$$

where we neglected $(2k_y l_y - \delta_{\mathbf{p}})^2$ because it would give rise to subleading terms in $g$. To find a renormalization factor, we expand $\Sigma_A$ with respect to $\mathbf{P}$ as $\Sigma_A = \Sigma_A^{(0)} + \Sigma_A^{(1)}(i\mathbf{P}\cdot\mathbf{\Gamma}) + \mathcal{O}(\mathbf{P}^2)$. Here, we focus on the term in the integrand, given by

$$\frac{1}{|K_1|+|K_2|+|K_3|}\left[\frac{K_1 K_2 K_3}{|K_1||K_2||K_3|} - \frac{K_1}{|K_1|} + \frac{K_2}{|K_2|} - \frac{K_3}{|K_3|}\right].$$

Setting $\mathbf{P} = 0$, we obtain

$$\frac{1}{|\mathbf{K}|+|\mathbf{K}+\mathbf{L}|+|\mathbf{L}|}\left\{\frac{|\mathbf{L}|^2\mathbf{K}\cdot\mathbf{\Gamma} + |\mathbf{K}|^2\mathbf{L}\cdot\mathbf{\Gamma}}{|\mathbf{K}||\mathbf{K}+\mathbf{L}||\mathbf{L}|} - \frac{\mathbf{K}\cdot\mathbf{\Gamma}}{|\mathbf{K}|} + \frac{(\mathbf{K}+\mathbf{L})\cdot\mathbf{\Gamma}}{|\mathbf{K}+\mathbf{L}|} - \frac{\mathbf{L}\cdot\mathbf{\Gamma}}{|\mathbf{L}|}\right\}.$$

This is odd in $\mathbf{K}$ and $\mathbf{L}$, implying that $\Sigma_A^{(0)}$ would vanish after integrated over $\mathbf{K}$ and $\mathbf{L}$.

In the leading order of $\mathbf{P}$, we find

$$\frac{1}{|\mathbf{K}|+|\mathbf{K}+\mathbf{L}|+|\mathbf{L}|}\left\{\frac{(\mathbf{P}\cdot\boldsymbol{\Gamma})(|\mathbf{K}|^2+|\mathbf{L}|^2+\mathbf{K}\cdot\boldsymbol{\Gamma}\mathbf{L}\cdot\boldsymbol{\Gamma})}{|\mathbf{K}||\mathbf{K}+\mathbf{L}||\mathbf{L}|}-\frac{\mathbf{P}\cdot\boldsymbol{\Gamma}}{|\mathbf{K}|}+\frac{\mathbf{P}\cdot\boldsymbol{\Gamma}}{|\mathbf{K}+\mathbf{L}|}-\frac{\mathbf{P}\cdot\boldsymbol{\Gamma}}{|\mathbf{L}|}\right\}$$

$$-\frac{1}{|\mathbf{K}|+|\mathbf{K}+\mathbf{L}|+|\mathbf{L}|}\left\{\frac{|\mathbf{L}|^2\mathbf{K}\cdot\boldsymbol{\Gamma}+|\mathbf{K}|^2\mathbf{L}\cdot\boldsymbol{\Gamma}}{|\mathbf{K}||\mathbf{K}+\mathbf{L}||\mathbf{L}|}\left(\frac{\mathbf{K}\cdot\mathbf{P}}{|\mathbf{K}|^2}+\frac{(\mathbf{K}+\mathbf{L})\cdot\mathbf{P}}{|\mathbf{K}+\mathbf{L}|^2}+\frac{\mathbf{L}\cdot\mathbf{P}}{|\mathbf{L}|^2}\right)\right.$$

$$\left.-\frac{(\mathbf{K}\cdot\mathbf{P})(\mathbf{K}\cdot\boldsymbol{\Gamma})}{|\mathbf{K}|^3}+\frac{(\mathbf{K}+\mathbf{L})\cdot\mathbf{P}(\mathbf{K}+\mathbf{L})\cdot\boldsymbol{\Gamma}}{|\mathbf{K}+\mathbf{L}|^3}-\frac{(\mathbf{L}\cdot\mathbf{P})(\mathbf{L}\cdot\boldsymbol{\Gamma})}{|\mathbf{L}|^3}\right\}$$

$$-\frac{1}{(|\mathbf{K}|+|\mathbf{K}+\mathbf{L}|+|\mathbf{L}|)^2}\left(\frac{\mathbf{K}\cdot\mathbf{P}}{|\mathbf{K}|}+\frac{(\mathbf{K}+\mathbf{L})\cdot\mathbf{P}}{|\mathbf{K}+\mathbf{L}|}+\frac{\mathbf{L}\cdot\mathbf{P}}{|\mathbf{L}|}\right)$$

$$\times\left\{\frac{|\mathbf{L}|^2\mathbf{K}\cdot\boldsymbol{\Gamma}+|\mathbf{K}|^2\mathbf{L}\cdot\boldsymbol{\Gamma}}{|\mathbf{K}||\mathbf{K}+\mathbf{L}||\mathbf{L}|}-\frac{\mathbf{K}\cdot\boldsymbol{\Gamma}}{|\mathbf{K}|}+\frac{(\mathbf{K}+\mathbf{L})\cdot\boldsymbol{\Gamma}}{|\mathbf{K}+\mathbf{L}|}-\frac{\mathbf{L}\cdot\boldsymbol{\Gamma}}{|\mathbf{L}|}\right\}.$$

We simplify this expression as

$$\frac{(\mathbf{P}\cdot\boldsymbol{\Gamma})}{(d-1)}\left[(d-2)\frac{|\mathbf{K}|^2+|\mathbf{L}|^2+|\mathbf{K}||\mathbf{L}|+\mathbf{K}\cdot\mathbf{L}-(|\mathbf{K}|+|\mathbf{L}|)|\mathbf{K}+\mathbf{L}|}{(|\mathbf{K}|+|\mathbf{L}|+|\mathbf{K}+\mathbf{L}|)|\mathbf{K}||\mathbf{L}||\mathbf{K}+\mathbf{L}|}\right.$$

$$\left.-\frac{|\mathbf{K}|+|\mathbf{L}|-|\mathbf{K}+\mathbf{L}|}{(|\mathbf{K}|+|\mathbf{L}|+|\mathbf{K}+\mathbf{L}|)^2|\mathbf{K}+\mathbf{L}|}\left(1+\frac{\mathbf{K}\cdot\mathbf{L}}{|\mathbf{K}||\mathbf{L}|}\right)-\frac{2|\mathbf{K}||\mathbf{L}|(|\mathbf{K}|+|\mathbf{L}|+2|\mathbf{K}+\mathbf{L}|)}{(|\mathbf{K}|+|\mathbf{L}|+|\mathbf{K}+\mathbf{L}|)^2|\mathbf{K}+\mathbf{L}|^3}\left(1-\frac{(\mathbf{K}\cdot\mathbf{L})^2}{|\mathbf{K}|^2|\mathbf{L}|^2}\right)\right], \quad \text{(C.7)}$$

where we have used the following identities satisfied inside the integral expression

$$(\mathbf{K}\cdot\boldsymbol{\Gamma})(\mathbf{L}\cdot\boldsymbol{\Gamma})=\mathbf{K}\cdot\mathbf{L}, \qquad (\mathbf{K}\cdot\mathbf{P})(\mathbf{K}\cdot\boldsymbol{\Gamma})=\frac{|\mathbf{K}|^2(\mathbf{P}\cdot\boldsymbol{\Gamma})}{(d-1)}, \qquad (\mathbf{L}\cdot\mathbf{P})(\mathbf{L}\cdot\boldsymbol{\Gamma})=\frac{|\mathbf{L}|^2(\mathbf{P}\cdot\boldsymbol{\Gamma})}{(d-1)},$$

$$(\mathbf{K}\cdot\mathbf{P})(\mathbf{L}\cdot\boldsymbol{\Gamma})=\frac{(\mathbf{P}\cdot\boldsymbol{\Gamma})(\mathbf{K}\cdot\mathbf{L})}{(d-1)}, \qquad (\mathbf{L}\cdot\mathbf{P})(\mathbf{K}\cdot\boldsymbol{\Gamma})=\frac{(\mathbf{P}\cdot\boldsymbol{\Gamma})(\mathbf{K}\cdot\mathbf{L})}{(d-1)}.$$

Resorting to Eq. (C.7), we obtain

$$\Sigma_A^{(1)}=\frac{g^{8/3}}{27B_d^{2/3}N^2}\int\frac{d\mathbf{K}d\mathbf{L}}{(2\pi)^{2d-2}}\frac{1}{[|\mathbf{K}||\mathbf{L}|]^{(d-1)/3}}\frac{1}{(d-1)}\left[(d-2)\frac{|\mathbf{K}|^2+|\mathbf{L}|^2+|\mathbf{K}||\mathbf{L}|+\mathbf{K}\cdot\mathbf{L}-(|\mathbf{K}|+|\mathbf{L}|)|\mathbf{K}+\mathbf{L}|}{(|\mathbf{K}|+|\mathbf{L}|+|\mathbf{K}+\mathbf{L}|)|\mathbf{K}||\mathbf{K}+\mathbf{L}||\mathbf{L}|}\right.$$

$$\left.-\frac{|\mathbf{K}|+|\mathbf{L}|-|\mathbf{K}+\mathbf{L}|}{(|\mathbf{K}|+|\mathbf{L}|+|\mathbf{K}+\mathbf{L}|)^2|\mathbf{K}+\mathbf{L}|}\left(1+\frac{\mathbf{K}\cdot\mathbf{L}}{|\mathbf{K}||\mathbf{L}|}\right)-\frac{2|\mathbf{K}||\mathbf{L}|(|\mathbf{K}|+|\mathbf{L}|+2|\mathbf{K}+\mathbf{L}|)}{(|\mathbf{K}|+|\mathbf{L}|+|\mathbf{K}+\mathbf{L}|)^2|\mathbf{K}+\mathbf{L}|^3}\left(1-\frac{(\mathbf{K}\cdot\mathbf{L})^2}{|\mathbf{K}|^2|\mathbf{L}|^2}\right)\right]. \quad \text{(C.8)}$$

Next, we calculate $\Sigma_B$. It gives a renormalization factor for $\delta_{\mathbf{p}}$. To find the renormalization factor, we expand it with respect to $\delta_{\mathbf{p}}$ as $\Sigma_B=\Sigma_B^{(0)}+\Sigma_B^{(1)}(i\delta_{\mathbf{p}}\gamma_{d-1})+\mathcal{O}(\delta_{\mathbf{p}}^2)$. We ignore $\Sigma_B^{(0)}$ because it would vanish after integrated over $k_y$ and $l_y$. Then, we have

$$\Sigma_B^{(1)}=\frac{g^4}{4N^2}\int\frac{d\mathbf{K}dk_yd\mathbf{L}dl_y}{(2\pi)^{2d}}\frac{(2k_yl_y)^2-(|K_1|+|K_2|+|K_3|)^2}{\left[(2k_yl_y)^2+(|K_1|+|K_2|+|K_3|)^2\right]^2}\left[\frac{K_2K_3}{|K_2||K_3|}-\frac{K_1K_3}{|K_1||K_3|}+\frac{K_1K_2}{|K_1||K_2|}-1\right]D_1(k)D_1(l).$$

Integrating over $k_y$ and $l_y$, we obtain

$$\Sigma_B^{(1)}=-\frac{g^{8/3}}{27B_d^{2/3}N^2}\int\frac{d\mathbf{K}d\mathbf{L}}{(2\pi)^{2d-2}}\frac{1}{\left[|\mathbf{K}||\mathbf{L}|\right]^{(d-1)/3}(|K_1|+|K_2|+|K_3|)^2}\left[\frac{K_2K_3}{|K_2||K_3|}-\frac{K_1K_3}{|K_1||K_3|}+\frac{K_1K_2}{|K_1||K_2|}-1\right],$$

where we neglected $(2k_yl_y)^2$ which would give rise to subleading terms in $g$. We set $\mathbf{P}=0$ because the renormalization factor is independent of $\mathbf{P}$. Then, we have

$$\frac{1}{(|\mathbf{K}|+|\mathbf{K}+\mathbf{L}|+|\mathbf{L}|)^2}\left\{\frac{(\mathbf{K}+\mathbf{L})\cdot\boldsymbol{\Gamma}(\mathbf{L}\cdot\boldsymbol{\Gamma})}{|\mathbf{K}+\mathbf{L}||\mathbf{L}|}-\frac{(\mathbf{K}\cdot\boldsymbol{\Gamma})(\mathbf{L}\cdot\boldsymbol{\Gamma})}{|\mathbf{K}||\mathbf{L}|}+\frac{(\mathbf{K}\cdot\boldsymbol{\Gamma})(\mathbf{K}+\mathbf{L})\cdot\boldsymbol{\Gamma}}{|\mathbf{K}||\mathbf{K}+\mathbf{L}|}-1\right\}$$

$$=\frac{|\mathbf{K}|+|\mathbf{L}|-|\mathbf{K}+\mathbf{L}|}{(|\mathbf{K}|+|\mathbf{L}|+|\mathbf{K}+\mathbf{L}|)^2|\mathbf{K}+\mathbf{L}|}\left(1+\frac{\mathbf{K}\cdot\mathbf{L}}{|\mathbf{K}||\mathbf{L}|}\right),$$

where we used $(\mathbf{K} \cdot \boldsymbol{\Gamma})(\mathbf{L} \cdot \boldsymbol{\Gamma}) = \mathbf{K} \cdot \mathbf{L}$ in the second line. As a result, we obtain

$$\Sigma_B^{(1)} = -\frac{g^{8/3}}{27B_d^{2/3}N^2} \int \frac{d\mathbf{K}d\mathbf{L}}{(2\pi)^{2d-2}} \frac{1}{[|\mathbf{K}||\mathbf{L}|]^{(d-1)/3}} \frac{|\mathbf{K}|+|\mathbf{L}|-|\mathbf{K}+\mathbf{L}|}{(|\mathbf{K}|+|\mathbf{L}|+|\mathbf{K}+\mathbf{L}|)^2|\mathbf{K}+\mathbf{L}|}\left(1+\frac{\mathbf{K}\cdot\mathbf{L}}{|\mathbf{K}||\mathbf{L}|}\right). \quad \text{(C.9)}$$

Lastly, we complete the calculation of Eqs. (C.8) and (C.9). Introducing coordinates of $\mathbf{K}\cdot\mathbf{L} = KL\cos\theta$ and changing a variable as $L = Kl$, we have

$$\Sigma_A^{(1)} = \frac{\Omega' g^{8/3}}{27B_d^{2/3}N^2} \int_{|\mathbf{P}|}^\infty dK K^{\frac{4d-13}{3}} \int_0^\infty dl\, l^{\frac{2d-5}{3}} \int_0^\pi d\theta \sin^{d-3}\theta$$
$$\times\left[\frac{(d-2)}{(d-1)}\frac{1+l+l^2+l\cos\theta-(1+l)\eta}{l\,\eta\,(1+l+\eta)} - \frac{(1+l-\eta)(1+\cos\theta)}{(d-1)(1+l+\eta)^2\eta} - \frac{2l(1+l+2\eta)\left(1-\cos^2\theta\right)}{(d-1)(1+l+\eta)^2\eta^3}\right],$$
$$\Sigma_B^{(1)} = -\frac{\Omega' g^{8/3}}{27B_d^{2/3}N^2} \int_{|\mathbf{P}|}^\infty dK K^{\frac{4d-13}{3}} \int_0^\infty dl\, l^{\frac{2d-5}{3}} \int_0^\pi d\theta \sin^{d-3}\theta \frac{(1+l-\eta)(1+\cos\theta)}{\eta(1+l+\eta)^2},$$

where $\Omega' \equiv \frac{4}{(4\pi)^{d-1}\sqrt{\pi}\Gamma(\frac{d-1}{2})\Gamma(\frac{d-2}{2})}$ and $\eta = \sqrt{1+l^2+2l\cos\theta}$. We find an $\epsilon$ pole from the $K$ integral as $\int_{|\mathbf{P}|}^\infty dK K^{\frac{4d-13}{3}} = \frac{3}{4\epsilon} + \mathcal{O}(1)$. The remaining integrals are done as

$$\int_0^\infty dl \int_0^\pi \frac{d\theta}{\sqrt{\sin\theta}}\left[\frac{1}{3}\frac{1+l+l^2+l\cos\theta-(1+l)\eta}{l\,\eta\,(1+l+\eta)} - \frac{2}{3}\frac{(1+l-\eta)(1+\cos\theta)}{(1+l+\eta)^2\eta}\right.$$
$$\left. - \frac{4}{3}\frac{l(1+l+2\eta)\left(1-\cos^2\theta\right)}{(1+l+\eta)^2\eta^3}\right] = \frac{\sqrt{\pi}\Gamma(\frac{d-2}{2})}{\Gamma(\frac{d-1}{2})}(-0.1120),$$
$$\int_0^\infty dl \int_0^\pi \frac{d\theta}{\sqrt{\sin\theta}}\left[\frac{(1+l-\eta)(1+\cos\theta)}{\eta(1+l+\eta)^2}\right] = \frac{\sqrt{\pi}\Gamma(\frac{d-2}{2})}{\Gamma(\frac{d-1}{2})}(0.03770).$$

As a result, we obtain

$$\Sigma(1) = (-0.3361)\frac{\tilde{g}^2}{\epsilon}(i\mathbf{P}\cdot\boldsymbol{\Gamma}) + (-0.1131)\frac{\tilde{g}^2}{\epsilon}(i\delta_{\mathbf{p}}\gamma_{d-1}). \quad \text{(C.10)}$$

### C.2.2 Feynman diagram FS2-2

The fermion self-energy correction in Table 4 FS2-2 is given by

$$\Sigma(2) = \Delta_f^2 \int \frac{d^{d+1}k d^{d+1}l}{(2\pi)^{2d}} \delta(k_0)\delta(l_0)\gamma_{d-1}G_0(k+p)\gamma_{d-1}G_0(k+l+p)\gamma_{d-1}G_0(l+p)\gamma_{d-1}$$
$$= i\Delta_f^2 \int \frac{d^{d+1}k d^{d+1}l}{(2\pi)^{2d}} \delta(k_0)\delta(l_0)\frac{\mathcal{N}}{\mathcal{D}},$$

where $\mathcal{D}$ and $\mathcal{N}$ are given in Eq. (C.5). Integrating over $k_x$ and $l_x$ (the integration is the same with $\Sigma(1)$), we find

$$\Sigma_A = \frac{i\Delta_f^2}{4} \int \frac{d\mathbf{K}dk_y\,d\mathbf{L}dl_y}{(2\pi)^{2d-2}}\delta(k_0)\delta(l_0)\frac{|K_1|+|K_2|+|K_3|}{(2k_yl_y-\delta_{\mathbf{p}})^2+(|K_1|+|K_2|+|K_3|)^2}\left[\frac{K_1K_2K_3}{|K_1||K_2||K_3|}-\frac{K_1}{|K_1|}+\frac{K_2}{|K_2|}-\frac{K_3}{|K_3|}\right],$$
$$\Sigma_B = \frac{i\Delta_f^2}{4} \int \frac{d\mathbf{K}dk_y\,d\mathbf{L}dl_y}{(2\pi)^{2d-2}}\delta(k_0)\delta(l_0)\frac{(2k_yl_y-\delta_{\mathbf{p}})\gamma_{d-1}}{(2k_yl_y-\delta_{\mathbf{p}})^2+(|K_1|+|K_2|+|K_3|)^2}\left[\frac{K_2K_3}{|K_2||K_3|}-\frac{K_1K_3}{|K_1||K_3|}+\frac{K_1K_2}{|K_1||K_2|}-1\right],$$

where $\Sigma(2) = \Sigma_A + \Sigma_B$, and $K_a$, $|K_a|$ with $a = 1, 2, 3$ are given in Eq. (C.6).

$\Sigma_B$ vanishes upon integrating over $(k_y, l_y)$ in the infinite range. Meanwhile, $\Sigma_A$ is divergent under the same integral. The divergence comes from the sections given by $k_y = 0$ and $l_y = 0$.

We regularize the integral by avoiding these sections, i.e. constraining the integral range of $k_y$ as $\int_{-\infty}^{\infty} dk_y \rightarrow \int_{\Lambda}^{\infty} dk_y + \int_{-\infty}^{-\Lambda} dk_y$ and similarly with the $l_y$ integral. We ignore $\delta_{\mathbf{p}}$ for simplicity. Integrating over $(k_y, l_y)$ this way, we obtain

$$\Sigma_A = \frac{i\Delta_f^2}{32\pi^2} \int \frac{d\mathbf{K}d\mathbf{L}}{(2\pi)^{2d-4}} \delta(k_0)\delta(l_0)\left(\text{Im}\left[\text{PolyLog}(2, IA^2)\right] - \pi \ln A\right)\left[\frac{K_1 K_2 K_3}{|K_1||K_2||K_3|} - \frac{K_1}{|K_1|} + \frac{K_2}{|K_2|} - \frac{K_3}{|K_3|}\right],$$

where $\text{PolyLog}[a, b]$ is the polylogarithm function and $A = \frac{\Lambda}{\sqrt{|K_1|+|K_2|+|K_3|}}$. As $A \rightarrow 0$, $\text{Im}\left[\text{PolyLog}(2, IA^2)\right]$ becomes $\text{Im}\left[\text{PolyLog}(2, IA^2)\right] \approx A^2 = \frac{\Lambda^2}{|K_1|+|K_2|+|K_3|}$. The power-counting tells that the contribution arising from $\text{Im}\left[\text{PolyLog}(2, IA^2)\right]$ is only finite due to the additional momentum factor, $|K_1| + |K_2| + |K_3|$, in the denominator. Furthermore, the logarithm term, $-\pi \ln A$, only gives double poles. Thus, we conclude that the epsilon pole is absent in this diagram:

$$\Sigma(2) = 0. \tag{C.11}$$

### C.2.3 Feynman diagram FS2-3

The fermion self-energy correction in Table 4 FS2-3 is

$$\Sigma(3) = \frac{2g^2\Delta_f}{N} \int \frac{d^{d+1}k d^{d+1}l}{(2\pi)^{2d+1}} \delta(l_0)\gamma_{d-1}G_0(k+p)\gamma_{d-1}G_0(k+l+p)\gamma_{d-1}G_0(l+p)\gamma_{d-1}D_1(k)$$

$$= \frac{2ig^2\Delta_f}{N} \int \frac{d^{d+1}k d^{d+1}l}{(2\pi)^{2d+1}} \delta(l_0)\frac{\mathcal{N}}{\mathcal{D}},$$

where $\mathcal{D}$ and $\mathcal{N}$ are given in Eq. (C.5). Integrating over $k_x$ and $l_x$, where the integration is the same with $\Sigma(1)$, we have

$$\Sigma_A = \frac{ig^2\Delta_f}{2N} \int \frac{d\mathbf{K}dk_y d\mathbf{L}dl_y}{(2\pi)^{2d-1}} \delta(l_0)\frac{|K_1|+|K_2|+|K_3|}{(2k_y l_y - \delta_{\mathbf{p}})^2 + (|K_1|+|K_2|+|K_3|)^2}\left[\frac{K_1 K_2 K_3}{|K_1||K_2||K_3|} - \frac{K_1}{|K_1|} - \frac{K_2}{|K_2|} + \frac{K_3}{|K_3|}\right]D_1(k),$$

$$\Sigma_B = \frac{ig^2\Delta_f}{2N} \int \frac{d\mathbf{K}dk_y d\mathbf{L}dl_y}{(2\pi)^{2d-1}} \delta(l_0)\frac{(2k_y l_y - \delta_{\mathbf{p}})\gamma_{d-1}}{(2k_y l_y - \delta_{\mathbf{p}})^2 + (|K_1|+|K_2|+|K_3|)^2}\left[\frac{K_2 K_3}{|K_2||K_3|} + \frac{K_1 K_3}{|K_1||K_3|} - \frac{K_1 K_2}{|K_1||K_2|} - 1\right]D_1(k).$$

Here, $\Sigma(3)$ is decomposed into $\Sigma(3) = \Sigma_A + \Sigma_B$, and $K_a$, $|K_a|$ with $a = 1, 2, 3$ are given in Eq. (C.6). Integrating over $l_y$, we obtain

$$\Sigma_A = \frac{i\Delta_f g^2}{8N} \int \frac{d\mathbf{K}dk_y d\mathbf{L}}{(2\pi)^{2d-2}} \delta(l_0)\frac{1}{|k_y|\left[k_y^2 + g^2 B_d \frac{|\mathbf{K}|^{d-1}}{|k_y|}\right]}\left[\frac{K_1 K_2 K_3}{|K_1||K_2||K_3|} - \frac{K_1}{|K_1|} + \frac{K_2}{|K_2|} - \frac{K_3}{|K_3|}\right],$$

where $\Sigma_B$ vanishes due to the identity of $\int_{-\infty}^{\infty} dx \frac{x^2 - a^2}{(x^2 + a^2)^2} = 0$. Integrating over $k_y$, we obtain $\Sigma_A$ as

$$\Sigma_A = \frac{i\Delta_f g^{2/3}}{12\sqrt{3}B_d^{2/3}N} \int \frac{d\mathbf{K}d\mathbf{L}}{(2\pi)^{2d-3}} \delta(l_0)\frac{1}{|\mathbf{K}|^{2(d-1)/3}}\left[\frac{K_1 K_2 K_3}{|K_1||K_2||K_3|} - \frac{K_1}{|K_1|} + \frac{K_2}{|K_2|} - \frac{K_3}{|K_3|}\right].$$

We expand $\Sigma_A$ with respect to $\mathbf{P}$ as $\Sigma_A = \Sigma_A^{(0)} + \Sigma_{A,1}^{(1)}(ip_0\gamma_0) + \Sigma_{A,2}^{(1)}(i\mathbf{p}_\perp \cdot \gamma_\perp)$. We do the similar thing with $\Sigma_A$ of $\Sigma(1)$, noticing $l_0 = 0$ in this case. Then, we obtain

$$\Sigma_{A,1}^{(1)} = \frac{\Delta_f g^{2/3}}{12\sqrt{3}B_d^{2/3}N} \int \frac{d\mathbf{K}d\mathbf{L}}{(2\pi)^{2d-3}} \frac{\delta(l_0)}{|\mathbf{K}|^{2(d-1)/3}} \left[ \frac{\mathbf{K}^2 + \mathbf{L}^2 + \mathbf{K}\cdot\mathbf{L}}{|\mathbf{K}||\mathbf{K}+\mathbf{L}||\mathbf{L}|} - \frac{1}{|\mathbf{K}|} + \frac{1}{|\mathbf{K}+\mathbf{L}|} - \frac{1}{|\mathbf{L}|} \right.$$
$$\left. -k_0^2\left( \frac{|\mathbf{L}|}{|\mathbf{K}|^3|\mathbf{K}+\mathbf{L}|} + \frac{|\mathbf{L}|}{|\mathbf{K}||\mathbf{K}+\mathbf{L}|^3} - \frac{1}{|\mathbf{K}|^3} + \frac{1}{|\mathbf{K}+\mathbf{L}|^3} \right) \right],$$

$$\Sigma_{A,2}^{(1)} = \frac{\Delta_f g^{2/3}}{12\sqrt{3}B_d^{2/3}N} \int \frac{d\mathbf{K}d\mathbf{L}}{(2\pi)^{2d-3}} \frac{\delta(l_0)}{|\mathbf{K}|^{2(d-1)/3}} \left[ \frac{d-3}{d-2}\left( \frac{\mathbf{K}^2 + \mathbf{L}^2 + \mathbf{K}\cdot\mathbf{L}}{|\mathbf{K}||\mathbf{K}+\mathbf{L}||\mathbf{L}|} - \frac{1}{|\mathbf{K}|} + \frac{1}{|\mathbf{K}+\mathbf{L}|} - \frac{1}{|\mathbf{L}|} \right) \right.$$
$$\left. - \frac{2|\mathbf{K}||\mathbf{L}|}{(d-2)|\mathbf{K}+\mathbf{L}|^3}\left( 1 - \frac{(\mathbf{K}\cdot\mathbf{L})^2}{|\mathbf{K}|^2|\mathbf{L}|^2} \right) \right].$$

Introducing coordinates of $\mathbf{k}_\perp \cdot \mathbf{1}_\perp = kl\cos\theta$ and scaling variables as $l \to kl$ and $k_0 \to kk_0$, we have

$$\Sigma_{A,1}^{(1)} = \frac{\Omega \Delta_f g^{2/3}}{12\pi\sqrt{3}B_d^{2/3}N} \int_{p_0}^\infty dk\, k^{\frac{4d-13}{3}} \int_0^\infty dl\, l^{d-3} \int_0^\infty dk_0 \int_0^\pi d\theta \frac{\sin^{d-4}\theta}{(1+k_0^2)^{(d-1)/3}}$$
$$\times \left[ \left( \frac{1+k_0^2+l^2+l\cos\theta}{l\eta\sqrt{1+k_0^2}} - \frac{1}{\sqrt{1+k_0^2}} + \frac{1}{\eta} - \frac{1}{l} \right) - k_0^2\left( \frac{l}{\eta(1+k_0^2)^{3/2}} + \frac{l}{\eta^3\sqrt{1+k_0^2}} - \frac{1}{(1+k_0^2)^{3/2}} + \frac{1}{\eta^3} \right) \right],$$

$$\Sigma_{A,2}^{(1)} = \frac{\Omega \Delta_f g^{2/3}}{12\pi\sqrt{3}B_d^{2/3}N} \int_{p_0}^\infty dk\, k^{\frac{4d-13}{3}} \int_0^\infty dl\, l^{d-3} \int_0^\infty dk_0 \int_0^\pi d\theta \frac{\sin^{d-4}\theta}{(1+k_0^2)^{(d-1)/3}}$$
$$\times \left[ \frac{d-3}{d-2}\left( \frac{1+k_0^2+l^2+l\cos\theta}{l\eta\sqrt{1+k_0^2}} - \frac{1}{\sqrt{1+k_0^2}} + \frac{1}{\eta} - \frac{1}{l} \right) - \frac{2l\sqrt{1+k_0^2}}{(d-2)\eta^3}\left( 1 - \frac{\cos^2\theta}{1+k_0^2} \right) \right],$$

where $\eta = \sqrt{1+k_0^2+l^2+2l\cos\theta}$. We find an $\epsilon$ pole from the $k$ integral as $\int_{p_0}^\infty dk\, k^{\frac{4d-13}{3}} = \frac{3}{4\epsilon} + \mathcal{O}(1)$. The remaining integrals can be done as

$$\int_0^\infty dl\, l^{d-3} \int_0^\infty dk_0 \int_0^\pi d\theta \frac{\sin^{d-4}\theta}{(1+k_0^2)^{(d-1)/3}} \left[ \left( \frac{1+k_0^2+l^2+l\cos\theta}{l\eta\sqrt{1+k_0^2}} - \frac{1}{\sqrt{1+k_0^2}} + \frac{1}{\eta} - \frac{1}{l} \right) \right.$$
$$\left. -k_0^2\left( \frac{l}{\eta(1+k_0^2)^{3/2}} + \frac{l}{\eta^3\sqrt{1+k_0^2}} - \frac{1}{(1+k_0^2)^{3/2}} + \frac{1}{\eta^3} \right) \right] = \frac{\sqrt{\pi}\Gamma\left(\frac{d-3}{2}\right)}{\Gamma\left(\frac{d-2}{2}\right)}(-0.5290),$$

$$\int_0^\infty dl\, l^{d-3} \int_0^\infty dk_0 \int_0^\pi d\theta \frac{\sin^{d-4}\theta}{(1+k_0^2)^{(d-1)/3}} \left[ \frac{d-3}{d-2}\left( \frac{1+k_0^2+l^2+l\cos\theta}{l\eta\sqrt{1+k_0^2}} - \frac{1}{\sqrt{1+k_0^2}} + \frac{1}{\eta} - \frac{1}{l} \right) \right.$$
$$\left. - \frac{2l\sqrt{1+k_0^2}}{(d-2)\eta^3}\left( 1 - \frac{\cos^2\theta}{1+k_0^2} \right) \right] = \frac{\sqrt{\pi}\Gamma\left(\frac{d-3}{2}\right)}{\Gamma\left(\frac{d-2}{2}\right)}(-18.68).$$

As a result, we obtain

$$\Sigma(3) = (-0.4461)\frac{\tilde{\Delta}_f \sqrt{\tilde{g}}}{\sqrt{N}\epsilon}(ip_0\gamma_0) + (-15.75)\frac{\tilde{\Delta}_f \sqrt{\tilde{g}}}{\sqrt{N}\epsilon}(i\mathbf{p}_\perp \cdot \gamma_\perp). \tag{C.12}$$

### C.2.4 Feynman diagram FS2-4

The fermion self-energy correction in Table 4 FS2-4 is

$$\Sigma(4) = 2\Delta_f\Delta_b\int\frac{d^{d+1}kd^{d+1}l}{(2\pi)^{2d}}\delta(k_0)\delta(l_0)\gamma_{d-1}G_0(k+p)G_0^*(-k-l-p)\gamma_{d-1}G_0^*(-l-p)$$

$$= 2i\Delta_f\Delta_b\int\frac{d^dkd^dl}{(2\pi)^{2d}}\frac{\mathcal{N}}{\mathcal{D}},$$

where $\mathcal{D}$ and $\mathcal{N}$ are given by

$$\mathcal{D} = \left[(\mathbf{K}+\mathbf{P})^2 + \delta_{\mathbf{k}+\mathbf{p}}^2\right]\left[(\mathbf{K}+\mathbf{L}+\mathbf{P})^2 + \delta_{-\mathbf{k}-\mathbf{l}-\mathbf{p}}^2\right]\left[(\mathbf{L}+\mathbf{P})^2 + \delta_{-\mathbf{l}-\mathbf{p}}^2\right], \tag{C.13a}$$

$$\mathcal{N} = \left[(\mathbf{K}+\mathbf{P})\cdot\mathbf{\Gamma}(\mathbf{K}+\mathbf{L}+\mathbf{P})\cdot\mathbf{\Gamma}(\mathbf{L}+\mathbf{P})\cdot\mathbf{\Gamma} - (\mathbf{K}+\mathbf{P})\cdot\mathbf{\Gamma}\delta_{-\mathbf{k}-\mathbf{l}-\mathbf{p}}\delta_{-\mathbf{l}-\mathbf{p}} - (\mathbf{K}+\mathbf{L}+\mathbf{P})\cdot\mathbf{\Gamma}\delta_{\mathbf{k}+\mathbf{p}}\delta_{-\mathbf{l}-\mathbf{p}}\right. \tag{C.13b}$$

$$\left.-(\mathbf{L}+\mathbf{P})\cdot\mathbf{\Gamma}\delta_{\mathbf{k}+\mathbf{p}}\delta_{-\mathbf{k}-\mathbf{l}-\mathbf{p}}\right] + \gamma_{d-1}\left[-(\mathbf{K}+\mathbf{P})\cdot\mathbf{\Gamma}(\mathbf{K}+\mathbf{L}+\mathbf{P})\cdot\mathbf{\Gamma}\delta_{-\mathbf{l}-\mathbf{p}} - (\mathbf{K}+\mathbf{L}+\mathbf{P})\cdot\mathbf{\Gamma}(\mathbf{L}+\mathbf{P})\cdot\mathbf{\Gamma}\delta_{\mathbf{k}+\mathbf{p}}\right.$$

$$\left.-(\mathbf{K}+\mathbf{P})\cdot\mathbf{\Gamma}(\mathbf{L}+\mathbf{P})\cdot\mathbf{\Gamma}\delta_{-\mathbf{k}-\mathbf{l}-\mathbf{p}} + \delta_{\mathbf{k}+\mathbf{p}}\delta_{-\mathbf{k}-\mathbf{l}-\mathbf{p}}\delta_{-\mathbf{l}-\mathbf{p}}\right].$$

Integrating over $k_x$ and $l_x$, where the integration is similar with $\Sigma(1)$, we have

$$\Sigma_A = \frac{i\Delta_f\Delta_b}{2}\int\frac{d\mathbf{K}dk_ydLdl_y}{(2\pi)^{2d-2}}\delta(k_0)\delta(l_0)\frac{|K_1|+|K_2|+|K_3|}{(-2k_y\tilde{l}_d - \delta_{\mathbf{p}})^2 + (|K_1|+|K_2|+|K_3|)^2}\left[\frac{K_1K_2K_3}{|K_1||K_2||K_3|} - \frac{K_1}{|K_1|} - \frac{K_2}{|K_2|} + \frac{K_3}{|K_3|}\right],$$

$$\Sigma_B = \frac{i\Delta_f\Delta_b}{2}\int\frac{d\mathbf{K}dk_ydLdl_y}{(2\pi)^{2d-2}}\delta(k_0)\delta(l_0)\frac{(-2k_y\tilde{l}_d - \delta_{\mathbf{p}})\gamma_{d-1}}{(-2k_y\tilde{l}_d - \delta_{\mathbf{p}})^2 + (|K_1|+|K_2|+|K_3|)^2}\left[\frac{K_2K_3}{|K_2||K_3|} + \frac{K_1K_3}{|K_1||K_3|} - \frac{K_1K_2}{|K_1||K_2|} - 1\right].$$

Here, $\Sigma(4)$ is decomposed into $\Sigma(4) = \Sigma_A + \Sigma_B$ with $\tilde{l}_d = l_y + k_y + p_y$. $K_a$ and $|K_a|$ with $a = 1, 2, 3$ are given in Eq. (C.6). There are some differences between these expressions and those of $\Sigma(2)$, where $-2k_y\tilde{l}_d - \delta_{\mathbf{p}}$ appears instead of $2k_yl_y - \delta_{\mathbf{p}}$ and some terms in the brackets differ in sign. However, these differences can be eliminated with variable changes, given by $l_y \to l_y - k_y - p_y$, $k_y \to -k_y$, $\mathbf{k}_\perp \to \mathbf{k}_\perp$, and $\mathbf{l}_\perp \to \mathbf{l}_\perp + \mathbf{k}_\perp$. As a result, we obtain

$$\Sigma(4) = 0. \tag{C.14}$$

### C.2.5 Feynman diagram FS2-5

The fermion self-energy correction in Table 4 FS2-5 is

$$\Sigma(5) = 2g^2\Delta_b\int\frac{d^{d+1}kd^{d+1}l}{(2\pi)^{2d+1}}\delta(k_0)\gamma_{d-1}G_0(k+p) - G_0^*(-k-l-p)\gamma_{d-1}G_0^*(-l-p)D_1(k)$$

$$= 2ig^2\Delta_b\int\frac{d^{d+1}kd^{d+1}l}{(2\pi)^{2d+1}}\delta(l_0)\frac{\mathcal{N}}{\mathcal{D}},$$

where $\mathcal{D}$ and $\mathcal{N}$ are given in Eq. (C.13). Integrating over $k_x$ and $l_x$, where the integration is similar with $\Sigma(1)$, we have

$$\Sigma_A = \frac{ig^2\Delta_b}{2}\int\frac{d\mathbf{K}dk_ydLdl_y}{(2\pi)^{2d-1}}\delta(l_0)\frac{|K_1|+|K_2|+|K_3|}{(-2k_y\tilde{l}_d - \delta_{\mathbf{p}})^2 + (|K_1|+|K_2|+|K_3|)^2}\left[\frac{K_1K_2K_3}{|K_1||K_2||K_3|} - \frac{K_1}{|K_1|} - \frac{K_2}{|K_2|} + \frac{K_3}{|K_3|}\right]D_1(k),$$

$$\Sigma_B = \frac{ig^2\Delta_b}{2}\int\frac{d\mathbf{K}dk_ydLdl_y}{(2\pi)^{2d-1}}\delta(l_0)\frac{(-2k_y\tilde{l}_d - \delta_{\mathbf{p}})\gamma_{d-1}}{(-2k_y\tilde{l}_d - \delta_{\mathbf{p}})^2 + (|K_1|+|K_2|+|K_3|)^2}\left[\frac{K_2K_3}{|K_2||K_3|} + \frac{K_1K_3}{|K_1||K_3|} - \frac{K_1K_2}{|K_1||K_2|} - 1\right]D_1(k).$$

Here, $\Sigma(5)$ is decomposed into $\Sigma(5) = \Sigma_A + \Sigma_B$ with $\tilde{l}_y = l_y + k_y + p_y$. $K_a$ and $|K_a|$ with $a = 1, 2, 3$ are given in Eq. (C.6). Resorting to the following change of variables as $l_y \to l_y - k_y - p_y$, $k_y \to -k_y$, $\mathbf{k}_\perp \to \mathbf{k}_\perp$, and $\mathbf{l}_\perp \to \mathbf{l}_\perp + \mathbf{k}_\perp$, we find the same expression as $\Sigma(3)$. As a result, we obtain

$$\Sigma(5) = (-0.4461)\frac{\tilde{\Delta}_b\sqrt{\tilde{g}}}{\sqrt{N}\epsilon}(ip_0\gamma_0) + (-15.75)\frac{\tilde{\Delta}_b\sqrt{\tilde{g}}}{\sqrt{N}\epsilon}(i\mathbf{p}_\perp\cdot\gamma_\perp). \tag{C.15}$$

Table 5: Feynman diagrams for two-loop vertex corrections for the forward disorder scattering $\Delta_f$. Here, $A_{\Delta_f}$ represents the coefficient of the $\epsilon$ poles computed from the corresponding Feynman diagrams.

| Diagram No. | FV2-1 | FV2-2 | FV2-3 | FV2-4 | FV2-5 | FV2-6 |
|---|---|---|---|---|---|---|
| Feynman Diagram | | | | | | |
| Renormalization factors | $A_{\Delta_f}=0$ | $A_{\Delta_f}=0$ | $A_{\Delta_f}=0$ | $A_{\Delta_f}=0$ | $A_{\Delta_f}=0$ | $A_{\Delta_f}=0$ |
| Diagram No. | FV2-7 | FV2-8 | FV2-9 | FV2-10 | FV2-11 | FV2-12 |
| Feynman Diagram | | | | | | |
| Renormalization factors | $A_{\Delta_f}=0$ | $A_{\Delta_f}=0$ | $A_{\Delta_f}=0$ | $A_{\Delta_f}=1.795\tilde{\Delta}_f\tilde{g}$ | $A_{\Delta_f}=-4.162\tilde{\Delta}_f\sqrt{\frac{\tilde{g}}{N}}$ | $A_{\Delta_f}=-4.162\frac{\tilde{\Delta}_b^2}{\tilde{\Delta}_f}\sqrt{\frac{\tilde{g}}{N}}$ |
| Diagram No. | FV2-13 | FV2-14 | FV2-15 | FV2-16 | FV2-17 | FV2-18 |
| Feynman Diagram | | | | | | |
| Renormalization factors | $A_{\Delta_f}=0$ | $A_{\Delta_f}=0$ | $A_{\Delta_f}=0$ | $A_{\Delta_f}=0$ | $A_{\Delta_f}=0$ | $A_{\Delta_f}=0$ |
| Diagram No. | FV2-19 | FV2-20 | FV2-21 | | | |
| Feynman Diagram | | | | | | |
| Renormalization factors | $A_{\Delta_f}=0.2765\tilde{\Delta}_f\tilde{g}$ | $A_{\Delta_f}=0$ | $A_{\Delta_f}=0$ | | | |

# D  Two-loop vertex corrections

## D.1  Forward scattering

### D.1.1  Feynman diagram FV2-1

The vertex correction in Table 5 FV2-1 is given by

$$\mathcal{M}(1)=-2\Delta_f^3\int\frac{d^{d+1}k\,d^{d+1}l}{(2\pi)^{2d}}\delta(k_0)\delta(l_0)\gamma_{d-1}G_0(k+l+p_1)\gamma_{d-1}G_0(l+p_1)\gamma_{d-1}$$
$$\otimes\gamma_{d-1}G_0(k+p_2)\gamma_{d-1}G_0(-l+p_2)\gamma_{d-1}\,.$$

Using Eq. (B.1), we find the renormalization factor $\delta\Delta_f(1)$ as

$$\delta\Delta_f(1)=-2\Delta_f^3\int\frac{d^dk\,d^dl}{(2\pi)^{2d}}\frac{\left[\delta_{\mathbf{k+l}}\delta_{\mathbf{l}}-(\mathbf{k}_\perp+\mathbf{l}_\perp)\cdot\mathbf{l}_\perp\right]\left[\delta_{\mathbf{k}}\delta_{-\mathbf{l}}+\mathbf{k}_\perp\cdot\mathbf{l}_\perp\right]}{\left[\delta_{\mathbf{k+l}}^2+(\mathbf{k}_\perp+\mathbf{l}_\perp)^2\right]\left[\delta_{\mathbf{k}}^2+\mathbf{k}_\perp^2\right]\left[\delta_{\mathbf{l}}^2+\mathbf{l}_\perp^2\right]\left[\delta_{-\mathbf{l}}^2+\mathbf{l}_\perp^2\right]}\,.$$

Table 6: Feynman diagrams for two-loop vertex corrections for the backward disorder scattering $\Delta_b$. Here, $A_{\Delta_b}$ represents the coefficient of the $\epsilon$ poles computed from the corresponding Feynman diagrams.

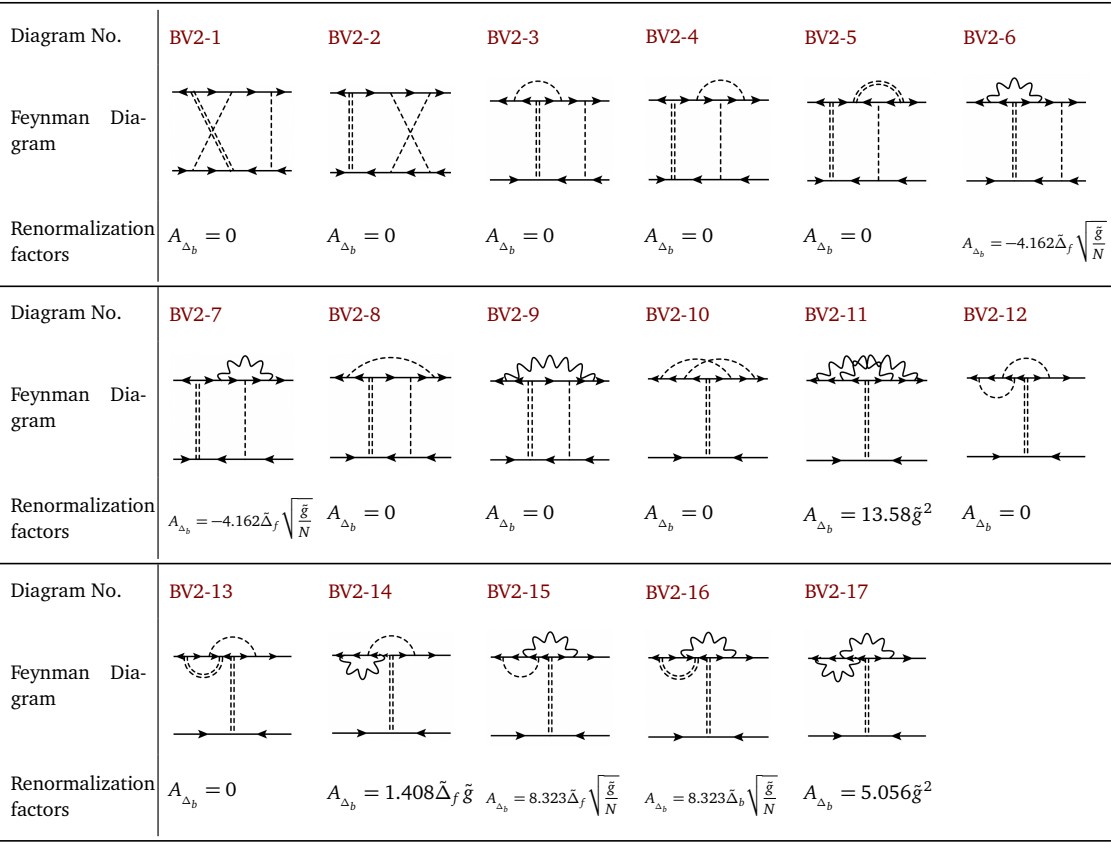

| Diagram No. | BV2-1 | BV2-2 | BV2-3 | BV2-4 | BV2-5 | BV2-6 |
|---|---|---|---|---|---|---|
| Feynman Diagram | | | | | | |
| Renormalization factors | $A_{\Delta_b}=0$ | $A_{\Delta_b}=0$ | $A_{\Delta_b}=0$ | $A_{\Delta_b}=0$ | $A_{\Delta_b}=0$ | $A_{\Delta_b}=-4.162\tilde{\Delta}_f\sqrt{\frac{\tilde{g}}{N}}$ |
| Diagram No. | BV2-7 | BV2-8 | BV2-9 | BV2-10 | BV2-11 | BV2-12 |
| Feynman Diagram | | | | | | |
| Renormalization factors | $A_{\Delta_b}=-4.162\tilde{\Delta}_f\sqrt{\frac{\tilde{g}}{N}}$ | $A_{\Delta_b}=0$ | $A_{\Delta_b}=0$ | $A_{\Delta_b}=0$ | $A_{\Delta_b}=13.58\tilde{g}^2$ | $A_{\Delta_b}=0$ |
| Diagram No. | BV2-13 | BV2-14 | BV2-15 | BV2-16 | BV2-17 | |
| Feynman Diagram | | | | | | |
| Renormalization factors | $A_{\Delta_b}=0$ | $A_{\Delta_b}=1.408\tilde{\Delta}_f\tilde{g}$ | $A_{\Delta_b}=8.323\tilde{\Delta}_f\sqrt{\frac{\tilde{g}}{N}}$ | $A_{\Delta_b}=8.323\tilde{\Delta}_b\sqrt{\frac{\tilde{g}}{N}}$ | $A_{\Delta_b}=5.056\tilde{g}^2$ | |

Integrating over $k_x$, we have

$$\delta\Delta_f(1)=-\Delta_f^3\int\frac{d\mathbf{k}_\perp dk_y d^d l}{(2\pi)^{2d-1}}\frac{(|\mathbf{k}_\perp+\mathbf{1}_\perp|+|\mathbf{k}_\perp|)\left(\delta_1\delta_{-1}-\frac{(\mathbf{k}_\perp+\mathbf{1}_\perp)\cdot\mathbf{1}_\perp\mathbf{k}_\perp\cdot\mathbf{1}_\perp}{|\mathbf{k}_\perp+\mathbf{1}_\perp||\mathbf{k}_\perp|}\right)+(\delta_1+2k_yl_y)\left(\delta_{-1}\frac{(\mathbf{k}_\perp+\mathbf{1}_\perp)\cdot\mathbf{1}_\perp}{|\mathbf{k}_\perp+\mathbf{1}_\perp|}+\delta_1\frac{\mathbf{k}_\perp\cdot\mathbf{1}_\perp}{|\mathbf{k}_\perp|}\right)}{\left[(\delta_1+2k_yl_y)^2+(|\mathbf{k}_\perp+\mathbf{1}_\perp|+|\mathbf{k}_\perp|)^2\right]\left[\delta_1^2+\mathbf{1}_\perp^2\right]\left[\delta_{-1}^2+\mathbf{1}_\perp^2\right]}.$$

Integrating over $k_y$, we obtain

$$\delta\Delta_f(1)=-\frac{\Delta_f^3}{4}\int\frac{d\mathbf{k}_\perp d\mathbf{1}_\perp dl_x dl_y}{(2\pi)^{2d-2}}\frac{1}{|l_y|\left[\delta_1^2+\mathbf{1}_\perp^2\right]\left[\delta_{-1}^2+\mathbf{1}_\perp^2\right]}\left(\delta_1\delta_{-1}-\frac{(\mathbf{k}_\perp+\mathbf{1}_\perp)\cdot\mathbf{1}_\perp\mathbf{k}_\perp\cdot\mathbf{1}_\perp}{|\mathbf{k}_\perp+\mathbf{1}_\perp||\mathbf{k}_\perp|}\right).$$

Integrating over $l_x$, we have

$$\delta\Delta_f(1)=\frac{\Delta_f^3}{16}\int\frac{d\mathbf{k}_\perp d\mathbf{1}_\perp dl_y}{(2\pi)^{2d-3}}\frac{|\mathbf{1}_\perp|}{|l_y|\left[l_y^4+\mathbf{1}_\perp^2\right]}\left(1+\frac{(\mathbf{k}_\perp+\mathbf{1}_\perp)\cdot\mathbf{1}_\perp\mathbf{k}_\perp\cdot\mathbf{1}_\perp}{|\mathbf{k}_\perp+\mathbf{1}_\perp||\mathbf{1}_\perp||\mathbf{k}_\perp||\mathbf{1}_\perp|}\right).$$

The integral for $l_y$ is divergent near $l_y=0$. We regularize this integral with a cutoff $\Lambda$ as

$$\delta\Delta_f(1)=\frac{\Delta_f^3}{16}\int\frac{d\mathbf{k}_\perp d\mathbf{1}_\perp}{(2\pi)^{2d-4}}\int_\Lambda^\infty\frac{dl_y}{2\pi}\frac{2|\mathbf{1}_\perp|}{|l_y|\left[l_y^4+\mathbf{1}_\perp^2\right]}\left(1+\frac{(\mathbf{k}_\perp+\mathbf{1}_\perp)\cdot\mathbf{1}_\perp\mathbf{k}_\perp\cdot\mathbf{1}_\perp}{|\mathbf{k}_\perp+\mathbf{1}_\perp||\mathbf{1}_\perp||\mathbf{k}_\perp||\mathbf{1}_\perp|}\right)$$

$$=\frac{\Delta_f^3}{64\pi}\int\frac{d\mathbf{k}_\perp d\mathbf{1}_\perp}{(2\pi)^{2d-4}}\frac{\ln\left(1+\mathbf{1}_\perp^2/\Lambda^4\right)}{|\mathbf{1}_\perp|}\left(1+\frac{(\mathbf{k}_\perp+\mathbf{1}_\perp)\cdot\mathbf{1}_\perp\mathbf{k}_\perp\cdot\mathbf{1}_\perp}{|\mathbf{k}_\perp+\mathbf{1}_\perp||\mathbf{1}_\perp||\mathbf{k}_\perp||\mathbf{1}_\perp|}\right).$$

Introducing coordinates of $\mathbf{k}_\perp \cdot \mathbf{1}_\perp = kl\cos\theta$ and scaling variables as $k \to lk$, we have

$$\delta\Delta_f(1) = \frac{\Omega\Delta_f^3}{64\pi}\int_{p_0}^\infty dl\, l^{2d-6}\ln(l^2/\Lambda^4 + 1)\int_0^\infty dk\int_0^\pi d\theta \sin^{d-4}\theta\left(1 + \frac{\cos\theta(k\cos\theta + 1)}{k\sqrt{1 + 2k\cos\theta + k^2}}\right).$$

The integral for $l$ gives

$$\int_{p_0}^\infty dl\, l^{2d-6}\ln(l^2/\Lambda^4 + 1) = \frac{1}{2\epsilon^2} - \frac{\ln(\Lambda^4/p_0^2)}{2\epsilon} + \mathcal{O}(1).$$

The logarithmic term, $-\frac{\ln(\Lambda^4/p_0^2)}{2\epsilon}$, would be cancelled to the counterterm diagram associated with the one-loop counterterm. As a result, we conclude

$$\delta\Delta_f(1) = 0.$$

### D.1.2 Feynman diagram FV2-2

From the vertex correction in Table 5 FV2-2, we find a renormalization factor as

$$\delta\Delta_f(2) = -2\Delta_f^3\int\frac{d^d k\, d^d l}{(2\pi)^{2d}}\frac{\left[\delta_{\mathbf{k+l}}\delta_{\mathbf{l}} - (\mathbf{k}_\perp + \mathbf{1}_\perp)\cdot\mathbf{1}_\perp\right]\left[\delta_{-\mathbf{k}}\delta_{\mathbf{l}} + \mathbf{k}_\perp\cdot\mathbf{1}_\perp\right]}{\left[\delta_{\mathbf{k+l}}^2 + (\mathbf{k}_\perp + \mathbf{1}_\perp)^2\right]\left[\delta_{-\mathbf{k}}^2 + \mathbf{k}_\perp^2\right]\left[\delta_{\mathbf{l}}^2 + \mathbf{1}_\perp^2\right]^2}.$$

Integrating over $k_x$, we have

$$\delta\Delta_f(2) = \Delta_f^3\int\frac{d\mathbf{k}_\perp dk_y d^d l}{(2\pi)^{2d-1}}\frac{(|\mathbf{k}_\perp + \mathbf{1}_\perp| + |\mathbf{k}_\perp|)\left(\delta_{\mathbf{l}}^2 + \frac{(\mathbf{k}_\perp+\mathbf{1}_\perp)\cdot\mathbf{1}_\perp\,\mathbf{k}_\perp\cdot\mathbf{1}_\perp}{|\mathbf{k}_\perp+\mathbf{1}_\perp||\mathbf{k}_\perp|}\right) + (\delta_{\mathbf{l}} + 2k_y l_y + 2k_y^2)\delta_{\mathbf{l}}\left(\frac{(\mathbf{k}_\perp+\mathbf{1}_\perp)\cdot\mathbf{1}_\perp}{|\mathbf{k}_\perp+\mathbf{1}_\perp|} - \frac{\mathbf{k}_\perp\cdot\mathbf{1}_\perp}{|\mathbf{k}_\perp|}\right)}{\left[(\delta_{\mathbf{l}} + 2k_y l_y + 2k_y^2)^2 + (|\mathbf{k}_\perp + \mathbf{1}_\perp| + |\mathbf{k}_\perp|)^2\right]\left[\delta_{\mathbf{l}}^2 + \mathbf{1}_\perp^2\right]^2}.$$

Integrating over $l_x$, we obtain

$$\delta\Delta_f(2) = \frac{\Delta_f^3}{4}\int\frac{d\mathbf{k}_\perp dk_y d\mathbf{1}_\perp dl_y}{(2\pi)^{2d-2}}\frac{(2k_y l_y + 2k_y^2)^2 - (|\mathbf{k}_\perp| + |\mathbf{k}_\perp + \mathbf{1}_\perp| + |\mathbf{1}_\perp|)^2}{\left[(2k_y l_y + 2k_y^2)^2 + (|\mathbf{k}_\perp| + |\mathbf{k}_\perp + \mathbf{1}_\perp| + |\mathbf{1}_\perp|)^2\right]^2}\left(1 - \frac{(\mathbf{k}_\perp + \mathbf{1}_\perp)\cdot\mathbf{1}_\perp}{|\mathbf{k}_\perp + \mathbf{1}_\perp||\mathbf{1}_\perp|}\right)\left(1 + \frac{\mathbf{k}_\perp\cdot\mathbf{1}_\perp}{|\mathbf{k}_\perp||\mathbf{1}_\perp|}\right).$$

Introducing the coordinates $k_y = r\cos\theta$ and $l_y = \sin\theta$, we rewrite the integral for $k_y$ and $l_y$ as

$$\delta\Delta_f(2) = \frac{\Delta_f^3}{4}\int\frac{d\mathbf{k}_\perp d\mathbf{1}_\perp}{(2\pi)^{2d-4}}\int_0^{2\pi}\frac{d\theta}{2\pi}\int_0^\infty\frac{dr\, r}{2\pi}\frac{4r^4(\cos\theta\sin\theta + \cos^2\theta)^2 - (|\mathbf{k}_\perp| + |\mathbf{k}_\perp + \mathbf{1}_\perp| + |\mathbf{1}_\perp|)^2}{\left[4r^4(\cos\theta\sin\theta + \cos^2\theta)^2 + (|\mathbf{k}_\perp| + |\mathbf{k}_\perp + \mathbf{1}_\perp| + |\mathbf{1}_\perp|)^2\right]^2}$$
$$\times\left(1 - \frac{(\mathbf{k}_\perp + \mathbf{1}_\perp)\cdot\mathbf{1}_\perp}{|\mathbf{k}_\perp + \mathbf{1}_\perp||\mathbf{1}_\perp|}\right)\left(1 + \frac{\mathbf{k}_\perp\cdot\mathbf{1}_\perp}{|\mathbf{k}_\perp||\mathbf{1}_\perp|}\right).$$

Integrated over $r$, this vanishes. As a result, we obtain

$$\delta\Delta_f(2) = 0.$$

### D.1.3 Feynman diagram FV2-3

From the vertex correction in Table 5 FV2-3, we find a renormalization factor as

$$\delta\Delta_f(3) = -\Delta_f\Delta_b^2\int\frac{d^d k\, d^d l}{(2\pi)^{2d}}\frac{\left[\delta_{\mathbf{k+l}}\delta_{-\mathbf{l}} - (\mathbf{k}_\perp + \mathbf{1}_\perp)\cdot\mathbf{1}_\perp\right]\left[\delta_{\mathbf{k}}\delta_{\mathbf{l}} + \mathbf{k}_\perp\cdot\mathbf{1}_\perp\right]}{\left[\delta_{\mathbf{k+l}}^2 + (\mathbf{k}_\perp + \mathbf{1}_\perp)^2\right]\left[\delta_{\mathbf{k}}^2 + \mathbf{k}_\perp^2\right]\left[\delta_{\mathbf{l}}^2 + \mathbf{1}_\perp^2\right]\left[\delta_{-\mathbf{l}}^2 + \mathbf{1}_\perp^2\right]}.$$

Integrating over $k_x$, we have

$$\delta\Delta_f(3) = -\Delta_f\Delta_b^2\int\frac{d\mathbf{k}_\perp dk_y d^d l}{(2\pi)^{2d-1}}\frac{(|\mathbf{k}_\perp + \mathbf{1}_\perp| + |\mathbf{k}_\perp|)\left(\delta_{\mathbf{l}}\delta_{-\mathbf{l}} - \frac{(\mathbf{k}_\perp+\mathbf{1}_\perp)\cdot\mathbf{1}_\perp\,\mathbf{k}_\perp\cdot\mathbf{1}_\perp}{|\mathbf{k}_\perp+\mathbf{1}_\perp||\mathbf{k}_\perp|}\right) + (\delta_{\mathbf{l}} + 2k_y l_y)\left(\delta_{\mathbf{l}}\frac{(\mathbf{k}_\perp+\mathbf{1}_\perp)\cdot\mathbf{1}_\perp}{|\mathbf{k}_\perp+\mathbf{1}_\perp|} + \delta_{-\mathbf{l}}\frac{\mathbf{k}_\perp\cdot\mathbf{1}_\perp}{|\mathbf{k}_\perp|}\right)}{\left[(\delta_{\mathbf{l}} + 2k_y l_y)^2 + (|\mathbf{k}_\perp + \mathbf{1}_\perp| + |\mathbf{k}_\perp|)^2\right]\left[\delta_{\mathbf{l}}^2 + \mathbf{1}_\perp^2\right]\left[\delta_{-\mathbf{l}}^2 + \mathbf{1}_\perp^2\right]}.$$

Integrating over $k_y$, we obtain

$$\delta\Delta_f(3) = -\frac{\Delta_f\Delta_b^2}{4}\int\frac{d\mathbf{k}_\perp d^d l}{(2\pi)^{2d-2}}\frac{1}{|l_y|\big[\delta_1^2 + \mathbf{l}_\perp^2\big]\big[\delta_{-1}^2 + \mathbf{l}_\perp^2\big]}\Big(\delta_1\delta_{-1} - \frac{(\mathbf{k}_\perp + \mathbf{1}_\perp)\cdot\mathbf{1}_\perp\mathbf{k}_\perp\cdot\mathbf{1}_\perp}{|\mathbf{k}_\perp + \mathbf{1}_\perp||\mathbf{k}_\perp|}\Big).$$

This is the same with $\delta\Delta_f(1)$. As a result, we obtain

$$\delta\Delta_f(3) = 0. \tag{D.1}$$

### D.1.4 Feynman diagram FV2-4

From the vertex correction in Table 5 FV2-4, we find a renormalization factor as

$$\delta\Delta_f(4) = -4\Delta_f\Delta_b^2\int\frac{d^d k d^d l}{(2\pi)^{2d}}\frac{\big[\delta_{\mathbf{k+1}}\delta_\mathbf{l} - (\mathbf{k}_\perp + \mathbf{1}_\perp)\cdot\mathbf{1}_\perp\big]\big[\delta_\mathbf{k}\delta_\mathbf{l} + \mathbf{k}_\perp\cdot\mathbf{1}_\perp\big]}{\big[\delta_{\mathbf{k+1}}^2 + (\mathbf{k}_\perp + \mathbf{1}_\perp)^2\big]\big[\delta_\mathbf{k}^2 + \mathbf{k}_\perp^2\big]\big[\delta_\mathbf{l}^2 + \mathbf{l}_\perp^2\big]^2}.$$

Integrating over $k_x$, we obtain

$$\delta\Delta_f(4) = -2\Delta_f\Delta_b^2\int\frac{d\mathbf{k}_\perp dk_y d^d l}{(2\pi)^{2d-1}}\frac{(|\mathbf{k}_\perp + \mathbf{1}_\perp| + |\mathbf{k}_\perp|)\big(\delta_1^2 - \frac{(\mathbf{k}_\perp + \mathbf{1}_\perp)\cdot\mathbf{1}_\perp\mathbf{k}_\perp\cdot\mathbf{1}_\perp}{|\mathbf{k}_\perp + \mathbf{1}_\perp||\mathbf{k}_\perp|}\big) + (\delta_1 + 2k_y l_y)\delta_1\big(\frac{\mathbf{k}_\perp\cdot\mathbf{1}_\perp}{|\mathbf{k}_\perp|} + \frac{(\mathbf{k}_\perp + \mathbf{1}_\perp)\cdot\mathbf{1}_\perp}{|\mathbf{k}_\perp + \mathbf{1}_\perp|}\big)}{\big[(\delta_1 + 2k_y l_y)^2 + (|\mathbf{k}_\perp + \mathbf{1}_\perp| + |\mathbf{k}_\perp|)^2\big]\big[\delta_1^2 + \mathbf{l}_\perp^2\big]^2}.$$

Integrating over $l_x$, we have

$$\delta\Delta_f(4) = -\frac{\Delta_f\Delta_b^2}{2}\int\frac{d\mathbf{k}_\perp dk_y d\mathbf{l}_\perp dl_y}{(2\pi)^{2d-2}}\frac{(2k_y l_y)^2 - (|\mathbf{k}_\perp| + |\mathbf{k}_\perp + \mathbf{1}_\perp| + |\mathbf{1}_\perp|)^2}{\big[(2k_y l_y)^2 + (|\mathbf{k}_\perp| + |\mathbf{k}_\perp + \mathbf{1}_\perp| + |\mathbf{1}_\perp|)^2\big]^2}\Big(1 - \frac{(\mathbf{k}_\perp + \mathbf{1}_\perp)\cdot\mathbf{1}_\perp}{|\mathbf{k}_\perp + \mathbf{1}_\perp||\mathbf{1}_\perp|}\Big)\Big(1 - \frac{\mathbf{k}_\perp\cdot\mathbf{1}_\perp}{|\mathbf{k}_\perp||\mathbf{1}_\perp|}\Big).$$

Integrated over $k_y$ and $l_y$, this vanishes. As a result, we obtain

$$\delta\Delta_f(4) = 0.$$

### D.1.5 Feynman diagram FV2-5

From the vertex correction in Table 5 FV2-5, we find a renormalization factor as

$$\delta\Delta_f(5) = -4\Delta_f^3\int\frac{d^d k d^d l}{(2\pi)^{2d}}\frac{\big[\delta_{\mathbf{k+1}}\delta_\mathbf{k}\delta_\mathbf{l} - (\mathbf{k}_\perp + \mathbf{1}_\perp)\cdot\mathbf{k}_\perp\delta_\mathbf{l} - \mathbf{k}_\perp\cdot\mathbf{1}_\perp\delta_{\mathbf{k+1}} - (\mathbf{k}_\perp + \mathbf{1}_\perp)\cdot\mathbf{1}_\perp\delta_\mathbf{k}\big]\delta_{-1}}{\big[\delta_{\mathbf{k+1}}^2 + (\mathbf{k}_\perp + \mathbf{1}_\perp)^2\big]\big[\delta_\mathbf{k}^2 + \mathbf{k}_\perp^2\big]\big[\delta_1^2 + \mathbf{l}_\perp^2\big]\big[\delta_{-1}^2 + \mathbf{l}_\perp^2\big]}.$$

Integrating over $k_x$, we obtain

$$\delta\Delta_f(5) = -2\Delta_f^3\int\frac{d\mathbf{k}_\perp dk_y d^d l}{(2\pi)^{2d-1}}\frac{\delta_1\delta_{-1}(|\mathbf{k}_\perp + \mathbf{1}_\perp| + |\mathbf{k}_\perp|)\big(1 - \frac{(\mathbf{k}_\perp + \mathbf{1}_\perp)\cdot\mathbf{k}_\perp}{|\mathbf{k}_\perp + \mathbf{1}_\perp||\mathbf{k}_\perp|}\big) + (\delta_1 + 2k_y l_y)\delta_{-1}\big(\frac{(\mathbf{k}_\perp + \mathbf{1}_\perp)\cdot\mathbf{1}_\perp}{|\mathbf{k}_\perp + \mathbf{1}_\perp|} - \frac{\mathbf{k}_\perp\cdot\mathbf{1}_\perp}{|\mathbf{k}_\perp|}\big)}{\big[(\delta_1 + 2k_y l_y)^2 + (|\mathbf{k}_\perp + \mathbf{1}_\perp| + |\mathbf{k}_\perp|)^2\big]\big[\delta_1^2 + \mathbf{l}_\perp^2\big]\big[\delta_{-1}^2 + \mathbf{l}_\perp^2\big]}.$$

Integrating over $k_y$, we have

$$\delta\Delta_f(5) = -\frac{\Delta_f^3}{2}\int\frac{d\mathbf{k}_\perp d^d l}{(2\pi)^{2d-2}}\frac{\delta_1\delta_{-1}}{|l_y|\big[\delta_1^2 + \mathbf{l}_\perp^2\big]\big[\delta_{-1}^2 + \mathbf{l}_\perp^2\big]}\Big(1 - \frac{(\mathbf{k}_\perp + \mathbf{1}_\perp)\cdot\mathbf{k}_\perp}{|\mathbf{k}_\perp + \mathbf{1}_\perp||\mathbf{k}_\perp|}\Big).$$

Integrating over $l_x$, we obtain

$$\delta\Delta_f(5) = \frac{\Delta_f^3}{8}\int\frac{d\mathbf{k}_\perp d\mathbf{l}_\perp dl_y}{(2\pi)^{2d-3}}\frac{|\mathbf{l}_\perp|}{|l_y|\big[l_y^4 + \mathbf{l}_\perp^2\big]}\Big(1 - \frac{(\mathbf{k}_\perp + \mathbf{1}_\perp)\cdot\mathbf{k}_\perp}{|\mathbf{k}_\perp + \mathbf{1}_\perp||\mathbf{k}_\perp|}\Big).$$

Integrating over $l_y$, we have

$$\delta\Delta_f(5) = \frac{\Delta_f^3}{32\pi}\int\frac{d\mathbf{k}_\perp d\mathbf{l}_\perp}{(2\pi)^{2d-4}}\frac{\ln(1 + \mathbf{l}_\perp^2/\Lambda^4)}{|\mathbf{l}_\perp|}\Big(1 - \frac{(\mathbf{k}_\perp + \mathbf{1}_\perp)\cdot\mathbf{k}_\perp}{|\mathbf{k}_\perp + \mathbf{1}_\perp||\mathbf{k}_\perp|}\Big).$$

We drop this correction because it does not give a simple pole responsible for renormalization. As a result, we obtain

$$\delta\Delta_f(5) = 0. \tag{D.2}$$

### D.1.6 Feynman diagram FV2-6

From the vertex correction in Table 5 FV2-6, we find a renormalization factor as

$$\delta\Delta_f(6) = -4\Delta_f^3 \int \frac{d^dk d^dl}{(2\pi)^{2d}} \frac{\left[\delta_{k+l}\delta_k\delta_l - (\mathbf{k}_\perp + \mathbf{l}_\perp)\cdot\mathbf{k}_\perp\delta_l - \mathbf{k}_\perp\cdot\mathbf{l}_\perp\delta_{k+l} - (\mathbf{k}_\perp + \mathbf{l}_\perp)\cdot\mathbf{l}_\perp\delta_k\right]\delta_l}{\left[\delta_{k+l}^2 + (\mathbf{k}_\perp + \mathbf{l}_\perp)^2\right]\left[\delta_k^2 + \mathbf{k}_\perp^2\right]\left[\delta_l^2 + \mathbf{l}_\perp^2\right]^2}.$$

Integrating over $k_x$, we have

$$\delta\Delta_f(6) = -2\Delta_f^3 \int \frac{d\mathbf{k}_\perp dk_y d^dl}{(2\pi)^{2d-1}} \frac{\delta_l^2(|\mathbf{k}_\perp + \mathbf{l}_\perp| + |\mathbf{k}_\perp|)\left(1 - \frac{(\mathbf{k}_\perp + \mathbf{l}_\perp)\cdot\mathbf{k}_\perp}{|\mathbf{k}_\perp + \mathbf{l}_\perp||\mathbf{k}_\perp|}\right) + (\delta_l + 2k_y l_y)\delta_l\left(\frac{(\mathbf{k}_\perp + \mathbf{l}_\perp)\cdot\mathbf{l}_\perp}{|\mathbf{k}_\perp + \mathbf{l}_\perp|} - \frac{\mathbf{k}_\perp\cdot\mathbf{l}_\perp}{|\mathbf{k}_\perp|}\right)}{\left[(\delta_l + 2k_y l_y)^2 + (|\mathbf{k}_\perp + \mathbf{l}_\perp| + |\mathbf{k}_\perp|)^2\right]\left[\delta_l^2 + \mathbf{l}_\perp^2\right]^2}.$$

Integrating over $l_x$, we obtain

$$\delta\Delta_f(6) = -\frac{\Delta_f^3}{2} \int \frac{d\mathbf{k}_\perp dk_y d\mathbf{l}_\perp dl_y}{(2\pi)^{2d-2}} \frac{(2k_y l_y)^2 - (|\mathbf{k}_\perp| + |\mathbf{k}_\perp + \mathbf{l}_\perp| + |\mathbf{l}_\perp|)^2}{\left[(2k_y l_y)^2 + (|\mathbf{k}_\perp| + |\mathbf{k}_\perp + \mathbf{l}_\perp| + |\mathbf{l}_\perp|)^2\right]^2}$$

$$\times\left(1 - \frac{(\mathbf{k}_\perp + \mathbf{l}_\perp)\cdot\mathbf{k}_\perp}{|\mathbf{k}_\perp + \mathbf{l}_\perp||\mathbf{k}_\perp|} + \frac{\mathbf{k}_\perp\cdot\mathbf{l}_\perp}{|\mathbf{k}_\perp||\mathbf{l}_\perp|} - \frac{(\mathbf{k}_\perp + \mathbf{l}_\perp)\cdot\mathbf{l}_\perp}{|\mathbf{k}_\perp + \mathbf{l}_\perp||\mathbf{l}_\perp|}\right).$$

Integrated over $k_y$ and $l_y$, this vanishes. As a result, we obtain

$$\delta\Delta_f(6) = 0.$$

### D.1.7 Feynman diagram FV2-7

From the vertex correction in Table 5 FV2-7, we find a renormalization factor as

$$\delta\Delta_f(7) = -4\Delta_f^2\Delta_b \int \frac{d^dk d^dl}{(2\pi)^{2d}} \frac{\left[\delta_{k+l}\delta_k\delta_{-l} - (\mathbf{k}_\perp + \mathbf{l}_\perp)\cdot\mathbf{k}_\perp\delta_{-l} - \mathbf{k}_\perp\cdot\mathbf{l}_\perp\delta_{k+l} - (\mathbf{k}_\perp + \mathbf{l}_\perp)\cdot\mathbf{l}_\perp\delta_k\right]\delta_l}{\left[\delta_{k+l}^2 + (\mathbf{k}_\perp + \mathbf{l}_\perp)^2\right]\left[\delta_k^2 + \mathbf{k}_\perp^2\right]\left[\delta_l^2 + \mathbf{l}_\perp^2\right]\left[\delta_{-l}^2 + \mathbf{l}_\perp^2\right]}.$$

Integrating over $k_x$, we have

$$\delta\Delta_f(7) = -2\Delta_f^2\Delta_b \int \frac{d\mathbf{k}_\perp dk_y d^dl}{(2\pi)^{2d-1}} \frac{\delta_l\delta_{-l}(|\mathbf{k}_\perp + \mathbf{l}_\perp| + |\mathbf{k}_\perp|)\left(1 - \frac{(\mathbf{k}_\perp + \mathbf{l}_\perp)\cdot\mathbf{k}_\perp}{|\mathbf{k}_\perp + \mathbf{l}_\perp||\mathbf{k}_\perp|}\right) + (\delta_l + 2k_y l_y)\delta_l\left(\frac{(\mathbf{k}_\perp + \mathbf{l}_\perp)\cdot\mathbf{l}_\perp}{|\mathbf{k}_\perp + \mathbf{l}_\perp|} - \frac{\mathbf{k}_\perp\cdot\mathbf{l}_\perp}{|\mathbf{k}_\perp|}\right)}{\left[(\delta_l + 2k_y l_y)^2 + (|\mathbf{k}_\perp + \mathbf{l}_\perp| + |\mathbf{k}_\perp|)^2\right]\left[\delta_l^2 + \mathbf{l}_\perp^2\right]\left[\delta_{-l}^2 + \mathbf{l}_\perp^2\right]}.$$

Integrating over $k_y$, we obtain

$$\delta\Delta_f(7) = -\frac{\Delta_f^2\Delta_b}{2} \int \frac{d\mathbf{k}_\perp d^dl}{(2\pi)^{2d-2}} \frac{\delta_l\delta_{-l}}{|l_y|\left[\delta_l^2 + \mathbf{l}_\perp^2\right]\left[\delta_{-l}^2 + \mathbf{l}_\perp^2\right]}\left(1 - \frac{(\mathbf{k}_\perp + \mathbf{l}_\perp)\cdot\mathbf{k}_\perp}{|\mathbf{k}_\perp + \mathbf{l}_\perp||\mathbf{k}_\perp|}\right).$$

We drop this correction because it does not give a simple pole responsible for renormalization. As a result, we obtain

$$\delta\Delta_f(7) = 0.$$

### D.1.8 Feynman diagram FV2-8

From the vertex correction in Table 5 FV2-8, we find a renormalization factor as

$$\delta\Delta_f(8) = -4\Delta_f^2\Delta_b \int \frac{d^dk d^dl}{(2\pi)^{2d}} \frac{\left[\delta_{k+l}\delta_k\delta_{-l} - \mathbf{k}_\perp\cdot(\mathbf{k}_\perp + \mathbf{l}_\perp)\delta_{-l} - \mathbf{k}_\perp\cdot\mathbf{l}_\perp\delta_{k+l} - \mathbf{l}_\perp\cdot(\mathbf{k}_\perp + \mathbf{l}_\perp)\delta_k\right]\delta_{-l}}{\left[\delta_{k+l}^2 + (\mathbf{k}_\perp + \mathbf{l}_\perp)^2\right]\left[\delta_k^2 + \mathbf{k}_\perp^2\right]\left[\delta_{-l}^2 + \mathbf{l}_\perp^2\right]^2}.$$

Integrating over $k_x$, we have

$$\delta\Delta_f(8) = -2\Delta_f^2\Delta_b \int \frac{d\mathbf{k}_\perp dk_y d^d l}{(2\pi)^{2d-1}} \frac{\delta_{-1}^2(|\mathbf{k}_\perp + \mathbf{1}_\perp| + |\mathbf{k}_\perp|)\left(1 - \frac{(\mathbf{k}_\perp + \mathbf{1}_\perp)\cdot\mathbf{k}_\perp}{|\mathbf{k}_\perp + \mathbf{1}_\perp||\mathbf{k}_\perp|}\right) + (\delta_1 + 2k_y l_y)\delta_{-1}\left(\frac{(\mathbf{k}_\perp + \mathbf{1}_\perp)\cdot\mathbf{1}_\perp}{|\mathbf{k}_\perp + \mathbf{1}_\perp|} - \frac{\mathbf{k}_\perp\cdot\mathbf{1}_\perp}{|\mathbf{k}_\perp|}\right)}{\left[(\delta_1 + 2k_y l_y)^2 + (|\mathbf{k}_\perp + \mathbf{1}_\perp| + |\mathbf{k}_\perp|)^2\right]\left[\delta_{-1}^2 + \mathbf{1}_\perp^2\right]^2} .$$

Integrating over $l_x$, we obtain

$$\delta\Delta_f(8) = -\frac{\Delta_f^2\Delta_b}{2} \int \frac{d\mathbf{k}_\perp dk_y d\mathbf{1}_\perp dl_y}{(2\pi)^{2d-2}} \frac{(2k_y l_y + 2l_y^2)^2 - (|\mathbf{k}_\perp| + |\mathbf{k}_\perp + \mathbf{1}_\perp| + |\mathbf{1}_\perp|)^2}{\left[(2k_y l_y + 2l_y^2)^2 + (|\mathbf{k}_\perp| + |\mathbf{k}_\perp + \mathbf{1}_\perp| + |\mathbf{1}_\perp|)^2\right]^2}$$
$$\times \left(1 - \frac{(\mathbf{k}_\perp + \mathbf{1}_\perp)\cdot\mathbf{k}_\perp}{|\mathbf{k}_\perp + \mathbf{1}_\perp||\mathbf{k}_\perp|} - \frac{\mathbf{k}_\perp\cdot\mathbf{1}_\perp}{|\mathbf{k}_\perp||\mathbf{1}_\perp|} + \frac{(\mathbf{k}_\perp + \mathbf{1}_\perp)\cdot\mathbf{1}_\perp}{|\mathbf{k}_\perp + \mathbf{1}_\perp||\mathbf{1}_\perp|}\right).$$

Integrated over $k_y$ and $l_y$, this vanishes. As a result, we obtain

$$\delta\Delta_f(8) = 0.$$

### D.1.9 Feynman diagram FV2-9

From the vertex correction in Table 5 FV2-9, we find a renormalization factor as

$$\delta\Delta_f(9) = -4\Delta_f\Delta_b^2 \int \frac{d^d k d^d l}{(2\pi)^{2d}} \frac{\left[\delta_{\mathbf{k}+1}\delta_{-\mathbf{k}}\delta_1 - \mathbf{k}_\perp\cdot(\mathbf{k}_\perp + \mathbf{1}_\perp)\delta_1 - \mathbf{k}_\perp\cdot\mathbf{1}_\perp\delta_{\mathbf{k}+1} - \mathbf{1}_\perp\cdot(\mathbf{k}_\perp + \mathbf{1}_\perp)\delta_{-\mathbf{k}}\right]\delta_1}{\left[\delta_{\mathbf{k}+1}^2 + (\mathbf{k}_\perp + \mathbf{1}_\perp)^2\right]\left[\delta_{-\mathbf{k}}^2 + \mathbf{k}_\perp^2\right]\left[\delta_1^2 + \mathbf{1}_\perp^2\right]^2} .$$

Integrating over $k_x$, we have

$$\delta\Delta_f(9) = 2\Delta_f\Delta_b^2 \int \frac{d\mathbf{k}_\perp dk_y d^d l}{(2\pi)^{2d-1}} \frac{\delta_1^2(|\mathbf{k}_\perp + \mathbf{1}_\perp| + |\mathbf{k}_\perp|)\left(1 + \frac{(\mathbf{k}_\perp + \mathbf{1}_\perp)\cdot\mathbf{k}_\perp}{|\mathbf{k}_\perp + \mathbf{1}_\perp||\mathbf{k}_\perp|}\right) + (\delta_1 + 2k_y l_y + 2k_y^2)\delta_1\left(\frac{\mathbf{k}_\perp\cdot\mathbf{1}_\perp}{|\mathbf{k}_\perp|} + \frac{(\mathbf{k}_\perp + \mathbf{1}_\perp)\cdot\mathbf{1}_\perp}{|\mathbf{k}_\perp + \mathbf{1}_\perp|}\right)}{\left[(\delta_1 + 2k_y l_y + 2k_y^2)^2 + (|\mathbf{k}_\perp + \mathbf{1}_\perp| + |\mathbf{k}_\perp|)^2\right]\left[\delta_1^2 + \mathbf{1}_\perp^2\right]^2} .$$

Integrating over $l_x$, we obtain

$$\delta\Delta_f(9) = \frac{\Delta_f\Delta_b^2}{2} \int \frac{d\mathbf{k}_\perp dk_y d\mathbf{1}_\perp dl_y}{(2\pi)^{2d-2}} \frac{(2k_y l_y + 2k_y^2)^2 - (|\mathbf{k}_\perp| + |\mathbf{k}_\perp + \mathbf{1}_\perp| + |\mathbf{1}_\perp|)^2}{\left[(2k_y l_y + 2k_y^2)^2 + (|\mathbf{k}_\perp| + |\mathbf{k}_\perp + \mathbf{1}_\perp| + |\mathbf{1}_\perp|)^2\right]^2}$$
$$\times \left(1 + \frac{(\mathbf{k}_\perp + \mathbf{1}_\perp)\cdot\mathbf{k}_\perp}{|\mathbf{k}_\perp + \mathbf{1}_\perp||\mathbf{k}_\perp|} - \frac{\mathbf{k}_\perp\cdot\mathbf{1}_\perp}{|\mathbf{k}_\perp||\mathbf{1}_\perp|} - \frac{(\mathbf{k}_\perp + \mathbf{1}_\perp)\cdot\mathbf{1}_\perp}{|\mathbf{k}_\perp + \mathbf{1}_\perp||\mathbf{1}_\perp|}\right).$$

Integrated over $k_y$ and $l_y$, this vanishes. As a result, we obtain

$$\delta\Delta_f(9) = 0.$$

### D.1.10 Feynman diagram FV2-10

From the vertex correction in Table 5 FV2-10, we find a renormalization factor as

$$\delta\Delta_f(10) = -\frac{4g^2\Delta_f^2}{N} \int \frac{d^{d+1}k d^{d+1}l}{(2\pi)^{2d+1}} \delta(l_0) \frac{\left[\delta_{\mathbf{k}+1}\delta_{\mathbf{k}}\delta_1 - (\mathbf{K}+\mathbf{L})\cdot\mathbf{K}\delta_1 - \mathbf{K}\cdot\mathbf{L}\delta_{\mathbf{k}+1} - (\mathbf{K}+\mathbf{L})\cdot\mathbf{L}\delta_{\mathbf{k}}\right]\delta_{-1}}{\left[\delta_{\mathbf{k}+1}^2 + (\mathbf{K}+\mathbf{L})^2\right]\left[\delta_{\mathbf{k}}^2 + \mathbf{K}^2\right]\left[\delta_1^2 + \mathbf{L}^2\right]\left[\delta_{-1}^2 + \mathbf{L}^2\right]\left[k_y^2 + g^2 B_d \frac{|\mathbf{K}|^{d-1}}{|k_y|}\right]} .$$

Integrating over $k_x$, we obtain

$$\delta\Delta_f(10) = -\frac{2g^2\Delta_f^2}{N} \int \frac{d\mathbf{K} dk_y d^{d+1}l}{(2\pi)^{2d}} \delta(l_0) \frac{\delta_1\delta_{-1}(|\mathbf{K}+\mathbf{L}| + |\mathbf{K}|)\left(1 - \frac{(\mathbf{K}+\mathbf{L})\cdot\mathbf{K}}{|\mathbf{K}+\mathbf{L}||\mathbf{K}|}\right) + \delta_1\delta_{-1}\left(\frac{(\mathbf{K}+\mathbf{L})\cdot\mathbf{L}}{|\mathbf{K}+\mathbf{L}|} - \frac{\mathbf{K}\cdot\mathbf{L}}{|\mathbf{K}|}\right)}{\left[\delta_1^2 + (|\mathbf{K}+\mathbf{L}| + |\mathbf{K}|)^2\right]\left[\delta_1^2 + \mathbf{L}^2\right]\left[\delta_{-1}^2 + \mathbf{L}^2\right]\left[k_y^2 + g^2 B_d \frac{|\mathbf{K}|^{d-1}}{|k_y|}\right]} ,$$

where we have neglected $k_y l_y$ in the fermionic part since it would give rise to subleading terms in $g$. Integrating over $l_x$, we have

$$\delta\Delta_f(10) = -\frac{2g^2\Delta_f^2}{N}\int\frac{d\mathbf{K}dk_y d\mathbf{L}dl_y}{(2\pi)^{2d-1}}\delta(l_0)\frac{(2l_y^2)^2 - 2|\mathbf{L}|(|\mathbf{K}| + |\mathbf{K}+\mathbf{L}| + |\mathbf{L}|)}{\left[k_y^2 + g^2 B_d\frac{|\mathbf{K}|^{d-1}}{|k_y|}\right]\left[(2l_y^2)^2 + (|\mathbf{K}| + |\mathbf{K}+\mathbf{L}| + |\mathbf{L}|)^2\right]\left[(2l_y^2)^2 + 4|\mathbf{L}|^2\right]}$$

$$\times\left[\frac{|\mathbf{K}| + |\mathbf{K}+\mathbf{L}|}{(|\mathbf{K}| + |\mathbf{K}+\mathbf{L}| + |\mathbf{L}|)}\left(1 - \frac{(\mathbf{K}+\mathbf{L})\cdot\mathbf{K}}{|\mathbf{K}+\mathbf{L}||\mathbf{K}|}\right) + \frac{|\mathbf{L}|}{(|\mathbf{K}| + |\mathbf{K}+\mathbf{L}| + |\mathbf{L}|)}\left(\frac{(\mathbf{K}+\mathbf{L})\cdot\mathbf{L}}{|\mathbf{K}+\mathbf{L}||\mathbf{L}|} - \frac{\mathbf{K}\cdot\mathbf{L}}{|\mathbf{K}||\mathbf{L}|}\right)\right].$$

Integrating over $k_y$ and $l_y$, we obtain

$$\delta\Delta_f(10) = \frac{g^{4/3}\Delta_f^2}{3\sqrt{3}B_d^{1/3}N}\int\frac{d\mathbf{K}d\mathbf{L}}{(2\pi)^{2d-3}}\frac{\delta(l_0)}{|\mathbf{K}|^{(d-1)/3}}\frac{1}{\sqrt{2|\mathbf{L}|}(|\mathbf{K}| + |\mathbf{K}+\mathbf{L}| + |\mathbf{L}|) + 2|\mathbf{L}|\sqrt{|\mathbf{K}| + |\mathbf{K}+\mathbf{L}| + |\mathbf{L}|}}$$

$$\times\left[\frac{|\mathbf{K}| + |\mathbf{K}+\mathbf{L}|}{|\mathbf{K}| + |\mathbf{K}+\mathbf{L}| + |\mathbf{L}|}\left(1 - \frac{(\mathbf{K}+\mathbf{L})\cdot\mathbf{K}}{|\mathbf{K}+\mathbf{L}||\mathbf{K}|}\right) + \frac{|\mathbf{L}|}{|\mathbf{K}| + |\mathbf{K}+\mathbf{L}| + |\mathbf{L}|}\left(\frac{(\mathbf{K}+\mathbf{L})\cdot\mathbf{L}}{|\mathbf{K}+\mathbf{L}||\mathbf{L}|} - \frac{\mathbf{K}\cdot\mathbf{L}}{|\mathbf{K}||\mathbf{L}|}\right)\right].$$

Introducing coordinates as $\mathbf{k}_\perp\cdot\mathbf{l}_\perp = Kl\cos\theta$, and scaling variables as $l \to Kl$ and $k_0 \to Kk$, we get

$$\delta\Delta_f(10) = \frac{\Omega g^{4/3}\Delta_f^2}{3\pi\sqrt{3}B_d^{1/3}N}\int_{|\mathbf{P}|}^\infty dK K^{\frac{10d-31}{6}}\int_0^\infty dl l^{d-3}\int_0^\infty\frac{dk}{(1+k^2)^{(d-1)/6}}\int_0^\pi d\theta\sin^{d-4}\theta\frac{1}{\sqrt{2l}\eta + 2l\sqrt{\eta}}$$

$$\times\left[\frac{\eta - l}{\eta}\left(1 - \frac{1 + k^2 + l\cos\theta}{\sqrt{1+k^2}\sqrt{1+k^2+l^2+2l\cos\theta}}\right) + \frac{l}{\eta}\left(\frac{l + \cos\theta}{\sqrt{1+k^2+l^2+2l\cos\theta}} - \frac{\cos\theta}{\sqrt{1+k^2}}\right)\right],$$

where $\Omega \equiv \frac{4}{(4\pi)^{d-2}\sqrt{\pi}\Gamma(\frac{d-2}{2})\Gamma(\frac{d-3}{2})}$ and $\eta = \sqrt{1+k^2} + l + \sqrt{1+k^2+l^2+2l\cos\theta}$. We find an $\epsilon$ pole from the $K$ integral as $\int_{|\mathbf{P}|}^\infty dK K^{\frac{10d-31}{6}} = \frac{3}{5\epsilon} + \mathcal{O}(1)$. The remaining integral can be done numerically as

$$\int_0^\infty dl l^{d-3}\int_0^\infty\frac{dk}{(1+k^2)^{(d-1)/6}}\int_0^\pi d\theta\sin^{d-4}\theta\frac{1}{\sqrt{2l}\eta + 2l\sqrt{\eta}}\left[\frac{\eta - l}{\eta}\left(1 - \frac{1+k^2+l\cos\theta}{\sqrt{1+k^2}\sqrt{1+k^2+l^2+2l\cos\theta}}\right)\right.$$

$$\left. + \frac{l}{\eta}\left(\frac{l+\cos\theta}{\sqrt{1+k^2+l^2+2l\cos\theta}} - \frac{\cos\theta}{\sqrt{1+k^2}}\right)\right] = \frac{\sqrt{\pi}\Gamma(\frac{d-3}{2})}{\Gamma(\frac{d-2}{2})}(0.4415).$$

As a result, we obtain

$$\delta\Delta_f(10) = (1.7951)\frac{\Delta_f\tilde{\Delta}_f\tilde{g}}{\epsilon}. \tag{D.3}$$

### D.1.11 Feynman diagram FV2-11

From the vertex correction in Table 5 FV2-11, we find a renormalization factor as

$$\delta\Delta_f(11) = -\frac{4g^2\Delta_f^2}{N}\int\frac{d^{d+1}k d^{d+1}l}{(2\pi)^{2d+1}}\delta(l_0)\frac{\left[\delta_{\mathbf{k+l}}\delta_{\mathbf{k}}\delta_{\mathbf{l}} - (\mathbf{K}+\mathbf{L})\cdot\mathbf{L}\delta_{\mathbf{k}} - \mathbf{K}\cdot\mathbf{L}\delta_{\mathbf{k+l}} - (\mathbf{K}+\mathbf{L})\cdot\mathbf{K}\delta_{\mathbf{l}}\right]\delta_{\mathbf{l}}}{\left[\delta_{\mathbf{k+l}}^2 + (\mathbf{K}+\mathbf{L})^2\right]\left[\delta_{\mathbf{k}}^2 + \mathbf{K}^2\right]\left[\delta_{\mathbf{l}}^2 + \mathbf{L}^2\right]^2\left[k_y^2 + g^2 B_d\frac{|\mathbf{K}|^{d-1}}{|k_y|}\right]}.$$

Integrating over $k_x$, we get

$$\delta\Delta_f(11) = -\frac{2g^2\Delta_f^2}{N}\int\frac{d\mathbf{K}dk_y d^{d+1}l}{(2\pi)^{2d}}\delta(l_0)\frac{\delta_{\mathbf{l}}^2(|\mathbf{K}+\mathbf{L}| + |\mathbf{K}|)\left(1 - \frac{(\mathbf{K}+\mathbf{L})\cdot\mathbf{K}}{|\mathbf{K}+\mathbf{L}||\mathbf{K}|}\right) + (\delta_{\mathbf{l}} + 2k_y l_y)\delta_{\mathbf{l}}\left(\frac{(\mathbf{K}+\mathbf{L})\cdot\mathbf{L}}{|\mathbf{K}+\mathbf{L}|} - \frac{\mathbf{K}\cdot\mathbf{L}}{|\mathbf{K}|}\right)}{\left[(\delta_{\mathbf{l}} + 2k_y l_y)^2 + (|\mathbf{K}+\mathbf{L}| + |\mathbf{K}|)^2\right]\left[\delta_{\mathbf{l}}^2 + \mathbf{L}^2\right]^2\left[k_y^2 + g^2 B_d\frac{|\mathbf{K}|^{d-1}}{|k_y|}\right]}.$$

Integrating over $l_x$, we obtain

$$\delta\Delta_f(11) = -\frac{g^2\Delta_f^2}{2N}\int\frac{d\mathbf{K}dk_y d\mathbf{L}dl_y}{(2\pi)^{2d-1}}\frac{\delta(l_0)}{k_y^2 + g^2 B_d\frac{|\mathbf{K}|^{d-1}}{|k_y|}}\left[\frac{|\mathbf{K}| + |\mathbf{K}+\mathbf{L}| + |\mathbf{L}|}{|\mathbf{L}|\left[(2k_y l_y)^2 + (|\mathbf{K}| + |\mathbf{K}+\mathbf{L}| + |\mathbf{L}|)^2\right]}\left(1 - \frac{(\mathbf{K}+\mathbf{L})\cdot\mathbf{K}}{|\mathbf{K}+\mathbf{L}||\mathbf{K}|}\right)\right.$$

$$\left. + \frac{(2k_y l_y)^2 - (|\mathbf{K}| + |\mathbf{K}+\mathbf{L}| + |\mathbf{L}|)^2}{\left[(2k_y l_y)^2 + (|\mathbf{K}| + |\mathbf{K}+\mathbf{L}| + |\mathbf{L}|)^2\right]^2}\left(1 - \frac{(\mathbf{K}+\mathbf{L})\cdot\mathbf{L}}{|\mathbf{K}+\mathbf{L}||\mathbf{L}|} + \frac{\mathbf{K}\cdot\mathbf{L}}{|\mathbf{K}||\mathbf{L}|} - \frac{(\mathbf{K}+\mathbf{L})\cdot\mathbf{K}}{|\mathbf{K}+\mathbf{L}||\mathbf{K}|}\right)\right].$$

Integrating over $k_y$ and $l_y$, we have

$$\delta\Delta_f(11) = -\frac{g^{2/3}\Delta_f^2}{12\sqrt{3}B_d^{2/3}N}\int\frac{d\mathbf{K}d\mathbf{L}}{(2\pi)^{2d-3}}\frac{\delta(l_0)}{|\mathbf{L}||\mathbf{K}|^{2(d-1)/3}}\left(1-\frac{(\mathbf{K}+\mathbf{L})\cdot\mathbf{K}}{|\mathbf{K}+\mathbf{L}||\mathbf{K}|}\right).$$

Introducing coordinates as $\mathbf{k}_\perp\cdot\mathbf{l}_\perp = Kl\cos\theta$ and scaling variables as $l\to Kl$ and $k_0\to Kk$, we obtain

$$\delta\Delta_f(11) = -\frac{\Omega g^{2/3}\Delta_f^2}{12\pi\sqrt{3}B_d^{2/3}N}\int_{|\mathbf{P}|}^\infty dK K^{\frac{4d-13}{3}}\int_0^\infty dl\, l^{d-4}\int_0^\infty\frac{dk}{(1+k^2)^{\frac{d-1}{3}}}\int_0^\pi d\theta\sin^{d-4}\theta$$
$$\times\left(1-\frac{1+k^2+l\cos\theta}{\sqrt{1+k^2}\sqrt{1+k^2+l^2+2l\cos\theta}}\right).$$

We find an $\epsilon$ pole from the $K$ integral as $\int_{|\mathbf{P}|}^\infty dK K^{\frac{4d-13}{3}} = \frac{3}{4\epsilon}+\mathcal{O}(1)$. The remaining integral can be done numerically as

$$\int_0^\infty dl\, l^{d-4}\int_0^\infty\frac{dk}{(1+k^2)^{\frac{d-1}{3}}}\int_0^\pi d\theta\sin^{d-4}\theta\left(1-\frac{1+k^2+l\cos\theta}{\sqrt{1+k^2}\sqrt{1+k^2+l^2+2l\cos\theta}}\right) = \frac{\sqrt{\pi}\Gamma(\frac{d-3}{2})}{\Gamma(\frac{d-2}{2})}(4.934).$$

As a result, we obtain

$$\delta\Delta_f(11) = (-4.162)\frac{\Delta_f\tilde{\Delta}_f\sqrt{\tilde{g}}}{\sqrt{N}\epsilon}. \tag{D.4}$$

### D.1.12 Feynman diagram FV2-12

From the vertex correction in Table 5 FV2-12, we find a renormalization factor as

$$\delta\Delta_f(12) = -\frac{4g^2\Delta_b^2}{N}\int\frac{d^{d+1}k d^{d+1}l}{(2\pi)^{2d+1}}\delta(l_0)\frac{\left[\delta_{\mathbf{k+l}}\delta_{-\mathbf{k}}\delta_{\mathbf{l}}-\mathbf{K}\cdot(\mathbf{K}+\mathbf{L})\delta_{\mathbf{l}}-\mathbf{K}\cdot\mathbf{L}\delta_{\mathbf{k+l}}-\mathbf{L}\cdot(\mathbf{K}+\mathbf{L})\delta_{-\mathbf{k}}\right]\delta_{\mathbf{l}}}{\left[\delta_{\mathbf{k+l}}^2+(\mathbf{K}+\mathbf{L})^2\right]\left[\delta_{-\mathbf{k}}^2+\mathbf{K}^2\right]\left[\delta_{\mathbf{l}}^2+\mathbf{L}^2\right]^2\left[k_y^2+g^2B_d\frac{|\mathbf{K}|^{d-1}}{|k_y|}\right]}.$$

Integrating over $k_x$, we obtain

$$\delta\Delta_f(12) = \frac{2g^2\Delta_b^2}{N}\int\frac{d\mathbf{K}dk_y d^{d+1}l}{(2\pi)^{2d}}\delta(l_0)\frac{\delta_{\mathbf{l}}^2(|\mathbf{K}+\mathbf{L}|+|\mathbf{K}|)\left(1+\frac{(\mathbf{K}+\mathbf{L})\cdot\mathbf{K}}{|\mathbf{K}+\mathbf{L}||\mathbf{K}|}\right)+(\delta_{\mathbf{l}}+2k_y l_y+2k_y^2)\delta_{\mathbf{l}}\left(\frac{\mathbf{K}\cdot\mathbf{L}}{|\mathbf{K}|}+\frac{(\mathbf{K}+\mathbf{L})\cdot\mathbf{L}}{|\mathbf{K}+\mathbf{L}|}\right)}{\left[(\delta_{\mathbf{l}}+2k_y l_y+2k_y^2)^2+(|\mathbf{K}+\mathbf{L}|+|\mathbf{K}|)^2\right]\left[\delta_{\mathbf{l}}^2+\mathbf{L}^2\right]^2\left[k_y^2+g^2B_d\frac{|\mathbf{K}|^{d-1}}{|k_y|}\right]}.$$

Integrating over $l_x$, we get

$$\delta\Delta_f(12) = \frac{g^2\Delta_b^2}{2N}\int\frac{d\mathbf{K}dk_y d\mathbf{L}dl_y}{(2\pi)^{2d-1}}\frac{\delta(l_0)}{k_y^2+g^2B_d\frac{|\mathbf{K}|^{d-1}}{|k_y|}}\left[\frac{|\mathbf{K}|+|\mathbf{K}+\mathbf{L}|+|\mathbf{L}|}{|\mathbf{L}|\left[(2k_y l_y)^2+(|\mathbf{K}|+|\mathbf{K}+\mathbf{L}|+|\mathbf{L}|)^2\right]}\left(1+\frac{(\mathbf{K}+\mathbf{L})\cdot\mathbf{K}}{|\mathbf{K}+\mathbf{L}||\mathbf{K}|}\right)\right.$$
$$\left.+\frac{(2k_y l_y)^2-(|\mathbf{K}|+|\mathbf{K}+\mathbf{L}|+|\mathbf{L}|)^2}{\left[(2k_y l_y)^2+(|\mathbf{K}|+|\mathbf{K}+\mathbf{L}|+|\mathbf{L}|)^2\right]^2}\left(1+\frac{(\mathbf{K}+\mathbf{L})\cdot\mathbf{L}}{|\mathbf{K}+\mathbf{L}||\mathbf{L}|}-\frac{\mathbf{K}\cdot\mathbf{L}}{|\mathbf{K}||\mathbf{L}|}-\frac{(\mathbf{K}+\mathbf{L})\cdot\mathbf{K}}{|\mathbf{K}+\mathbf{L}||\mathbf{K}|}\right)\right],$$

where we have ignored the $k_y^2$ terms in the fermionic part since they would give rise to sub-leading terms in $g$. Integrating over $k_y$ and $l_y$, we have

$$\delta\Delta_f(12) = \frac{g^{2/3}\Delta_b^2}{12\sqrt{3}B_d^{2/3}N}\int\frac{d\mathbf{K}d\mathbf{L}}{(2\pi)^{2d-3}}\frac{\delta(l_0)}{|\mathbf{L}||\mathbf{K}|^{2(d-1)/3}}\left(1+\frac{(\mathbf{K}+\mathbf{L})\cdot\mathbf{K}}{|\mathbf{K}+\mathbf{L}||\mathbf{K}|}\right).$$

The term of $|\mathbf{L}|^{-1}|\mathbf{K}|^{-2(d-1)/3}$ does not give rise to an $\epsilon$ pole, so we drop it. Then, we have

$$\delta\Delta_f(12) = \frac{g^{2/3}\Delta_b^2}{12\sqrt{3}B_d^{2/3}N}\int\frac{d\mathbf{K}d\mathbf{L}}{(2\pi)^{2d-3}}\frac{\delta(l_0)}{|\mathbf{L}|}\frac{(\mathbf{K}+\mathbf{L})\cdot\mathbf{K}}{|\mathbf{K}+\mathbf{L}||\mathbf{K}|^{\frac{2d+1}{3}}}.$$

Integrating over $\mathbf{K}$, we obtain

$$\delta\Delta_f(12) = \frac{g^{2/3}\Delta_b^2}{12\sqrt{3}B_d^{2/3}N} \int_0^1 dx \frac{x^{-\frac{1}{2}}(1-x)^{\frac{2d-5}{6}}}{\Gamma(\frac{1}{2})\Gamma(\frac{2d+1}{6})} \int \frac{d\mathbf{L}}{(2\pi)^{d-2}} \delta(l_0) \frac{-4\Gamma(\frac{7-d}{6})}{(4\pi)^{\frac{d-1}{2}}|\mathbf{L}|\left[x(1-x)\mathbf{L}^2\right]^{\frac{1-d}{6}}}$$

$$= \frac{g^{2/3}\Delta_b^2}{12\sqrt{3}B_d^{2/3}N} \frac{-4\Gamma(\frac{7-d}{6})}{(4\pi)^{d-2}\sqrt{\pi}\Gamma(\frac{d-2}{2})} \int_0^1 dx \frac{x^{\frac{d-4}{6}}(1-x)^{\frac{3d-6}{6}}}{\Gamma(\frac{1}{2})\Gamma(\frac{2d+1}{6})} \int_{p_0}^\infty dL L^{\frac{4d-13}{3}}.$$

We find an $\epsilon$ pole from the $L$ integral as $\int_{p_0}^\infty dL L^{\frac{4d-13}{3}} = \frac{3}{4\epsilon} + \mathcal{O}(1)$. The remaining integral can be done as

$$\int_0^1 dx \frac{x^{\frac{d-4}{6}}(1-x)^{\frac{3d-6}{6}}}{\Gamma(\frac{1}{2})\Gamma(\frac{2d+1}{6})} = \frac{\sqrt{\pi}}{2\sqrt{2}} + \mathcal{O}(\epsilon).$$

As a result, we obtain

$$\delta\Delta_f(12) = (-4.162)\frac{\Delta_f\tilde{\Delta}_b^2\sqrt{\tilde{g}}}{\tilde{\Delta}_f\sqrt{N}\epsilon}. \tag{D.5}$$

### D.1.13 Feynman diagram FV2-13

From the vertex correction in Table 5 FV2-13, we find a renormalization factor as

$$\delta\Delta_f(13) = -2\Delta_f^3 \int \frac{d^dk d^dl}{(2\pi)^{2d}} \frac{\left[\delta_{\mathbf{k+l}}\delta_{\mathbf{l}}^2 - 2(\mathbf{k}_\perp + \mathbf{l}_\perp)\cdot\mathbf{l}_\perp\delta_{\mathbf{l}} - \mathbf{l}_\perp^2\delta_{\mathbf{k+l}}\right]\delta_{-\mathbf{k}}}{\left[\delta_{\mathbf{k+l}}^2 + (\mathbf{k}_\perp + \mathbf{l}_\perp)^2\right]\left[\delta_{-\mathbf{k}}^2 + \mathbf{k}_\perp^2\right]\left[\delta_{\mathbf{l}}^2 + \mathbf{l}_\perp^2\right]^2}.$$

Integrating over $k_x$, we obtain

$$\delta\Delta_f(13) = \Delta_f^3 \int \frac{d^dk d\mathbf{l}_\perp dl_y}{(2\pi)^{2d-1}} \frac{(\delta_{\mathbf{l}}^2 - \mathbf{l}_\perp^2)(|\mathbf{k}_\perp + \mathbf{l}_\perp| + |\mathbf{k}_\perp|) + 2(\delta_{\mathbf{l}} + 2k_y l_y + 2k_y^2)\delta_{\mathbf{l}}\frac{(\mathbf{k}_\perp + \mathbf{l}_\perp)\cdot\mathbf{l}_\perp}{|\mathbf{k}_\perp + \mathbf{l}_\perp|}}{\left[(\delta_{\mathbf{l}} + 2k_y l_y + 2k_y^2)^2 + (|\mathbf{k}_\perp + \mathbf{l}_\perp| + |\mathbf{k}_\perp|)^2\right]\left[\delta_{\mathbf{l}}^2 + \mathbf{l}_\perp^2\right]^2}.$$

Integrating over $l_x$, we get

$$\delta\Delta_f(13) = \frac{\Delta_f^3}{2} \int \frac{d\mathbf{k}_\perp dk_y d\mathbf{l}_\perp dl_y}{(2\pi)^{2d-2}} \frac{(2k_y l_y + 2k_y^2)^2 - (|\mathbf{k}_\perp| + |\mathbf{k}_\perp + \mathbf{l}_\perp| + |\mathbf{l}_\perp|)^2}{\left[(2k_y l_y + 2l_y^2)^2 + (|\mathbf{k}_\perp| + |\mathbf{k}_\perp + \mathbf{l}_\perp| + |\mathbf{l}_\perp|)^2\right]^2}\left(1 - \frac{(\mathbf{k}_\perp + \mathbf{l}_\perp)\cdot\mathbf{l}_\perp}{|\mathbf{k}_\perp + \mathbf{l}_\perp||\mathbf{l}_\perp|}\right).$$

Integrated over $k_y$ and $l_y$, this vanishes. As a result, we obtain

$$\delta\Delta_f(13) = 0.$$

### D.1.14 Feynman diagram FV2-14

From the vertex correction in Table 5 FV2-14, we find a renormalization factor as

$$\delta\Delta_f(14) = -2\Delta_f^3 \int \frac{d^dk d^dl}{(2\pi)^{2d}} \frac{\left[\delta_{\mathbf{k+l}}\delta_{\mathbf{l}}^2 - 2(\mathbf{k}_\perp + \mathbf{l}_\perp)\cdot\mathbf{l}_\perp\delta_{\mathbf{l}} - \mathbf{l}_\perp^2\delta_{\mathbf{k+l}}\right]\delta_{\mathbf{k}}}{\left[\delta_{\mathbf{k+l}}^2 + (\mathbf{k}_\perp + \mathbf{l}_\perp)^2\right]\left[\delta_{\mathbf{k}}^2 + \mathbf{k}_\perp^2\right]\left[\delta_{\mathbf{l}}^2 + \mathbf{l}_\perp^2\right]^2}.$$

Integrating over $k_x$, we have

$$\delta\Delta_f(14) = -\Delta_f^3 \int \frac{d\mathbf{k}_\perp dk_y d^dl}{(2\pi)^{2d-1}} \frac{(\delta_{\mathbf{l}}^2 - \mathbf{l}_\perp^2)(|\mathbf{k}_\perp + \mathbf{l}_\perp| + |\mathbf{k}_\perp|) + 2(\delta_{\mathbf{l}} + 2k_y l_y)\delta_{\mathbf{l}}\frac{(\mathbf{k}_\perp + \mathbf{l}_\perp)\cdot\mathbf{l}_\perp}{|\mathbf{k}_\perp + \mathbf{l}_\perp|}}{\left[(\delta_{\mathbf{l}} + 2k_y l_y)^2 + (|\mathbf{k}_\perp + \mathbf{l}_\perp| + |\mathbf{k}_\perp|)^2\right]\left[\delta_{\mathbf{l}}^2 + \mathbf{l}_\perp^2\right]^2}.$$

Integrating over $l_x$, we get

$$\delta\Delta_f(14) = -\frac{\Delta_f^3}{2}\int \frac{d\mathbf{k}_\perp dk_y d\mathbf{l}_\perp dl_y}{(2\pi)^{2d-2}} \frac{(2k_y l_y)^2 - (|\mathbf{k}_\perp| + |\mathbf{k}_\perp + \mathbf{l}_\perp| + |\mathbf{l}_\perp|)^2}{\left[(2k_y l_y)^2 + (|\mathbf{k}_\perp| + |\mathbf{k}_\perp + \mathbf{l}_\perp| + |\mathbf{l}_\perp|)^2\right]^2}$$
$$\times \left(1 - \frac{(\mathbf{k}_\perp + \mathbf{l}_\perp)\cdot \mathbf{l}_\perp}{|\mathbf{k}_\perp + \mathbf{l}_\perp||\mathbf{l}_\perp|}\right).$$

Integrated over $k_y$ and $l_y$, this vanishes. As a result, we obtain

$$\delta\Delta_f(14) = 0.$$

### D.1.15   Feynman diagram FV2-15

From the vertex correction in Table 5 FV2-15, we find a renormalization factor as

$$\delta\Delta_f(15) = -2\Delta_f \Delta_b^2 \int \frac{d^d k\, d^d l}{(2\pi)^{2d}} \frac{\left[\delta_{\mathbf{k+l}}\delta_{-\mathbf{l}}^2 - 2(\mathbf{k}_\perp + \mathbf{l}_\perp)\cdot \mathbf{l}_\perp \delta_{-\mathbf{l}} - \mathbf{l}_\perp^2 \delta_{\mathbf{k+l}}\right]\delta_\mathbf{k}}{\left[\delta_{\mathbf{k+l}}^2 + (\mathbf{k}_\perp + \mathbf{l}_\perp)^2\right]\left[\delta_\mathbf{k}^2 + \mathbf{k}_\perp^2\right]\left[\delta_{-\mathbf{l}}^2 + \mathbf{l}_\perp^2\right]^2}.$$

Integrating over $k_x$, we have

$$\delta\Delta_f(15) = -\Delta_f \Delta_b^2 \int \frac{d\mathbf{k}_\perp dk_y d^d l}{(2\pi)^{2d-1}} \frac{(\delta_{-\mathbf{l}}^2 - \mathbf{l}_\perp^2)(|\mathbf{k}_\perp + \mathbf{l}_\perp| + |\mathbf{k}_\perp|) + 2(\delta_\mathbf{l} + 2k_y l_y)\delta_{-\mathbf{l}}\frac{(\mathbf{k}_\perp + \mathbf{l}_\perp)\cdot \mathbf{l}_\perp}{|\mathbf{k}_\perp + \mathbf{l}_\perp|}}{\left[(\delta_\mathbf{l} + 2k_y l_y)^2 + (|\mathbf{k}_\perp + \mathbf{l}_\perp| + |\mathbf{k}_\perp|)^2\right]\left[\delta_{-\mathbf{l}}^2 + \mathbf{l}_\perp^2\right]^2}.$$

Integrating over $l_x$, we obtain

$$\delta\Delta_f(15) = -\frac{\Delta_f \Delta_b^2}{2}\int \frac{d\mathbf{k}_\perp dk_y d\mathbf{l}_\perp dl_y}{(2\pi)^{2d-2}} \frac{(2k_y l_y + 2l_y^2)^2 - (|\mathbf{k}_\perp| + |\mathbf{k}_\perp + \mathbf{l}_\perp| + |\mathbf{l}_\perp|)^2}{\left[(2k_y l_y + 2l_y^2)^2 + (|\mathbf{k}_\perp| + |\mathbf{k}_\perp + \mathbf{l}_\perp| + |\mathbf{l}_\perp|)^2\right]^2}$$
$$\times \left(1 + \frac{(\mathbf{k}_\perp + \mathbf{l}_\perp)\cdot \mathbf{l}_\perp}{|\mathbf{k}_\perp + \mathbf{l}_\perp||\mathbf{l}_\perp|}\right).$$

Integrated over $k_y$ and $l_y$, this vanishes. As a result, we obtain

$$\delta\Delta_f(15) = 0.$$

### D.1.16   Feynman diagram FV2-16

From the vertex correction in Table 5 FV2-16, we find a renormalization factor as

$$\delta\Delta_f(16) = -2\Delta_f^2 \Delta_b \int \frac{d^d k\, d^d l}{(2\pi)^{2d}} \frac{\left[\delta_{\mathbf{k+l}}\delta_\mathbf{l}^2 - 2(\mathbf{k}_\perp + \mathbf{l}_\perp)\cdot \mathbf{l}_\perp \delta_\mathbf{l} - \mathbf{l}_\perp^2 \delta_{\mathbf{k+l}}\right]\delta_\mathbf{k}}{\left[\delta_{\mathbf{k+l}}^2 + (\mathbf{k}_\perp + \mathbf{l}_\perp)^2\right]\left[\delta_\mathbf{k}^2 + \mathbf{k}_\perp^2\right]\left[\delta_\mathbf{l}^2 + \mathbf{l}_\perp^2\right]^2}.$$

The integration is the same with $\delta\Delta_f(14)$. As a result, we obtain

$$\delta\Delta_f(16) = 0.$$

### D.1.17   Feynman diagram FV2-17

From the vertex correction in Table 5 FV2-17, we find a renormalization factor as

$$\delta\Delta_f(17) = -2\Delta_f^2 \Delta_b \int \frac{d^d k\, d^d l}{(2\pi)^{2d}} \frac{\left[\delta_{\mathbf{k+l}}\delta_\mathbf{l}^2 - 2(\mathbf{k}_\perp + \mathbf{l}_\perp)\cdot \mathbf{l}_\perp \delta_\mathbf{l} - \mathbf{l}_\perp^2 \delta_{\mathbf{k+l}}\right]\delta_{-\mathbf{k}}}{\left[\delta_{\mathbf{k+l}}^2 + (\mathbf{k}_\perp + \mathbf{l}_\perp)^2\right]\left[\delta_{-\mathbf{k}}^2 + \mathbf{k}_\perp^2\right]\left[\delta_\mathbf{l}^2 + \mathbf{l}_\perp^2\right]^2}.$$

The integration is the same with $\delta\Delta_f(13)$. As a result, we obtain

$$\delta\Delta_f(17) = 0.$$

### D.1.18 Feynman diagram FV2-18

From the vertex correction in Table 5 FV2-18, we find a renormalization factor as

$$\delta\Delta_f(18) = -2\Delta_b^3 \int \frac{d^d k d^d l}{(2\pi)^{2d}} \frac{\left[\delta_{\mathbf{k+l}}\delta_{-\mathbf{l}}^2 - 2(\mathbf{k}_\perp + \mathbf{l}_\perp)\cdot \mathbf{l}_\perp \delta_{-\mathbf{l}} - \mathbf{l}_\perp^2 \delta_{\mathbf{k+l}}\right]\delta_{\mathbf{k}}}{\left[\delta_{\mathbf{k+l}}^2 + (\mathbf{k}_\perp + \mathbf{l}_\perp)^2\right]\left[\delta_{\mathbf{k}}^2 + \mathbf{k}_\perp^2\right]\left[\delta_{-\mathbf{l}}^2 + \mathbf{l}_\perp^2\right]^2}.$$

The integration is the same with $\delta\Delta_f(15)$. As a result, we obtain

$$\delta\Delta_f(18) = 0.$$

### D.1.19 Feynman diagram FV2-19

From the vertex correction in Table 5 FV2-19, we find a renormalization factor as

$$\delta\Delta_f(19) = -\frac{2g^2\Delta_f^2}{N} \int \frac{d^{d+1} k d^{d+1} l}{(2\pi)^{2d+1}} \delta(k_0) \frac{\left[\delta_{\mathbf{k+l}}\delta_{\mathbf{l}}^2 - 2(\mathbf{K}+\mathbf{L})\cdot\mathbf{L}\delta_{\mathbf{l}} - \mathbf{L}^2\delta_{\mathbf{k+l}}\right]\delta_{-\mathbf{k}}}{\left[\delta_{\mathbf{k+l}}^2 + (\mathbf{K}+\mathbf{L})^2\right]\left[\delta_{-\mathbf{k}}^2 + \mathbf{K}^2\right]\left[\delta_{\mathbf{l}}^2 + \mathbf{L}^2\right]^2\left[l_y^2 + g^2 B_d \frac{|\mathbf{L}|^{d-1}}{|l_y|}\right]}.$$

Integrating over $k_x$, we get

$$\delta\Delta_f(19) = \frac{g^2\Delta_f^2}{N} \int \frac{d\mathbf{K} d^{d+1} l}{(2\pi)^{2d}} \delta(k_0) \frac{(\delta_{\mathbf{l}}^2 - \mathbf{L}^2)(|\mathbf{K}+\mathbf{L}|+|\mathbf{K}|) + 2(\delta_{\mathbf{l}} + 2k_y l_y + 2k_y^2)\delta_{\mathbf{l}}\frac{(\mathbf{K}+\mathbf{L})\cdot\mathbf{L}}{|\mathbf{K}+\mathbf{L}|}}{\left[(\delta_{\mathbf{l}} + 2k_y l_y + 2k_y^2)^2 + (|\mathbf{K}+\mathbf{L}|+|\mathbf{K}|)^2\right]\left[\delta_{\mathbf{l}}^2 + \mathbf{L}^2\right]^2\left[l_y^2 + g^2 B_d \frac{|\mathbf{L}|^{d-1}}{|l_y|}\right]}.$$

Integrating over $l_x$, we have

$$\delta\Delta_f(19) = \frac{g^2\Delta_f^2}{2N} \int \frac{d\mathbf{K} dk_y d\mathbf{L} dl_y}{(2\pi)^{2d-1}} \frac{\delta(k_0)}{l_y^2 + g^2 B_d \frac{|\mathbf{L}|^{d-1}}{|l_y|}} \frac{(2k_y l_y + 2k_y^2)^2 - (|\mathbf{K}| + |\mathbf{K}+\mathbf{L}| + |\mathbf{L}|)^2}{\left[(2k_y l_y + 2k_y^2)^2 + (|\mathbf{K}| + |\mathbf{K}+\mathbf{L}| + |\mathbf{L}|)^2\right]^2}\left(1 - \frac{(\mathbf{K}+\mathbf{L})\cdot\mathbf{L}}{|\mathbf{K}+\mathbf{L}||\mathbf{L}|}\right).$$

We may ignore $k_y l_y$ since it would give rise to subleading terms in $g$. Integrating over $k_y$ and $l_y$, we obtain

$$\delta\Delta_f(19) = -\frac{g^{4/3}\Delta_f^2}{24\sqrt{3}B_d^{1/3}N} \int \frac{d\mathbf{K} d\mathbf{L}}{(2\pi)^{2d-3}} \frac{\delta(k_0)}{|\mathbf{L}|^{(d-1)/3}} \frac{1}{(|\mathbf{K}| + |\mathbf{K}+\mathbf{L}| + |\mathbf{L}|)^{3/2}}\left(1 - \frac{(\mathbf{K}+\mathbf{L})\cdot\mathbf{L}}{|\mathbf{K}+\mathbf{L}||\mathbf{L}|}\right).$$

Introducing coordinates as $\mathbf{K}\cdot\mathbf{L} = Kl\cos\theta$, $K = Lk$, and $l_0 = Ll$, we have

$$\delta\Delta_f(19) = -\frac{\Omega g^{4/3}\Delta_f^2}{24\pi\sqrt{3}B_d^{1/3}N} \int_{p_0}^\infty dL L^{\frac{10d-31}{6}} \int_0^\infty dk k^{d-3} \int_0^\infty \frac{dl}{(1+l^2)^{(d-1)/6}} \int_0^\pi d\theta \sin^{d-4}\theta$$
$$\times \frac{1}{\left[\sqrt{1+l^2} + k + \sqrt{1+l^2+k^2+2k\cos\theta}\right]^{3/2}}\left(1 - \frac{1+l^2+k\cos\theta}{\sqrt{1+l^2}\sqrt{1+l^2+k^2+2k\cos\theta}}\right).$$

We find an $\epsilon$ pole from the $K$ integral as $\int_{|\mathbf{P}|}^\infty dL L^{\frac{10d-31}{6}} = \frac{3}{5\epsilon} + \mathcal{O}(1)$. The remaining integral can be done numerically as

$$\int_0^\infty dk k^{d-3} \int_0^\infty \frac{dl}{(1+l^2)^{(d-1)/6}} \int_0^\pi d\theta \sin^{d-4}\theta \frac{1}{\left[\sqrt{1+l^2} + k + \sqrt{1+l^2+k^2+2k\cos\theta}\right]^{3/2}}$$
$$\times \left(1 - \frac{1+l^2+k\cos\theta}{\sqrt{1+l^2}\sqrt{1+l^2+k^2+2k\cos\theta}}\right) = \frac{\sqrt{\pi}\,\Gamma(\frac{d-3}{2})}{\Gamma(\frac{d-2}{2})}(-0.5439).$$

As a result, we obtain

$$\delta\Delta_f(19) = (0.2765)\Delta_f \frac{\tilde{\Delta}_f \tilde{g}}{\epsilon}. \tag{D.6}$$

### D.1.20   Feynman diagram FV2-20

From the vertex correction in Table 5 FV2-20, we find the renormalization factor as

$$\delta\Delta_f(20) = -\frac{2g^2\Delta_f^2}{N}\int\frac{d^{d+1}kd^{d+1}l}{(2\pi)^{2d+1}}\delta(l_0)\frac{\left[\delta_{\mathbf{k+l}}\delta_{\mathbf{l}}^2 - 2(\mathbf{K+L})\cdot\mathbf{L}\delta_{\mathbf{l}} - \mathbf{L}^2\delta_{\mathbf{k+l}}\right]\delta_{\mathbf{k}}}{\left[\delta_{\mathbf{k+l}}^2 + (\mathbf{K+L})^2\right]\left[\delta_{\mathbf{k}}^2 + \mathbf{K}^2\right]\left[\delta_{\mathbf{l}}^2 + \mathbf{L}^2\right]^2\left[l_y^2 + g^2B_d\frac{|\mathbf{L}|^{d-1}}{|l_y|}\right]}.$$

Integrating over $k_x$, we have

$$\delta\Delta_f(20) = -\frac{g^2\Delta_f^2}{N}\int\frac{d\mathbf{K}dk_yd^{d+1}l}{(2\pi)^{2d}}\delta(l_0)\frac{(\delta_{\mathbf{l}}^2 - \mathbf{L}^2)(|\mathbf{K+L}| + |\mathbf{K}|) + 2(\delta_{\mathbf{l}} + 2k_yl_y)\delta_{\mathbf{l}}\frac{(\mathbf{K+L})\cdot\mathbf{L}}{|\mathbf{K+L}|}}{\left[(\delta_{\mathbf{l}} + 2k_yl_y)^2 + (|\mathbf{K+L}| + |\mathbf{K}|)^2\right]\left[\delta_{\mathbf{l}}^2 + \mathbf{L}^2\right]^2\left[l_y^2 + g^2B_d\frac{|\mathbf{L}|^{d-1}}{|k_y|}\right]}.$$

Integrating over $l_x$, we obtain

$$\delta\Delta_f(20) = -\frac{g^2\Delta_f^2}{2N}\int\frac{d\mathbf{K}dk_yd\mathbf{L}dl_y}{(2\pi)^{2d-1}}\frac{\delta(l_0)}{l_y^2 + g^2B_d\frac{|\mathbf{L}|^{d-1}}{|l_y|}}\frac{(2k_yl_y)^2 - (|\mathbf{K}| + |\mathbf{K+L}| + |\mathbf{L}|)^2}{\left[(2k_yl_y)^2 + (|\mathbf{K}| + |\mathbf{K+L}| + |\mathbf{L}|)^2\right]^2}\left(1 - \frac{(\mathbf{K+L})\cdot\mathbf{L}}{|\mathbf{K+L}||\mathbf{L}|}\right).$$

Integrated over $k_y$ and $l_y$, this vanishes. As a result, we obtain

$$\delta\Delta_f(20) = 0\,.$$

### D.1.21   Feynman diagram FV2-21

From the vertex correction in Table 5 FV2-21, we find a renormalization factor as

$$\delta\Delta_f(21) = -\frac{2g^2\Delta_b^2}{N}\int\frac{d^{d+1}kd^{d+1}l}{(2\pi)^{2d+1}}\delta(k_0)\frac{\left[\delta_{\mathbf{k+l}}\delta_{-\mathbf{l}}^2 - 2\mathbf{L}\cdot(\mathbf{K+L})\delta_{-\mathbf{l}} - \mathbf{L}^2\delta_{\mathbf{k+l}}\right]\delta_{\mathbf{k}}}{\left[\delta_{\mathbf{k+l}}^2 + (\mathbf{K+L})^2\right]\left[\delta_{\mathbf{k}}^2 + \mathbf{K}^2\right]\left[\delta_{-\mathbf{l}}^2 + \mathbf{L}^2\right]^2\left[l_y^2 + g^2B_d\frac{|\mathbf{L}|^{d-1}}{|l_y|}\right]}.$$

Integrating over $k_x$, we have

$$\delta\Delta_f(21) = -\frac{g^2\Delta_b^2}{N}\int\frac{d\mathbf{K}dk_yd^{d+1}l}{(2\pi)^{2d}}\delta(k_0)\frac{(\delta_{-\mathbf{l}}^2 - \mathbf{L}^2)(|\mathbf{K+L}| + |\mathbf{K}|) + 2(\delta_{\mathbf{l}} + 2k_yl_y)\delta_{-\mathbf{l}}\frac{(\mathbf{K+L})\cdot\mathbf{L}}{|\mathbf{K+L}|}}{\left[(\delta_{\mathbf{l}} + 2k_yl_y)^2 + (|\mathbf{K+L}| + |\mathbf{K}|)^2\right]\left[\delta_{-\mathbf{l}}^2 + \mathbf{L}^2\right]^2\left[l_y^2 + g^2B_d\frac{|\mathbf{L}|^{d-1}}{|l_y|}\right]}.$$

Integrating over $l_x$, we get

$$\delta\Delta_f(21) = -\frac{g^2\Delta_b^2}{2N}\int\frac{d\mathbf{K}dk_yd\mathbf{L}dl_y}{(2\pi)^{2d-1}}\frac{\delta(k_0)}{l_y^2 + g^2B_d\frac{|\mathbf{L}|^{d-1}}{|l_y|}}\frac{(2k_yl_y + 2l_y^2)^2 - (|\mathbf{K}| + |\mathbf{K+L}| + |\mathbf{L}|)^2}{\left[(2k_yl_y + 2l_y^2)^2 + (|\mathbf{K}| + |\mathbf{K+L}| + |\mathbf{L}|)^2\right]^2}\left(1 + \frac{(\mathbf{K+L})\cdot\mathbf{L}}{|\mathbf{K+L}||\mathbf{L}|}\right).$$

Integrated over $k_y$ and $l_y$, this vanishes. As a result, we obtain

$$\delta\Delta_f(21) = 0\,.$$

## D.2   Backward scattering

### D.2.1   Feynman diagram BV2-1

From the vertex correction in Table 6 BV2-1, we find a renormalization factor as

$$\delta\Delta_b(1) = -4\Delta_f^2\Delta_b\int\frac{d^dkd^dl}{(2\pi)^{2d}}\frac{\left[\delta_{\mathbf{k+l}}\delta_{\mathbf{l}} - (\mathbf{k}_\perp + \mathbf{l}_\perp)\cdot\mathbf{l}_\perp\right]\left[\delta_{\mathbf{k}}\delta_{\mathbf{l}} + \mathbf{k}_\perp\cdot\mathbf{l}_\perp\right]}{\left[\delta_{\mathbf{k+l}}^2 + (\mathbf{k}_\perp + \mathbf{l}_\perp)^2\right]\left[\delta_{\mathbf{k}}^2 + \mathbf{k}_\perp^2\right]\left[\delta_{\mathbf{l}}^2 + \mathbf{l}_\perp^2\right]^2}.$$

The integration is the same with $\delta\Delta_f(4)$. As a result, we obtain

$$\delta\Delta_b(1) = 0\,.$$

### D.2.2 Feynman diagram BV2-2

From the vertex correction in Table 6 BV2-2, we find a renormalization factor as

$$\delta\Delta_b(2) = -2\Delta_f^2\Delta_b \int \frac{d^dk\,d^dl}{(2\pi)^{2d}} \frac{\left[\delta_{\mathbf{k+l}}\delta_{\mathbf{l}} - (\mathbf{k}_\perp + \mathbf{l}_\perp)\cdot\mathbf{l}_\perp\right]\left[\delta_{-\mathbf{k}}\delta_{\mathbf{l}} + \mathbf{k}_\perp\cdot\mathbf{l}_\perp\right]}{\left[\delta_{\mathbf{k+l}}^2 + (\mathbf{k}_\perp + \mathbf{l}_\perp)^2\right]\left[\delta_{-\mathbf{k}}^2 + \mathbf{k}_\perp^2\right]\left[\delta_{\mathbf{l}}^2 + \mathbf{l}_\perp^2\right]^2}.$$

The integration is the same with $\delta\Delta_f(2)$. As a result, we obtain

$$\delta\Delta_b(2) = 0.$$

### D.2.3 Feynman diagram BV2-3

From the vertex correction in Table 6 BV2-3, we find a renormalization factor as

$$\delta\Delta_b(3) = -4\Delta_f^2\Delta_b \int \frac{d^dk\,d^dl}{(2\pi)^{2d}} \frac{\left[\delta_{\mathbf{k+l}}\delta_{-\mathbf{k}}\delta_{\mathbf{l}} - \mathbf{k}_\perp\cdot(\mathbf{k}_\perp + \mathbf{l}_\perp)\delta_{\mathbf{l}} - \mathbf{k}_\perp\cdot\mathbf{l}_\perp\delta_{\mathbf{k+l}} - \mathbf{l}_\perp\cdot(\mathbf{k}_\perp + \mathbf{l}_\perp)\delta_{-\mathbf{k}}\right]\delta_{\mathbf{l}}}{\left[\delta_{\mathbf{k+l}}^2 + (\mathbf{k}_\perp + \mathbf{l}_\perp)^2\right]\left[\delta_{-\mathbf{k}}^2 + \mathbf{k}_\perp^2\right]\left[\delta_{\mathbf{l}}^2 + \mathbf{l}_\perp^2\right]^2}.$$

The integration is the same with $\delta\Delta_f(9)$. As a result, we obtain

$$\delta\Delta_b(3) = 0.$$

### D.2.4 Feynman diagram BV2-4

From the vertex correction in Table 6 BV2-4, we find a renormalization factor as

$$\delta\Delta_b(4) = -4\Delta_f^2\Delta_b \int \frac{d^dk\,d^dl}{(2\pi)^{2d}} \frac{\left[\delta_{\mathbf{k+l}}\delta_{\mathbf{k}}\delta_{\mathbf{l}} - \mathbf{k}_\perp\cdot(\mathbf{k}_\perp + \mathbf{l}_\perp)\delta_{\mathbf{l}} - \mathbf{k}_\perp\cdot\mathbf{l}_\perp\delta_{\mathbf{k+l}} - \mathbf{l}_\perp\cdot(\mathbf{k}_\perp + \mathbf{l}_\perp)\delta_{\mathbf{k}}\right]\delta_{\mathbf{l}}}{\left[\delta_{\mathbf{k+l}}^2 + (\mathbf{k}_\perp + \mathbf{l}_\perp)^2\right]\left[\delta_{\mathbf{k}}^2 + \mathbf{k}_\perp^2\right]\left[\delta_{\mathbf{l}}^2 + \mathbf{l}_\perp^2\right]^2}.$$

The integration is the same with $\delta\Delta_f(6)$. As a result, we obtain

$$\delta\Delta_b(4) = 0.$$

### D.2.5 Feynman diagram BV2-5

From the vertex correction in Table 6 BV2-5, we find a renormalization factor as

$$\delta\Delta_b(5) = -4\Delta_f\Delta_b^2 \int \frac{d^dk\,d^dl}{(2\pi)^{2d}} \frac{\left[\delta_{\mathbf{k+l}}\delta_{\mathbf{k}}\delta_{-\mathbf{l}} - \mathbf{k}_\perp\cdot(\mathbf{k}_\perp + \mathbf{l}_\perp)\delta_{-\mathbf{l}} - \mathbf{k}_\perp\cdot\mathbf{l}_\perp\delta_{\mathbf{k+l}} - \mathbf{l}_\perp\cdot(\mathbf{k}_\perp + \mathbf{l}_\perp)\delta_{\mathbf{k}}\right]\delta_{-\mathbf{l}}}{\left[\delta_{\mathbf{k+l}}^2 + (\mathbf{k}_\perp + \mathbf{l}_\perp)^2\right]\left[\delta_{\mathbf{k}}^2 + \mathbf{k}_\perp^2\right]\left[\delta_{-\mathbf{l}}^2 + \mathbf{l}_\perp^2\right]^2}.$$

The integration is the same with $\delta\Delta_f(8)$. As a result, we obtain

$$\delta\Delta_b(5) = 0.$$

### D.2.6 Feynman diagram BV2-6

From the vertex correction in Table 6 BV2-6, we find a renormalization factor as

$$\delta\Delta_b(6) = -\frac{4g^2\Delta_f\Delta_b}{N} \int \frac{d^{d+1}k\,d^{d+1}l}{(2\pi)^{2d+1}} \delta(l_0) \frac{\left[\delta_{\mathbf{k+l}}\delta_{-\mathbf{k}}\delta_{\mathbf{l}} - \mathbf{K}\cdot(\mathbf{K}+\mathbf{L})\delta_{\mathbf{l}} - \mathbf{K}\cdot\mathbf{L}\delta_{\mathbf{k+l}} - \mathbf{L}\cdot(\mathbf{K}+\mathbf{L})\delta_{-\mathbf{k}}\right]\delta_{\mathbf{l}}}{\left[\delta_{\mathbf{k+l}}^2 + (\mathbf{K}+\mathbf{L})^2\right]\left[\delta_{-\mathbf{k}}^2 + \mathbf{K}^2\right]\left[\delta_{\mathbf{l}}^2 + \mathbf{L}^2\right]^2\left[k_y^2 + g^2B_d\frac{|\mathbf{K}|^{d-1}}{|k_y|}\right]}.$$

The integration is the same with $\delta\Delta_f(12)$. As a result, we obtain

$$\delta\Delta_b(6) = 0.$$

### D.2.7 Feynman diagram BV2-7

From the vertex correction in Table 6 BV2-7, we find a renormalization factor as

$$\delta\Delta_b(7) = -\frac{4g^2\Delta_f\Delta_b}{N}\int\frac{d^{d+1}k\,d^{d+1}l}{(2\pi)^{2d+1}}\delta(l_0)\frac{\left[\delta_{\mathbf{k+l}}\delta_{\mathbf{k}}\delta_{\mathbf{l}} - \mathbf{K}\cdot(\mathbf{K+L})\delta_{\mathbf{l}} - \mathbf{K}\cdot\mathbf{L}\delta_{\mathbf{k+l}} - \mathbf{L}\cdot(\mathbf{K+L})\delta_{\mathbf{k}}\right]\delta_{\mathbf{l}}}{\left[\delta_{\mathbf{k+l}}^2 + (\mathbf{K+L})^2\right]\left[\delta_{\mathbf{k}}^2 + \mathbf{K}^2\right]\left[\delta_{\mathbf{l}}^2 + \mathbf{L}^2\right]^2\left[k_y^2 + g^2B_d\frac{|\mathbf{K}|^{d-1}}{|k_y|}\right]}.$$

The integration is the same with $\delta\Delta_f(11)$. As a result, we obtain

$$\delta\Delta_b(7) = 0.$$

### D.2.8 Feynman diagram BV2-8

From the vertex correction in Table 6 BV2-8, we find a renormalization factor as

$$\delta\Delta_b(8) = -4\Delta_f^2\Delta_b\int\frac{d^dk\,d^dl}{(2\pi)^{2d}}\frac{\left[\delta_{\mathbf{k+l}}\delta_{\mathbf{l}}\delta_{-\mathbf{l}} - \mathbf{l}_\perp\cdot(\mathbf{k}_\perp+\mathbf{l}_\perp)\delta_{\mathbf{l}} - \mathbf{l}_\perp\cdot(\mathbf{k}_\perp+\mathbf{l}_\perp)\delta_{-\mathbf{l}} - \mathbf{l}_\perp^2\delta_{\mathbf{k+l}}\right]\delta_{\mathbf{k}}}{\left[\delta_{\mathbf{k+l}}^2 + (\mathbf{k}_\perp+\mathbf{l}_\perp)^2\right]\left[\delta_{\mathbf{k}}^2 + \mathbf{k}_\perp^2\right]\left[\delta_{\mathbf{l}}^2 + \mathbf{l}_\perp^2\right]\left[\delta_{-\mathbf{l}}^2 + \mathbf{l}_\perp^2\right]}.$$

Integrating over $k_x$, we obtain

$$\delta\Delta_b(8) = -2\Delta_f^2\Delta_b\int\frac{d\mathbf{k}_\perp dk_y\,d^dl}{(2\pi)^{2d-1}}\frac{(\delta_{\mathbf{l}}\delta_{-\mathbf{l}} - \mathbf{l}_\perp^2)(|\mathbf{k}_\perp+\mathbf{l}_\perp| + |\mathbf{k}_\perp|) + (\delta_{\mathbf{l}} + 2k_yl_y)(\delta_{\mathbf{l}} + \delta_{-\mathbf{l}})\frac{(\mathbf{k}_\perp+\mathbf{l}_\perp)\cdot\mathbf{l}_\perp}{|\mathbf{k}_\perp+\mathbf{l}_\perp|}}{\left[(\delta_{\mathbf{l}} + 2k_yl_y)^2 + (|\mathbf{k}_\perp+\mathbf{l}_\perp| + |\mathbf{k}_\perp|)^2\right]\left[\delta_{\mathbf{l}}^2 + \mathbf{l}_\perp^2\right]\left[\delta_{-\mathbf{l}}^2 + \mathbf{l}_\perp^2\right]}.$$

Integrating over $k_y$, we get

$$\delta\Delta_b(8) = -\frac{\Delta_f^2\Delta_b}{2}\int\frac{d\mathbf{k}_\perp d^dl}{(2\pi)^{2d-2}}\frac{\delta_{\mathbf{l}}\delta_{-\mathbf{l}} - \mathbf{l}_\perp^2}{|l_y|\left[\delta_{\mathbf{l}}^2 + \mathbf{l}_\perp^2\right]\left[\delta_{-\mathbf{l}}^2 + \mathbf{l}_\perp^2\right]}.$$

Integrating over $l_x$, we have

$$\delta\Delta_b(8) = \frac{\Delta_f^2\Delta_b}{8}\int\frac{d\mathbf{k}_\perp d\mathbf{l}_\perp dl_y}{(2\pi)^{2d-3}}\frac{|\mathbf{l}_\perp|}{|l_y|\left[l_y^4 + \mathbf{l}_\perp^2\right]}.$$

We drop this correction because it does not give a simple pole. As a result, we obtain

$$\delta\Delta_b(8) = 0. \tag{D.7}$$

### D.2.9 Feynman diagram BV2-9

From the vertex correction in Table 6 BV2-9, we find a renormalization factor as

$$\delta\Delta_b(9) = -\frac{4g^2\Delta_f\Delta_b}{N}\int\frac{d^{d+1}k\,d^{d+1}l}{(2\pi)^{2d+1}}\delta(k_0)\frac{\left[\delta_{\mathbf{k+l}}\delta_{\mathbf{l}}\delta_{-\mathbf{l}} - \mathbf{L}\cdot(\mathbf{K+L})\delta_{\mathbf{l}} - \mathbf{L}\cdot(\mathbf{K+L})\delta_{-\mathbf{l}} - \mathbf{L}^2\delta_{\mathbf{k+l}}\right]\delta_{\mathbf{k}}}{\left[\delta_{\mathbf{k+l}}^2 + (\mathbf{K+L})^2\right]\left[\delta_{\mathbf{k}}^2 + \mathbf{K}^2\right]\left[\delta_{\mathbf{l}}^2 + \mathbf{L}^2\right]\left[\delta_{-\mathbf{l}}^2 + \mathbf{L}^2\right]\left[l_y^2 + g^2B_d\frac{|\mathbf{L}|^{d-1}}{|l_y|}\right]}.$$

Integrating over $k_x$, $k_y$, and $l_x$, we have

$$\delta\Delta_b(9) = \frac{g^2\Delta_f\Delta_b}{8N}\int\frac{d\mathbf{k}_\perp d\mathbf{L}\,dl_y}{(2\pi)^{2d-2}}\frac{|\mathbf{L}|}{|l_y|\left[l_y^4 + \mathbf{L}^2\right]\left[l_y^2 + g^2B_d\frac{|\mathbf{L}|^{d-1}}{|l_y|}\right]},$$

where the integration is the same with $\delta\Delta_b(8)$. We may ignore the $l_y^4$ term in the fermionic part since it would give rise to subleading terms in $g$. Integrating over $l_y$, we obtain

$$\delta\Delta_b(9) = \frac{g^{2/3}\Delta_f\Delta_b}{12\sqrt{3}B_d^{2/3}N}\int\frac{d\mathbf{k}_\perp d\mathbf{L}}{(2\pi)^{2d-3}}\frac{1}{|\mathbf{L}|^{\frac{2d+1}{3}}}.$$

We drop this correction because it does not give a simple pole. As a result, we obtain

$$\delta\Delta_b(9) = 0. \tag{D.8}$$

### D.2.10 Feynman diagram BV2-10

From the vertex correction in Table 6 BV2-10, we find a renormalization factor as

$$\delta\Delta_b(10) = -2\Delta_f^2\Delta_b \int \frac{d^dk\, d^dl}{(2\pi)^{2d}} \frac{[\delta_{\mathbf{k+1}}\delta_{\mathbf{k}} + \mathbf{k}_\perp \cdot (\mathbf{k}_\perp + \mathbf{1}_\perp)](\delta_{\mathbf{l}}\delta_{-\mathbf{l}} - \mathbf{l}_\perp^2) + 2l_y^2[\mathbf{k}_\perp \cdot \mathbf{1}_\perp \delta_{\mathbf{k+1}} - \mathbf{1}_\perp \cdot (\mathbf{k}_\perp + \mathbf{1}_\perp)\delta_{\mathbf{k}}]}{\left[\delta_{\mathbf{k+1}}^2 + (\mathbf{k}_\perp + \mathbf{1}_\perp)^2\right]\left[\delta_{\mathbf{k}}^2 + \mathbf{k}_\perp^2\right]\left[\delta_{\mathbf{l}}^2 + \mathbf{l}_\perp^2\right]\left[\delta_{-\mathbf{l}}^2 + \mathbf{l}_\perp^2\right]} \,.$$

Integrating over $k_x$, we get

$$\delta\Delta_b(10) = -\Delta_f^2\Delta_b \int \frac{d\mathbf{k}_\perp dk_y d^dl}{(2\pi)^{2d-1}} \frac{1}{\left[(\delta_{\mathbf{l}} + 2k_y l_y)^2 + (|\mathbf{k}_\perp + \mathbf{1}_\perp| + |\mathbf{k}_\perp|)^2\right]\left[\delta_{\mathbf{l}}^2 + \mathbf{l}_\perp^2\right]\left[\delta_{-\mathbf{l}}^2 + \mathbf{l}_\perp^2\right]}$$
$$\times \left[(\delta_{\mathbf{l}}\delta_{-\mathbf{l}} - \mathbf{l}_\perp^2)(|\mathbf{k}_\perp + \mathbf{1}_\perp| + |\mathbf{k}_\perp|)\left(1 + \frac{(\mathbf{k}_\perp + \mathbf{1}_\perp)\cdot\mathbf{k}_\perp}{|\mathbf{k}_\perp + \mathbf{1}_\perp||\mathbf{k}_\perp|}\right) + 2l_y^2(\delta_{\mathbf{l}} + 2k_y l_y)|\mathbf{1}_\perp|\left(\frac{\mathbf{k}_\perp \cdot \mathbf{1}_\perp}{|\mathbf{k}_\perp||\mathbf{1}_\perp|} + \frac{(\mathbf{k}_\perp + \mathbf{1}_\perp)\cdot\mathbf{1}_\perp}{|\mathbf{k}_\perp + \mathbf{1}_\perp||\mathbf{1}_\perp|}\right)\right].$$

Integrating over $k_y$, we obtain

$$\delta\Delta_b(10) = -\frac{\Delta_f^2\Delta_b}{4} \int \frac{d\mathbf{k}_\perp d^dl}{(2\pi)^{2d-2}} \frac{\delta_{\mathbf{l}}\delta_{-\mathbf{l}} - \mathbf{l}_\perp^2}{|l_y|\left[\delta_{\mathbf{l}}^2 + \mathbf{l}_\perp^2\right]\left[\delta_{-\mathbf{l}}^2 + \mathbf{l}_\perp^2\right]}\left(1 + \frac{(\mathbf{k}_\perp + \mathbf{1}_\perp)\cdot\mathbf{k}_\perp}{|\mathbf{k}_\perp + \mathbf{1}_\perp||\mathbf{k}_\perp|}\right).$$

Integrating over $l_x$, we have

$$\delta\Delta_b(10) = \frac{\Delta_f^2\Delta_b}{8} \int \frac{d\mathbf{k}_\perp d\mathbf{l}_\perp dl_y}{(2\pi)^{2d-3}} \frac{|\mathbf{l}_\perp|}{|l_y|\left[l_y^4 + \mathbf{l}_\perp^2\right]}\left(1 + \frac{(\mathbf{k}_\perp + \mathbf{1}_\perp)\cdot\mathbf{k}_\perp}{|\mathbf{k}_\perp + \mathbf{1}_\perp||\mathbf{k}_\perp|}\right).$$

We drop this correction because it would give only a double pole. As a result, we obtain

$$\delta\Delta_b(10) = 0\,. \tag{D.9}$$

### D.2.11 Feynman diagram BV2-11

From the vertex correction in Table 6 BV2-11, we find a renormalization factor as

$$\delta\Delta_b(11) = -\frac{2g^4\Delta_b}{N^2} \int \frac{d^{d+1}k\, d^{d+1}l}{(2\pi)^{2d+2}} \frac{\mathcal{N}}{\left[\delta_{\mathbf{k+1}}^2 + (\mathbf{K}+\mathbf{L})^2\right]\left[\delta_{-\mathbf{k-l}}^2 + (\mathbf{K}+\mathbf{L})^2\right]\left[\delta_{\mathbf{k}}^2 + \mathbf{K}^2\right]\left[\delta_{-\mathbf{l}}^2 + \mathbf{L}^2\right]} D_1(k)D_1(l)\,,$$

where $\mathcal{N}$ is given by

$$\mathcal{N} = \delta_{\mathbf{k+1}}\delta_{-\mathbf{k-l}}\delta_{\mathbf{k}}\delta_{-\mathbf{l}} + \mathbf{K}\cdot\mathbf{L}(\mathbf{K}+\mathbf{L})^2 - (\delta_{\mathbf{k+1}} + \delta_{-\mathbf{k-l}})\delta_{-\mathbf{l}}\mathbf{K}\cdot(\mathbf{K}+\mathbf{L})$$
$$- (\delta_{\mathbf{k+1}} + \delta_{-\mathbf{k-l}})\delta_{\mathbf{k}}\mathbf{L}\cdot(\mathbf{K}+\mathbf{L}) - \delta_{\mathbf{k}}\delta_{-\mathbf{l}}(\mathbf{K}+\mathbf{L})^2 - \delta_{\mathbf{k+1}}\delta_{-\mathbf{k-l}}\mathbf{K}\cdot\mathbf{L}\,.$$

We may ignore $k_y$ and $l_y$ in the fermionic part since they would give rise to subleading terms in $g$. Then, we have

$$\delta\Delta_b(11) = -\frac{2g^4\Delta_b}{N^2} \int \frac{d^{d+1}k\, d^{d+1}l}{(2\pi)^{2d+2}} \frac{\mathcal{N}'}{\left[(k_x + l_x)^2 + (\mathbf{K}+\mathbf{L})^2\right]^2\left[k_x^2 + \mathbf{K}^2\right]\left[l_x^2 + \mathbf{L}^2\right]} D_1(k)D_1(l)\,,$$

where $\mathcal{N}'$ is given by

$$\mathcal{N}' = (k_x + l_x)^2 k_x l_x + \mathbf{K}\cdot\mathbf{L}(\mathbf{K}+\mathbf{L})^2 + k_x l_x(\mathbf{K}+\mathbf{L})^2 + (k_x + l_x)^2\mathbf{K}\cdot\mathbf{L}\,.$$

Integrating over $k_x$ and $l_x$, we obtain

$$\delta\Delta_b(11) = \frac{g^4\Delta_b}{2N^2} \int \frac{d\mathbf{K}dk_y d\mathbf{L}dl_y}{(2\pi)^{2d}} \frac{1}{|\mathbf{K}+\mathbf{L}|(|\mathbf{K}| + |\mathbf{K}+\mathbf{L}| + |\mathbf{L}|)}\left(1 - \frac{\mathbf{K}\cdot\mathbf{L}}{|\mathbf{K}||\mathbf{L}|}\right) D_1(k)D_1(l)\,.$$

Integrating over $k_y$ and $l_y$, we get

$$\delta\Delta_b(11) = \frac{2g^{8/3}\Delta_b}{27B_d^{2/3}N^2}\int\frac{d\mathbf{K}d\mathbf{L}}{(2\pi)^{2d-2}}\frac{1}{|\mathbf{K}|^{\frac{d-1}{3}}|\mathbf{L}|^{\frac{d-1}{3}}|\mathbf{K}+\mathbf{L}|(|\mathbf{K}|+|\mathbf{K}+\mathbf{L}|+|\mathbf{L}|)}\left(1-\frac{\mathbf{K}\cdot\mathbf{L}}{|\mathbf{K}||\mathbf{L}|}\right).$$

Introducing coordinates as $\mathbf{K}\cdot\mathbf{L} = KL\cos\theta$ and scaling $K$ as $K = Lk$, we have

$$\delta\Delta_b(11) = \frac{2\Omega'g^{8/3}\Delta_b}{27B_d^{2/3}N^3}\int_{|\mathbf{P}|}^{\infty}dL\,L^{\frac{4d-13}{3}}\int_0^{\infty}dk\,k^{\frac{2d-5}{3}}\int_0^{\pi}d\theta\sin^{d-3}\theta\frac{1-\cos\theta}{\sqrt{1+k^2+2k\cos\theta}\left(1+k+\sqrt{1+k^2+2k\cos\theta}\right)},$$

where $\Omega' \equiv \frac{4}{(4\pi)^{d-1}\sqrt{\pi}\Gamma(\frac{d-1}{2})\Gamma(\frac{d-2}{2})}$. We find an $\epsilon$ pole from the $L$ integral as $\int_{|\mathbf{P}|}^{\infty}dL\,L^{\frac{4d-13}{3}} = \frac{3}{4\epsilon}+\mathcal{O}(1)$. The remaining integral is numerically done as

$$\int_0^{\infty}dk\,k^{\frac{2d-5}{3}}\int_0^{\pi}d\theta\sin^{d-3}\theta\frac{1-\cos\theta}{\sqrt{1+k^2+2k\cos\theta}\left(1+k+\sqrt{1+k^2+2k\cos\theta}\right)} = \frac{\sqrt{\pi}\Gamma(\frac{d-2}{2})}{\Gamma(\frac{d-1}{2})}(2.264).$$

As a result, we obtain

$$\delta\Delta_b(11) = (13.58)\Delta_b\frac{\tilde{g}^2}{\epsilon}. \tag{D.10}$$

### D.2.12 Feynman diagram BV2-12

From the vertex correction in Table 6 BV2-12, we find a renormalization factor as

$$\delta\Delta_b(12) = -4\Delta_f^2\Delta_b\int\frac{d^dkd^dl}{(2\pi)^{2d}}\frac{[\delta_\mathbf{k}\delta_{\mathbf{k+1}}-\mathbf{k}_\perp\cdot(\mathbf{k}_\perp+\mathbf{1}_\perp)](\delta_\mathbf{l}\delta_{-\mathbf{l}}-\mathbf{l}_\perp^2)-2l_y^2[\delta_\mathbf{k}\mathbf{l}_\perp\cdot(\mathbf{k}_\perp+\mathbf{1}_\perp)+\delta_{\mathbf{k+1}}\mathbf{k}_\perp\cdot\mathbf{1}_\perp]}{[\delta_\mathbf{k}^2+\mathbf{k}_\perp^2][\delta_{\mathbf{k+1}}^2+(\mathbf{k}_\perp+\mathbf{1}_\perp)^2][\delta_\mathbf{l}^2+\mathbf{l}_\perp^2][\delta_{-\mathbf{l}}^2+\mathbf{l}_\perp^2]}.$$

Integrating over $k_x$, we obtain

$$\delta\Delta_b(12) = -\frac{2\Delta_f^2\Delta_b}{N^3}\int\frac{d\mathbf{k}_\perp dk_yd^dl}{(2\pi)^{2d-1}}\frac{1}{[(\delta_\mathbf{l}+2k_yl_y)^2+(|\mathbf{k}_\perp+\mathbf{1}_\perp|+|\mathbf{k}_\perp|)^2][\delta_\mathbf{l}^2+\mathbf{l}_\perp^2][\delta_{-\mathbf{l}}^2+\mathbf{l}_\perp^2]}$$
$$\times\left[(\delta_\mathbf{l}\delta_{-\mathbf{l}}-\mathbf{l}_\perp^2)(|\mathbf{k}_\perp+\mathbf{1}_\perp|+|\mathbf{k}_\perp|)\left(1-\frac{(\mathbf{k}_\perp+\mathbf{1}_\perp)\cdot\mathbf{k}_\perp}{|\mathbf{k}_\perp+\mathbf{1}_\perp||\mathbf{k}_\perp|}\right)+2l_y^2(\delta_\mathbf{l}+2k_yl_y)|\mathbf{l}_\perp|\left(-\frac{\mathbf{k}_\perp\cdot\mathbf{l}_\perp}{|\mathbf{k}_\perp||\mathbf{l}_\perp|}+\frac{(\mathbf{k}_\perp+\mathbf{1}_\perp)\cdot\mathbf{l}_\perp}{|\mathbf{k}_\perp+\mathbf{1}_\perp||\mathbf{l}_\perp|}\right)\right].$$

Integrating over $k_y$, we have

$$\delta\Delta_b(12) = -\frac{\Delta_f^2\Delta_b}{2}\int\frac{d\mathbf{k}_\perp d^dl}{(2\pi)^{2d-2}}\frac{\delta_\mathbf{l}\delta_{-\mathbf{l}}-\mathbf{l}_\perp^2}{|l_y|[\delta_\mathbf{l}^2+\mathbf{l}_\perp^2][\delta_{-\mathbf{l}}^2+\mathbf{l}_\perp^2]}\left(1-\frac{(\mathbf{k}_\perp+\mathbf{1}_\perp)\cdot\mathbf{k}_\perp}{|\mathbf{k}_\perp+\mathbf{1}_\perp||\mathbf{k}_\perp|}\right).$$

Integrating over $l_x$, we get

$$\delta\Delta_b(12) = \frac{\Delta_f^2\Delta_b}{4}\int\frac{d\mathbf{k}_\perp d\mathbf{l}_\perp dl_y}{(2\pi)^{2d-3}}\frac{|\mathbf{l}_\perp|}{|l_y|[l_y^4+\mathbf{l}_\perp^2]}\left(1-\frac{(\mathbf{k}_\perp+\mathbf{1}_\perp)\cdot\mathbf{k}_\perp}{|\mathbf{k}_\perp+\mathbf{1}_\perp||\mathbf{k}_\perp|}\right).$$

We drop this correction because it would give only a double pole. As a result, we obtain

$$\delta\Delta_b(12) = 0. \tag{D.11}$$

### D.2.13 Feynman diagram BV2-13

From the vertex correction in Table 6 BV2-13, we find a renormalization factor as

$$\delta\Delta_b(13) = -4\Delta_f\Delta_b^2\int\frac{d^dkd^dl}{(2\pi)^{2d}}\frac{[\delta_\mathbf{k}\delta_{\mathbf{k+1}}-\mathbf{k}_\perp\cdot(\mathbf{k}_\perp+\mathbf{1}_\perp)](\delta_\mathbf{l}\delta_{-\mathbf{l}}-\mathbf{l}_\perp^2)-2l_y^2[\delta_\mathbf{k}\mathbf{l}_\perp\cdot(\mathbf{k}_\perp+\mathbf{1}_\perp)+\delta_{\mathbf{k+1}}\mathbf{k}_\perp\cdot\mathbf{1}_\perp]}{[\delta_\mathbf{k}^2+\mathbf{k}_\perp^2][\delta_{\mathbf{k+1}}^2+(\mathbf{k}_\perp+\mathbf{1}_\perp)^2][\delta_\mathbf{l}^2+\mathbf{l}_\perp^2][\delta_{-\mathbf{l}}^2+\mathbf{l}_\perp^2]}.$$

The integration is the same with $\delta\Delta_b(12)$. As a result, we obtain

$$\delta\Delta_b(13) = 0.$$

### D.2.14 Feynman diagram BV2-14

From the vertex correction in Table 6 BV2-14, we find a renormalization factor as

$$\delta\Delta_b(14) = -\frac{4g^2\Delta_f\Delta_b}{N}\int\frac{d^{d+1}kd^{d+1}l}{(2\pi)^{2d+1}}\delta(k_0)\frac{\left[\delta_{\mathbf{k}}\delta_{\mathbf{k+l}} - \mathbf{K}\cdot(\mathbf{K+L})\right](\delta_l\delta_{-l} - \mathbf{L}^2) - 2l_y^2\left[\delta_{\mathbf{k}}\mathbf{L}\cdot(\mathbf{K+L}) + \delta_{\mathbf{k+l}}\mathbf{K}\cdot\mathbf{L}\right]}{\left[\delta_{\mathbf{k}}^2 + \mathbf{K}^2\right]\left[\delta_{\mathbf{k+l}}^2 + (\mathbf{K+L})^2\right]\left[\delta_l^2 + \mathbf{L}^2\right]\left[\delta_{-l}^2 + \mathbf{L}^2\right]\left[k_y^2 + g^2B_d\frac{|\mathbf{K}|^{d-1}}{|k_y|}\right]}.$$

Integrating over $k_x$, we have

$$\delta\Delta_b(14) = -\frac{2g^2\Delta_f\Delta_b}{N}\int\frac{d\mathbf{k}dk_yd^{d+1}l}{(2\pi)^{2d}}\delta(l_0)\frac{1}{\left[\delta_l^2 + (|\mathbf{K+L}| + |\mathbf{K}|)^2\right]\left[\delta_l^2 + \mathbf{L}^2\right]\left[\delta_{-l}^2 + \mathbf{L}^2\right]\left[k_y^2 + g^2B_d\frac{|\mathbf{K}|^{d-1}}{|k_y|}\right]}$$

$$\times\left[(\delta_l\delta_{-l} - \mathbf{L}^2)(|\mathbf{K+L}| + |\mathbf{K}|)\left(1 - \frac{(\mathbf{K+L})\cdot\mathbf{K}}{|\mathbf{K+L}||\mathbf{K}|}\right) + 2l_y^2\delta_l|\mathbf{L}|\left(-\frac{\mathbf{K}\cdot\mathbf{L}}{|\mathbf{K}||\mathbf{L}|} + \frac{(\mathbf{K+L})\cdot\mathbf{L}}{|\mathbf{K+L}||\mathbf{L}|}\right)\right],$$

where we have neglected the $k_y l_y$ term since it would give rise to subleading terms in $g$. Integrating over $l_x$, $l_y$ and $k_y$, we obtain

$$\delta\Delta_b(14) = \frac{g^{4/3}\Delta_f\Delta_b}{6\sqrt{3}B_d^{1/3}N}\int\frac{d\mathbf{K}d\mathbf{L}}{(2\pi)^{2d-3}}\frac{\delta(l_0)}{|\mathbf{k}|^{(d-1)/3}}\left[\frac{\sqrt{2}}{(|\mathbf{K}| + |\mathbf{K+L}| + |\mathbf{L}|)\sqrt{|\mathbf{L}|}}\left(1 - \frac{(\mathbf{K+L})\cdot\mathbf{K}}{|\mathbf{K+L}||\mathbf{K}|}\right)\right.$$

$$\left. - \frac{\sqrt{|\mathbf{K}| + |\mathbf{K+L}| + |\mathbf{L}|} - \sqrt{2|\mathbf{L}|}}{(|\mathbf{K}| + |\mathbf{K+L}|)^2 - |\mathbf{L}|^2}\left(1 - \frac{(\mathbf{K+L})\cdot\mathbf{K}}{|\mathbf{K+L}||\mathbf{K}|} - \frac{\mathbf{K}\cdot\mathbf{L}}{|\mathbf{K}||\mathbf{L}|} + \frac{(\mathbf{K+L})\cdot\mathbf{L}}{|\mathbf{K+L}||\mathbf{L}|}\right)\right].$$

Introducing coordinates as $\mathbf{K}\cdot\mathbf{L} = KL\cos\theta$ and scaling variables as $L = Kl$ and $k_0 = Kk$, we have

$$\delta\Delta_b(14) = \frac{\Omega g^{4/3}\Delta_f\Delta_b}{6\pi\sqrt{3}B_d^{1/3}N}\int_{p_0}^{\infty}dK K^{\frac{10d-31}{6}}\int_0^{\infty}dl\,l^{d-3}\int_0^{\infty}\frac{dk}{(1+k^2)^{\frac{d-1}{6}}}\int_0^{\pi}d\theta\sin^{d-4}\theta$$

$$\times\left[\frac{\sqrt{2}}{\sqrt{l}(\sqrt{1+k^2}+l+\eta)}\left(1 - \frac{1+k^2+l\cos\theta}{\sqrt{1+k^2}\eta}\right) - \frac{(\sqrt{1+k^2}+l+\eta)^{1/2} - \sqrt{2l}}{(\sqrt{1+k^2}+\eta)^2 - l^2}\left(1 - \frac{\cos\theta}{\sqrt{1+k^2}}\right)\left(1 + \frac{l-\sqrt{1+k^2}}{\eta}\right)\right],$$

where $\eta = \sqrt{1+k^2+l^2+2l\cos\theta}$. We find an $\epsilon$ pole from the $L$ integral as $\int_{p_0}^{\infty}dL L^{\frac{10d-31}{6}} = \frac{3}{5\epsilon} + \mathcal{O}(1)$. The remaining integral are done numerically as

$$\int_0^{\infty}dl\,l^{d-3}\int_0^{\infty}\frac{dk}{(1+k^2)^{\frac{d-1}{6}}}\int_0^{\pi}d\theta\sin^{d-4}\theta\left[\frac{\sqrt{2}}{\sqrt{l}(\sqrt{1+k^2}+l+\eta)}\left(1 - \frac{1+k^2+l\cos\theta}{\sqrt{1+k^2}\eta}\right)\right.$$

$$\left. - \frac{(\sqrt{1+k^2}+l+\eta)^{1/2} - \sqrt{2l}}{(\sqrt{1+k^2}+\eta)^2 - l^2}\left(1 - \frac{\cos\theta}{\sqrt{1+k^2}}\right)\left(1 + \frac{l-\sqrt{1+k^2}}{\eta}\right)\right] = (0.6926)\frac{\sqrt{\pi}\Gamma(\frac{d-3}{2})}{\Gamma(\frac{d-2}{2})}.$$

As a result, we obtain

$$\delta\Delta_b(14) = (1.408)\Delta_b\frac{\tilde{\Delta}_f\tilde{g}}{\epsilon}.$$

### D.2.15 Feynman diagram BV2-15

From the vertex correction in Table 6 BV2-15, we find a renormalization factor as

$$\delta\Delta_b(15) = -\frac{4g^2\Delta_f\Delta_b}{N}\int\frac{d^{d+1}kd^{d+1}l}{(2\pi)^{2d+1}}\delta(k_0)\frac{\left[\delta_{\mathbf{k}}\delta_{\mathbf{k+l}} - \mathbf{K}\cdot(\mathbf{K+L})\right](\delta_l\delta_{-l} - \mathbf{L}^2) - 2l_y^2\left[\delta_{\mathbf{k}}\mathbf{L}\cdot(\mathbf{K+L}) + \delta_{\mathbf{k+l}}\mathbf{K}\cdot\mathbf{L}\right]}{\left[\delta_{\mathbf{k}}^2 + \mathbf{K}^2\right]\left[\delta_{\mathbf{k+l}}^2 + (\mathbf{K+L})^2\right]\left[\delta_l^2 + \mathbf{L}^2\right]\left[\delta_{-l}^2 + \mathbf{L}^2\right]\left[l_y^2 + g^2B_d\frac{|\mathbf{L}|^{d-1}}{|l_y|}\right]}.$$

Integrating over $k_x$, $k_y$ and $l_x$, we get

$$\delta\Delta_b(15) = \frac{g^2\Delta_f\Delta_b}{4N}\int\frac{d\mathbf{K}d\mathbf{L}dl_y}{(2\pi)^{2d-2}}\delta(k_0)\frac{|\mathbf{L}|}{|l_y|\left[l_y^4 + \mathbf{L}^2\right]\left[l_y^2 + g^2B_d\frac{|\mathbf{L}|^{d-1}}{|l_y|}\right]}\left(1 - \frac{(\mathbf{K+L})\cdot\mathbf{K}}{|\mathbf{K+L}||\mathbf{K}|}\right),$$

where the integration is the same with $\delta\Delta_b(12)$. We may neglect the $l_y^4$ term since it would give rise to subleading terms in $g$. Integrating over $l_y$, we have

$$\delta\Delta_b(15) = \frac{g^{2/3}\Delta_f\Delta_b}{6\sqrt{3}B_d^{2/3}N}\int\frac{d\mathbf{K}d\mathbf{L}}{(2\pi)^{2d-3}}\frac{\delta(k_0)}{|\mathbf{L}|^{\frac{2d+1}{3}}}\left(1-\frac{(\mathbf{K}+\mathbf{L})\cdot\mathbf{K}}{|\mathbf{K}+\mathbf{L}||\mathbf{K}|}\right).$$

Introducing coordinates as $\mathbf{K}\cdot\mathbf{L} = KL\cos\theta$ and scaling variables as $K = Lk$ and $l_0 = Ll$, we obtain

$$\delta\Delta_b(15) = \frac{\Omega g^{2/3}\Delta_f\Delta_b}{6\pi\sqrt{3}B_d^{2/3}N}\int_{|\mathbf{P}|}^{\infty}dL L^{\frac{4d-13}{3}}\int_0^{\infty}dk k^{d-3}\int_0^{\infty}dl\int_0^{\pi}d\theta\sin^{d-4}\theta\frac{1}{(1+l^2)^{\frac{2d+1}{6}}}\left(1-\frac{k+\cos\theta}{\sqrt{1+k^2+l^2+2k\cos\theta}}\right).$$

We find an $\epsilon$ pole from the $L$ integral as $\int_{|\mathbf{P}|}^{\infty}dL L^{\frac{4d-13}{3}} = \frac{3}{4\epsilon} + \mathcal{O}(1)$. The remaining integral can be done numerically as

$$\int_0^{\infty}dk k^{d-3}\int_0^{\infty}dl\int_0^{\pi}d\theta\sin^{d-4}\theta\frac{1}{(1+l^2)^{\frac{2d+1}{6}}}\left(1-\frac{k+\cos\theta}{\sqrt{1+k^2+l^2+2k\cos\theta}}\right) = (4.935)\frac{\sqrt{\pi}\Gamma(\frac{d-3}{2})}{\Gamma(\frac{d-2}{2})}.$$

As a result, we obtain

$$\delta\Delta_b(15) = (8.323)\Delta_b\frac{\tilde{\Delta}_f\sqrt{\tilde{g}}}{\sqrt{N}\epsilon}. \tag{D.12}$$

### D.2.16 Feynman diagram BV2-16

From the vertex correction in Table 6 BV2-16, we find a renormalization factor as

$$\delta\Delta_b(16) = -\frac{4g^2\Delta_b^2}{N}\int\frac{d^{d+1}kd^{d+1}l}{(2\pi)^{2d+1}}\delta(k_0)\frac{\left[\delta_{\mathbf{k}}\delta_{\mathbf{k+l}} - \mathbf{K}\cdot(\mathbf{K}+\mathbf{L})\right](\delta_l\delta_{-l} - \mathbf{L}^2) - 2l_y^2\left[\delta_{\mathbf{k}}\mathbf{L}\cdot(\mathbf{K}+\mathbf{L}) + \delta_{\mathbf{k+l}}\mathbf{K}\cdot\mathbf{L}\right]}{\left[\delta_{\mathbf{k}}^2 + \mathbf{K}^2\right]\left[\delta_{\mathbf{k+l}}^2 + (\mathbf{K}+\mathbf{L})^2\right]\left[\delta_l^2 + \mathbf{L}^2\right]\left[\delta_{-l}^2 + \mathbf{L}^2\right]\left[l_y^2 + g^2B_d\frac{|\mathbf{L}|^{d-1}}{|l_y|}\right]}.$$

The integration is the same with $\delta\Delta_b(15)$. As a result, we obtain

$$\delta\Delta_b(16) = 0.$$

### D.2.17 Feynman diagram BV2-17

From the vertex correction in Table 6 BV2-17, we find a renormalization factor as

$$\delta\Delta_b(17) = -\frac{4g^4\Delta_b}{N^2}\int\frac{d^{d+1}kd^{d+1}l}{(2\pi)^{2d+2}}\frac{\left[\delta_{\mathbf{k}}\delta_{\mathbf{k+l}} - \mathbf{K}\cdot(\mathbf{K}+\mathbf{L})\right](\delta_l\delta_{-l} - \mathbf{L}^2) - 2l_y^2\left[\delta_{\mathbf{k}}\mathbf{L}\cdot(\mathbf{K}+\mathbf{L}) + \delta_{\mathbf{k+l}}\mathbf{K}\cdot\mathbf{L}\right]}{\left[\delta_{\mathbf{k}}^2 + \mathbf{K}^2\right]\left[\delta_{\mathbf{k+l}}^2 + (\mathbf{K}+\mathbf{L})^2\right]\left[\delta_l^2 + \mathbf{L}^2\right]\left[\delta_{-l}^2 + \mathbf{L}^2\right]}D_1(k)D_1(l).$$

We may ignore $k_y$ and $l_y$ in the fermionic part since they would give subleading terms in $g$. Then, we have

$$\delta\Delta_b(17) = \frac{4g^4\Delta_b}{N^2}\int\frac{d^{d+1}kd^{d+1}l}{(2\pi)^{2d+2}}\frac{k_x(k_x+l_x) - \mathbf{K}\cdot(\mathbf{K}+\mathbf{L})}{\left[k_x^2 + \mathbf{K}^2\right]\left[(k_x+l_x)^2 + (\mathbf{K}+\mathbf{L})^2\right]\left[l_x^2 + \mathbf{L}^2\right]}D_1(k)D_1(l).$$

Integrating over $k_x$ and $l_x$, we get

$$\delta\Delta_b(17) = \frac{g^4\Delta_b}{N^2}\int\frac{d\mathbf{K}dk_y\mathbf{L}dl_y}{(2\pi)^{2d}}\frac{1}{|\mathbf{L}|(|\mathbf{K}| + |\mathbf{K}+\mathbf{L}| + |\mathbf{L}|)}\left(1 - \frac{(\mathbf{K}+\mathbf{L})\cdot\mathbf{K}}{|\mathbf{K}+\mathbf{K}||\mathbf{K}|}\right)D_1(k)D_1(l).$$

Integrating over $k_y$ and $l_y$, we obtain

$$\delta\Delta_b(17) = \frac{4g^{8/3}\Delta_b}{27B_d^{2/3}N^2}\int\frac{d\mathbf{k}d\mathbf{l}}{(2\pi)^{2d-2}}\frac{1}{|\mathbf{K}|^{\frac{d-1}{3}}|\mathbf{L}|^{\frac{d+2}{3}}(|\mathbf{K}| + |\mathbf{K}+\mathbf{L}| + |\mathbf{L}|)}\left(1 - \frac{(\mathbf{K}+\mathbf{L})\cdot\mathbf{K}}{|\mathbf{K}+\mathbf{L}||\mathbf{K}|}\right).$$

Introducing coordinates as $\mathbf{K} \cdot \mathbf{L} = KL\cos\theta$ and scaling variables as $K = Lk$, we have

$$\delta\Delta_b(17) = \frac{4\Omega' g^{8/3}\Delta_b}{27 B_d^{2/3} N^2} \int_{|\mathbf{P}|}^{\infty} dL\, L^{\frac{4d-13}{3}} \int_0^{\infty} dk\, k^{\frac{2d-5}{3}} \int_0^{\pi} d\theta \sin^{d-3}\theta \frac{1}{1 + k + \sqrt{1 + k^2 + 2k\cos\theta}} \left(1 - \frac{k + \cos\theta}{\sqrt{1 + k^2 + 2k\cos\theta}}\right).$$

We find an $\epsilon$ pole from the $L$ integral as $\int_{|\mathbf{P}|}^{\infty} dL\, L^{\frac{4d-13}{3}} = \frac{3}{4\epsilon} + \mathcal{O}(1)$. The remaining integral are done numerically as

$$\int_0^{\infty} dk\, k^{\frac{2d-5}{3}} \int_0^{\pi} d\theta \sin^{d-3}\theta \frac{1}{1 + k + \sqrt{1 + k^2 + 2k\cos\theta}} \left(1 - \frac{k + \cos\theta}{\sqrt{1 + k^2 + 2k\cos\theta}}\right) = \frac{\sqrt{\pi}\,\Gamma(\frac{d-2}{2})}{\Gamma(\frac{d-1}{2})}(0.4213).$$

The remaining integration is the same with $\delta\Delta_b(11)$. As a result, we obtain

$$\delta\Delta_b(17) = (5.056)\Delta_b \frac{\tilde{g}^2}{\epsilon}. \tag{D.13}$$

# E Derivation of critical exponents

## E.1 Critical exponents for fermion and boson fields

Here, we derive the Callan-Symanzik equations and define the critical exponents for the correlation functions. We first consider the following correlation function:

$$G^{(m,n)}(\{k_i\}, \mathbf{F}) \equiv \left\langle \bar{\Psi}(k_1) \cdots \Psi(k_m) \Phi(k_{m+1}) \cdots \right\rangle, \tag{E.1}$$

where $\mathbf{F} = (\tilde{g}, \tilde{\Delta}_f, \tilde{\Delta}_b)$. In a renormalized theory, the correlation function is expressed as

$$G^{(m,n)}_{ren.}(\{k_{i,r}\}, \mu, \mathbf{F}_r) = \mu^{-[G]} Z_\Psi^{-m/2} Z_\Phi^{-n/2} G^{(m,n)}_{bare}(\{k_i\}, \mathbf{F}), \tag{E.2}$$

where $G^{(m,n)}_{ren.}(\{k_{i,r}\}, \mu, \mathbf{F}_r)$ and $G^{(m,n)}_{bare}(\{k_i\}, \mathbf{F})$ represent the renormalized and bare correlation functions, respectively. Here, $\{k_{i,r}\}$ are scaled momenta, $\mathbf{F}_r$ are renormalized coupling constants, and $[G] = m[\Psi] + n[\Phi] + z + \bar{z}(d-2) + \frac{3}{2}$. Using the fact $\frac{dG^{(m,n)}_{bare}}{d\mu} = 0$, we obtain the following Callan-Symanzik equation for $G^{(m,n)}_{ren.}(\{k_{i,r}\}, \mu, \mathbf{F}_r)$:

$$\left[\sum_i k_i \cdot \nabla_{k_i} - \beta_\mathbf{F} \cdot \nabla_\mathbf{F} - m([\Psi] + \gamma_\Psi) - n([\Phi] + \gamma_\Phi) - D_{sc}\right] G^{(m,n)}_{ren.}(\{k_i\}, \mu, \mathbf{F}) = 0, \tag{E.3}$$

where the derivative expressions are defined as $k \cdot \nabla_k \equiv z k_0 \frac{\partial}{\partial k_0} + \bar{z}\mathbf{k}_\perp \cdot \nabla_{\mathbf{k}_\perp} + \delta_\mathbf{k} \frac{\partial}{\partial \delta_\mathbf{k}}$ and $\nabla_\mathbf{F} \equiv (\frac{2}{3}\frac{\partial}{\partial \tilde{g}}, \frac{\partial}{\partial \tilde{\Delta}_f}, \frac{\partial}{\partial \tilde{\Delta}_b})$, and the beta function is written in the vector form as $\beta_\mathbf{F} \equiv (\beta_{\tilde{g}}, \beta_{\tilde{\Delta}_f}, \beta_{\tilde{\Delta}_b})$, and $D_{sc} = z + \bar{z}(d-2) + \frac{3}{2}$. The critical exponents in the Callan-Symanzik equations are defined as

$$z = 1 + \frac{\partial \ln(Z_0/Z_2)}{\partial \ln\mu},$$

$$\bar{z} = 1 + \frac{\partial \ln(Z_1/Z_2)}{\partial \ln\mu},$$

$$\gamma_\Psi = \frac{\partial \ln Z_2}{\partial \ln\mu},$$

$$\gamma_\Phi = \frac{\partial \ln Z_3}{\partial \ln\mu}, \tag{E.4}$$

where $Z_{0,1,2,3}$ are the renormalization factors that relate the bare and scaled momenta as

$$k_{0,r} = \mu \frac{Z_0}{Z_2} k_0, \qquad \mathbf{k}_{\perp,r} = \mu \frac{Z_1}{Z_2} \mathbf{k}_\perp, \qquad k_{x,r} = \mu k_x, \qquad k_{y,r} = \mu^{\frac{1}{2}} k_y. \qquad \text{(E.5)}$$

We next consider another type of correlation function given by

$$G^{(m,n)}(\{k_i\}, \{\gamma_{\mu(j)}\}, \mathbf{F}) \equiv \left\langle \bar{\Psi}(k_1)\gamma_{\mu(1)}\Psi(k_2) \cdots \bar{\Psi}(k_{m-1})\gamma_{\mu(l)}\Psi(k_m)\phi(k_{m+1}) \cdots \right\rangle, \qquad \text{(E.6)}$$

where $\gamma_{\mu(j)}$ represent the gamma matrices for the corresponding coupling constants, i.e. $\gamma_{d-1}$ for $\tilde{g}$ and $\tilde{\Delta}_f$ or the identity matrix $I_2$ for $\tilde{\Delta}_b$. With a similar consideration for $G^{(m,n)}(\{k_i\}, \mathbf{F})$, we obtain the Callan-Symanzik equation for $G^{(m,n)}(\{k_i\}, \{\gamma_{\mu(j)}\}, \mathbf{F})$ as

$$\left[ \sum_i k_i \cdot \nabla_{k_i} - \beta_{\mathbf{F}} \cdot \nabla_{\mathbf{F}} - m\big([\Psi] + \gamma_\Psi\big) - n\big([\Phi] + \gamma_\Phi\big) - D_{sc} + \sum_{j=1}^l \gamma_{\mu(j)}^{ver} \right] G^{(n,m)}(\{k_i\}, \{\gamma_{\mu(j)}\}, \mu, \mathbf{F}) = 0. \tag{E.7}$$

Here, $\gamma_{\mu(j)}^{ver}$ $Z_{0,1,2,3}$ are the anomalous dimension of the coupling constants are defined as

$$\gamma_g = \frac{\partial \ln Z_g}{\partial \ln \mu},$$

$$\gamma_{\Delta_f} = \frac{\partial \ln Z_{\Delta_f}}{\partial \ln \mu},$$

$$\gamma_{\Delta_b} = \frac{\partial \ln Z_{\Delta_f}}{\partial \ln \mu}, \tag{E.8}$$

where $Z_g$, $Z_{\Delta_b}$, and $Z_{\Delta_f}$ are the renormalization factors that relate the bare and renormalized coupling constants as

$$Z_g g = \mu^{-\frac{\epsilon}{2}} (Z_0/Z_2)^{\frac{1}{2}} (Z_1/Z_2)^{\frac{d-2}{2}} Z_2 Z_3^{\frac{1}{2}} g_0,$$

$$Z_{\Delta_f} \Delta_f = \mu^{-\epsilon} (Z_1/Z_2)^{d-2} Z_2^2 \Delta_{f,0},$$

$$Z_{\Delta_b} \Delta_b = \mu^{-\epsilon} (Z_1/Z_2)^{d-2} Z_2^2 \Delta_{b,0}. \tag{E.9}$$

We finally compute the critical exponents for the correlation functions, which are defined in Eqs. (E.4) and (E.8), which are given in Eqs. (17) and (18) in the main text. We note that in the epsilon expansion the renormalization factors are given in the following forms:

$$Z_0 = 1 + \frac{A_0}{\epsilon} + \mathcal{O}\left(\frac{1}{\epsilon^2}\right),$$

$$Z_1 = 1 + \frac{A_1}{\epsilon} + \mathcal{O}\left(\frac{1}{\epsilon^2}\right),$$

$$Z_2 = 1 + \frac{A_2}{\epsilon} + \mathcal{O}\left(\frac{1}{\epsilon^2}\right),$$

$$Z_3 = 1 + \frac{A_3}{\epsilon} + \mathcal{O}\left(\frac{1}{\epsilon^2}\right),$$

$$Z_g = 1 + \frac{A_0}{\epsilon} + \mathcal{O}\left(\frac{1}{\epsilon^2}\right),$$

$$Z_{\Delta_f} = 1 + \frac{A_{\Delta_f}}{\epsilon} + \mathcal{O}\left(\frac{1}{\epsilon^2}\right),$$

$$Z_{\Delta_b} = 1 + \frac{A_{\Delta_b}}{\epsilon} + \mathcal{O}\left(\frac{1}{\epsilon^2}\right). \tag{E.10}$$

Inserting Eq. (E.10) into Eqs. (E.4) and (E.8) solving the resulting equations order by order in $\epsilon$, we obtain the following expressions:

$$\bar{z} = \left(1 + \mathbf{F} \cdot \nabla_{\mathbf{F}}(A_1 - A_2)\right)^{-1},$$
$$z = \bar{z}\left(1 - \mathbf{F} \cdot \nabla_{\mathbf{F}}(A_0 - A_1)\right),$$
$$\gamma_\Psi = -\frac{1}{2}\bar{z}\mathbf{F} \cdot \nabla_{\mathbf{F}} A_2,$$
$$\gamma_\Phi = -\frac{1}{2}\bar{z}\mathbf{F} \cdot \nabla_{\mathbf{F}} A_3,$$
$$\gamma_g = -\bar{z}\mathbf{F} \cdot \nabla_{\mathbf{F}} A_g,$$
$$\gamma_{\Delta_f} = -\bar{z}\mathbf{F} \cdot \nabla_{\mathbf{F}} A_{\Delta_f},$$
$$\gamma_{\Delta_b} = -\bar{z}\mathbf{F} \cdot \nabla_{\mathbf{F}} A_{\Delta_b}. \tag{E.11}$$

To obtain Eq. (E.11), we should note that the coupling constants have the $\mu$-factors in front of them as $\mu^{\frac{2}{3}\epsilon}\tilde{g}$, $\mu^{\frac{2}{3}\epsilon}\tilde{\Delta}_f$, and $\mu^\epsilon\tilde{\Delta}_b$, which we have ignored in the loop correction computations, for simplicity. To find $A_0$, $A_1$, $A_2$, $A_3$, $A_g$, $A_{\Delta_f}$, and $A_{\Delta_b}$, we gather all corresponding contributions from Sec. A, C, B, and D. As a result, we obtain

$$A_0 = -\tilde{g} - \tilde{\Delta}_f - \tilde{\Delta}_f - 0.54\tilde{g}^2 - 0.45\tilde{\Delta}_f\sqrt{\frac{\tilde{g}}{N}} - 0.45\tilde{\Delta}_b\sqrt{\frac{\tilde{g}}{N}},$$
$$A_1 = -\tilde{g} - 0.54\tilde{g}^2 - 16\tilde{\Delta}_f\sqrt{\frac{\tilde{g}}{N}} - 16\tilde{\Delta}_b\sqrt{\frac{\tilde{g}}{N}},$$
$$A_2 = -0.5\tilde{\Delta}_f - 0.5\tilde{\Delta}_f - 0.11\tilde{g}^2,$$
$$A_{\Delta_f} = -\tilde{\Delta}_f - \tilde{\Delta}_b - 0.75\frac{\tilde{\Delta}_b^2}{\tilde{\Delta}_f} - 0.23\tilde{g}^2 + 2.1\tilde{g}\tilde{\Delta}_f - 4.2\tilde{\Delta}_f\sqrt{\frac{\tilde{g}}{N}} - 4.2\frac{\tilde{\Delta}_b^2}{\tilde{\Delta}_f}\sqrt{\frac{\tilde{g}}{N}},$$
$$A_{\Delta_b} = 6\tilde{g} + 0.14\tilde{\Delta}_f + 18\tilde{g}^2 + 1.4\tilde{g}\tilde{\Delta}_f + 8.3\tilde{\Delta}_b\sqrt{\frac{\tilde{g}}{N}}, \tag{E.12}$$

where $A_g = A_2$ and $A_3 = 0$. Inserting Eq. (E.12) into Eq. (E.11), we obtain the critical exponents in Eqs. (17) and (18) in the main text.

## E.2  Critical exponents for thermodynamic quantities

Here, we explicitly compute the critical exponents for thermodynamic quantities, which are given in Eq. (37) in the main text with a heuristic argument. We start with the order parameter $m$ for the Ising-nematic order, which is defined as,

$$m \equiv -\frac{\partial f}{\partial h}\bigg|_{h \to 0} = \langle \Phi(x) + N(x) \rangle = m^{(1)} + m^{(2)}, \tag{E.13}$$

where $m^{(1)}$ and $m^{(2)}$ represent the fermion and boson contribution given as

$$m^{(1)} = \langle \phi(x) \rangle,$$
$$m^{(2)} = \langle j(x) \rangle = \int \frac{d^D k}{(2\pi)^D} \text{tr}[\gamma_{d-1} G(k)]. \tag{E.14}$$

By solving the Callan-Symanzik equation for $m^{(1)}$, which is given by

$$\left[\mu\partial_\mu + \beta_{\mathbf{F}} \cdot \nabla_{\mathbf{F}} + \frac{1}{2}(D - 1 + \gamma_\Phi)\right]m^{(1)} = 0, \tag{E.15}$$

we obtain the scaling behavior of $m^{(1)}$ as

$$m^{(1)} \sim \mu^{-\frac{1}{2}(D-1+\gamma_\Phi)} \sim (-r)^{\frac{\nu}{2}(D-1+\gamma_\Phi)}. \tag{E.16}$$

To find the scaling behavior of $m^{(2)}$, we consider the Callan-Symanzik equations for a fermion Green's function $G(k) = \langle \Psi(k)\bar{\Psi}(k) \rangle$, which is given by

$$\left[ k \cdot \nabla_k - \beta_{\mathbf{F}} \cdot \nabla_{\mathbf{F}} + 1 - \gamma_\Psi \right] G(k, \mu, \mathbf{F}) = 0. \tag{E.17}$$

The solution is represented as

$$G(k, \mu, \mathbf{F}) = \frac{1}{\mu^{\gamma_\Psi} |\delta_{\mathbf{k}}|^{1-\gamma_\Psi}} g(k_0 / |\delta_{\mathbf{k}}|^z). \tag{E.18}$$

Using this expression, we find the scaling behavior of $m^{(2)}$ as

$$m^{(2)} \sim \int \frac{d^D k}{(2\pi)^D} \frac{1}{|\delta_{\mathbf{k}}|^{1-\gamma_\Psi}} g\left( \frac{k_0}{|\delta_{\mathbf{k}}|^z}, \frac{r}{|\delta_{\mathbf{k}}|^{1/\nu}} \right) \sim (-r)^{\nu(D-1+\gamma_\Psi)}. \tag{E.19}$$

Using the values of $D = 5/2$, $\gamma_\Phi = 0$, and $\gamma_\Psi = 0.24$ at the DNFL fixed point, we obtain

$$m^{(1)} \sim (-r)^\nu, \qquad m^{(2)} \sim (-r)^{1.97\nu}. \tag{E.20}$$

We note that the bosonic contribution $m^{(1)}$ is much larger than the fermionic contribution $m^{(2)}$ near the critical point $r \approx 0$. This observation justifies ignoring the coupling of the external field with fermionic excitations.

We next compute the susceptibility for Ising-nematic order parameter, which is defined as

$$\chi \equiv \left. \frac{\partial^2 f}{\partial h^2} \right|_{h \to 0} = \int d^D x \, \langle \Phi(x)\Phi(0) + N(x)N(0) \rangle = \chi^{(1)} + \chi^{(2)}, \tag{E.21}$$

where $\chi^{(1)}$ and $\chi^{(2)}$ represent the fermion and boson contribution given as

$$\chi^{(1)} = \int d^D x \, \langle \phi(x)\phi(0) \rangle = \lim_{k \to 0} D(k),$$
$$\chi^{(2)} = \int d^D x \, \langle j(x)j(0) \rangle = \int \frac{d^D k}{(2\pi)^D} \langle j(k)j(-k) \rangle. \tag{E.22}$$

To find the scaling behavior of $\chi^{(1)}$, we consider the Callan-Symanzik equations for a boson Green's function $D(k) = \langle \phi(k)\phi(-k) \rangle$, which is given by

$$\left[ k \cdot \nabla_k - \beta_{\mathbf{F}} \cdot \nabla_{\mathbf{F}} + 1 - \gamma_\Phi \right] D(k, \mu, \mathbf{F}) = 0. \tag{E.23}$$

The solution is represented as

$$D(k, \mu, \mathbf{F}) = \frac{1}{\mu^{\gamma_\Phi} |k_y|^{2(1-\gamma_\Phi)}} d(k_0 / |k_y|^{2z}). \tag{E.24}$$

Using this expression, we find the scaling behavior of $\chi^{(1)}$ as

$$\chi^{(1)} = \lim_{k \to 0} \frac{1}{|k_y|^{2(1-\gamma_\Phi)}} d\left( \frac{k_0}{|k_y|^{2z}}, \frac{r}{|k_y|^{2/\nu}} \right) \sim |r|^{-\nu(1-\gamma_\Phi)}. \tag{E.25}$$

To find the scaling behavior of $\chi^{(2)}$, we consider the Callan-Symanzik equations for the correlation function $G^{(2)}(k) = \langle j(k)j(-k) \rangle$, which is given by

$$\left[ k \cdot \nabla_k - \beta_{\mathbf{F}} \cdot \nabla_{\mathbf{F}} + 2 + \gamma^{ms}_{\Delta_f} \right] G^{(2)}(k) = 0, \tag{E.26}$$

where $\gamma^{ms}_{\Delta_f} = \gamma_{\Delta_f} - 2\gamma_\Psi$. The solution is represented as

$$G^{(2)}(k) = \frac{1}{|\delta_{\mathbf{k}}|^{2+\gamma^{ms}_{\Delta_f}}} g^{(2)}\left( \frac{k_0}{|\delta_{\mathbf{k}}|^z}, \frac{r}{|\delta_{\mathbf{k}}|^{1/\nu}} \right). \tag{E.27}$$

Using this expression, we find the scaling behavior of $\chi^{(2)}$ as

$$\chi^{(2)} = \int \frac{d^D k}{(2\pi)^D} \frac{1}{|\delta_{\mathbf{k}}|^{2+\gamma^{ms}_{\Delta_f}}} g^{(2)}\left( \frac{k_0}{|\delta_{\mathbf{k}}|^z}, \frac{r}{|\delta_{\mathbf{k}}|^{1/\nu}} \right) \sim |r|^{\nu(D-2-\gamma^{ms}_{\Delta_f})}. \tag{E.28}$$

Using the value of $\gamma^{ms}_{\Delta_f} = -0.50$ at the DNFL fixed point, we find

$$\chi^{(1)} \sim |r|^{-\nu}, \qquad \chi^{(2)} \sim |r|^{\nu}. \tag{E.29}$$

We note that the bosonic contribution $\chi^{(1)}$ is again much larger than the fermionic contribution $\chi^{(2)}$ near the critical point $r \approx 0$. This observation again justifies ignoring the coupling of the external field with fermionic excitations.

## F  Ward identity

The effective field theory of Eq. (4) has a U(1) symmetry, given by $\Psi_j(k) \to e^{i\theta_v} \Psi_j(k)$. Associated with this symmetry, we derive the Schwinger-Dyson equation for $\langle \psi(x)\bar{\psi}(0) \rangle$ and find the following identity

$$\Gamma_{d-1}(p, 0) = \frac{\partial G^{-1}(p)}{\partial p_x}, \tag{F.1}$$

where $\Gamma_{d-1}(p+q, q)$ is the irreducible vertex function resulting from $\langle j_{d-1}(x')\psi(x)\bar{\psi}(0) \rangle$, and $G(p)$ is the fully renormalized fermion propagator. $j_{d-1} \equiv \bar{\psi}\gamma_{d-1}\psi$ is the conserved current related to the U(1) symmetry in the $(d-1)$ direction. The Ward identity of Eq. (F.1) implies that the vertex function for $\gamma_{d-1}$ and the fermion kinetic energy should be renormalized at the same rate. For the fermion-boson Yukawa coupling, where bosons are coupled to $j_{d-1}$ conserved currents, this equation implies the following relation

$$\gamma_g = 2\gamma_\psi, \tag{F.2}$$

which should be preserved in all loop corrections.

There is a similar identity for forward disorder scattering. To figure it out, we define $\gamma^{ss}_{\Delta_f} \equiv \gamma_{\Delta_f} - \gamma^{ms}_{\Delta_f}$, where $\gamma^{ss}_{\Delta_f}$ ($\gamma^{ms}_{\Delta_f}$) is the anomalous dimension involved with a single (multiple) scattering process. For example, in Table 3, the Feynman diagrams labeled as "FV1-3", "FV1-5", and "FV1-6" fall into the single scattering process while those labeled as "FV1-1", "FV1-2", and "FV1-4" fall into the multiple scattering process. Only $\gamma^{ss}_{\Delta_f}$ is subject to the Ward identity because the forward scattering acts effectively as a vertex function for $\gamma_{d-1}$ only in the single scattering process. Then, the Ward identity in Eq. (F.1) implies another relation,

$$\gamma^{ss}_{\Delta_f} = 4\gamma_\psi, \tag{F.3}$$

which should be preserved in all loop corrections.

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
