# Peer review of "Disordered non-Fermi liquid fixed point for two-dimensional metals at Ising-nematic quantum critical points"

_SciPost Physics, doi:SciPost Phys. 17, 059 (2024)_

## Round 2 · Referee Report · Anonymous (Referee 2) · 2024-6-23

Strengths
Report
Thank the authors for the response. I would recommend the publication of this work.
I would like to point out the following. In order to verify the existence of the disordered fixed point, there are further considerations which should be taken into account in the future study.
-
I notice that the authors have added a note in the end. In addition, the self-energy contribution to at least the two-loop order can affect the shape of the fermi surface and the dispersion on the Fermi surface. This can further affect the interaction effect.
-
As the fixed point at finite $\epsilon$ and finite disorder strength is not controlled, I think it is better to find a controlled large N expansion such that we can at least get correct $\epsilon_c$ at large N limit. In the current work, there are N flavors of fermions. Probably we can consider M flavors of bosons and do the large M expansion.
Recommendation
Publish (meets expectations and criteria for this Journal)

---

## Round 2 · Author Response

Report of Reviewer 1: The submission meets the criteria for publication in SciPost with a few minor revisions to clarify the effect of higher loops corrections on the disordered fixed point.
Our response: We would like to thank Reviewer 1 for the thoughtful review and the recommendation for publication after minor revisions. We value the constructive feedback and the suggestion to clarify the impact of higher-loop corrections on the disordered fixed point. Understanding the robustness of the disordered fixed point in the presence of higher-loop corrections is crucial to refining our results and guiding future studies.
Requested changes from Reviewer 1: 1- Addition of a discussion on the fate of the disordered fixed point to higher loop corrections, specifically how robust the critical value εc is to such corrections. A few comments on whether this critical value is prevented from being larger than the physical value ε = 0.5 are warranted.
Our response: To address the reviewer's concerns, we examined the effect of three-loop corrections on the critical value ϵc. Using a specific form of the three-loop beta functions, which we speculated from our two-loop-order results, we analyzed how these terms influence ϵc. We identified three possibilities regarding the existence of the disordered non-Fermi liquid fixed point.
First, our results suggest that three-loop antiscreening corrections for disorder scattering can increase ϵc, potentially destabilizing the disordered fixed point if ϵc exceeds ϵph, which is 0.5. Second, three-loop screening corrections can help stabilize the fixed point by reducing ϵc. Finally, as long as the magnitudes of the coefficients for the three-loop corrections are not too large, the critical value will remain below ϵph, thereby preserving the disordered fixed point within the perturbative regime.
These findings have been included in the revised manuscript, specifically in Figure 7 and the subsection “Higher-order corrections.” We also discuss the fate of the disordered fixed point, addressing the reviewer's question about whether the critical value could exceed the physical limit of ϵ = 0.5.
We sincerely thank the reviewer for the constructive comments, which have undoubtedly improved the clarity and scientific quality of our work. We look forward to the reviewer's final evaluation of our revised manuscript and hope for its formal acceptance for publication in SciPost Physics. Report of Reviewer 2: The author has investigated the effect of disorder at the Ising-nematic transition, claiming to have employed a controlled epsilon expansion to identify a new disordered fixed point at infinite N, up to the two-loop order. They further argue that this fixed point persists at finite N and with higher order corrections. This is a solid piece of work and is definitely publishable. However, I am still trying to understand more about this research. I will recommend its publication once my following questions are fully addressed:
Our response: We would like to thank Reviewer 2 for the thoughtful review and positive assessment of our manuscript. We appreciate the constructive feedback and the recommendation for publication. Below, we address all raised questions and comments to enhance the rigor, clarity, and scientific implications of our work.
Question 1 of Reviewer 2: 1. Is the calculation controlled around epsilon=0 and N=infty? This would imply that a double expansion in epsilon and 1/N is necessary. Was this paper actually employed this double expansion?
Our response: Our loop expansion is controlled around ϵ = 0 because higher-order corrections are neglected when deriving our two-loop beta functions. This can be formally validated by taking a small ϵ limit. Additionally, we used an extra parameter, N, representing the fermion flavor number, to control certain two-loop corrections that could hinder the emergence of the disordered fixed point. Therefore, we utilized a double expansion with two small parameters (ϵ and 1/N) to derive our two-loop beta functions.
Question 2 of Reviewer 2: 2. If the answer is yes, as the disordered fixed point manifest only at finite epsilon, does this suggest that the direct study of such a fixed point is not controlled?
Our response: In our study, the existence of the disordered fixed point requires finite values of ϵ > ϵc, making it challenging to directly apply a conventional scheme where a theory is solved with high accuracy for small ϵ values and then extrapolated to actual physical dimensions by setting ϵ = ϵph = 0.5. Because of this constraint, one must maintain ϵ > ϵc during the calculation, which might, then set ϵ = ϵph at the end.
Question 3 of Reviewer 2: 3. Do all higher-loop corrections vanish in the large N limit?
Our response: We do not expect all higher-order corrections to vanish in the large N limit. In the two-loop order, some diagrams, involving the mixing of Yukawa coupling and disorder scattering, exhibit an anomalous inverse power in N. These diagrams tend to vanish in the large N limit. However, other diagrams that don't follow this structure persist. Our expectation is that in a higher-order analysis, similar trends will appear, with some diagrams vanishing and others persisting in the large N limit.
Question 4 of Reviewer 2: 4. Will higher-loop corrections alter the location of the disordered fixed point? Considering that the presence or position of the fixed point is determined by the beta function's solutions, how can we ensure that these solutions consistently exist?
Our response: Higher-loop corrections could indeed alter the location of the disordered fixed point. Our qualitative three-loop analysis shows how each three-loop term impacts the position of the disordered fixed point. Furthermore, three-loop antiscreening corrections for disorder scattering can increase ϵc, potentially destabilizing the disordered fixed point when ϵc > ϵph. On the other hand, three-loop screening corrections help preserve the fixed point in a perturbative regime. These results suggest that the fixed point's presence might require a thorough examination of higher-loop corrections to ensure consistency within the current theoretical framework. Please refer to Figure 7 and the subsection “Higher-order corrections” for further details.
Our concluding remark: We thank the reviewer for the insightful questions and comments. They have helped clarify our approach and improve the scientific impact of our work. We hope our detailed responses adequately address the reviewer’s concerns and look forward to the reviewer’s final evaluation of our revised manuscript.

---

## Round 2 · List of Changes

-
We added a new figure (Figure 7) to illustrate the results of the three-loop-order analysis, highlighting the impact of higher-order corrections.
-
The subsection titled “1. Higher-order corrections” has been rewritten to discuss the stability of the disordered fixed point against three-loop corrections. This section now considers whether these corrections could cause the critical value, ϵc, to exceed the physical limit of ϵ = 0.5.
-
A new paragraph has been added to explain the role of two-loop corrections in “C. Role of two-loop corrections” which is utilized in predicting the disordered non-Fermi liquid fixed point's stability under higher-loop corrections.
-
Minor corrections have been made in Eqs. (21), (31), and (32), changing “0.66” to “0.67,” which leads to improved numerical accuracy in the critical exponents. Figure 5 has been updated accordingly.
-
A scientific note from a discussion with an expert scholar has been included after the acknowledgements section. This correspondence provides important insights that could be instrumental in refining our results in future studies.

---

## Editorial Decision

published